# Reciprocal inhibition of NOTCH and SOX2 shapes tumor cell plasticity and therapeutic escape in triple-negative breast cancer

Morgane Fournier[1,3], Joaquim Javary [ID][1,3], Vincent Roh[2], Nadine Fournier [ID][1,2] & Freddy Radtke [ID][1✉]

## Abstract

Cancer cell plasticity contributes significantly to the failure of chemo- and targeted therapies in triple-negative breast cancer (TNBC). Molecular mechanisms of therapy-induced tumor cell plasticity and associated resistance are largely unknown. Using a genome-wide CRISPR-Cas9 screen, we investigated escape mechanisms of NOTCH-driven TNBC treated with a gamma-secretase inhibitor (GSI) and identified SOX2 as a target of resistance to Notch inhibition. We describe a novel reciprocal inhibitory feedback mechanism between Notch signaling and SOX2. Specifically, Notch signaling inhibits SOX2 expression through its target genes of the *HEY* family, and SOX2 inhibits Notch signaling through direct interaction with RBPJ. This mechanism shapes divergent cell states with NOTCH positive TNBC being more epithelial-like, while SOX2 expression correlates with epithelial-mesenchymal transition, induces cancer stem cell features and GSI resistance. To counteract monotherapy-induced tumor relapse, we assessed GSI-paclitaxel and dasatinib-paclitaxel combination treatments in NOTCH inhibitor-sensitive and -resistant TNBC xenotransplants, respectively. These distinct preventive combinations and second-line treatment option dependent on NOTCH1 and SOX2 expression in TNBC are able to induce tumor growth control and reduce metastatic burden.

**Keywords** Notch Signaling; SOX2; Therapy Resistance; TNBC; Tumor Cell Plasticity
**Subject Categories** Cancer; Stem Cells & Regenerative Medicine

## Introduction

Triple-negative breast cancer (TNBC) is the most aggressive subtype of breast cancer (BC), classically defined by the lack of expression of hormone receptors (HR, estrogen and progesterone receptors) and human epidermal growth factor receptor 2 (HER2).

Luminal (HR$^+$/HER2$^-$) cancers account for ~60–70%, HER2$^+$ for 15%, and TNBC for the remaining 15–20% of BC patients (Nolan et al, 2023). While HR$^+$ and HER2$^+$ BC patients can be treated with targeted therapies and have relatively favorable prognoses, TNBC patients are primarily treated with chemotherapy (Nolan et al, 2023). Early-stage TNBC patients are treated with neoadjuvant chemotherapy, sometimes combined with platinum or immune checkpoint inhibitors (Bianchini et al, 2022). Advanced-stage disease and metastatic TNBC patients respond poorly to chemotherapy, have poor prognoses and reduced 5–year overall survival (OS) compared to nonmetastatic TNBC patients (Hsu et al, 2022).

Although TNBC is largely defined by a uniform lack of HR and HER2 expression, it represents a heterogenous disease. Transcriptomic analysis of TNBC specimens identified four molecular subtypes (basal-like immune activated, basal-like immune suppressed, mesenchymal and luminal androgen receptor type), which associate with differential response to standard-of-care chemotherapy (Lehmann et al, 2011, 2016; Burstein et al, 2015). Furthermore, prognostic and predictive biomarkers as well as therapeutically actionable targets were found using whole exome and transcriptome data derived from TNBC specimens allowing broadening and personalization of therapeutic options (Sukumar et al, 2021; Bianchini et al, 2022). These sequencing efforts identified dysregulation and mutations of various biomarkers including *NOTCH* receptors, exposing exploitation opportunities for targeted therapy in subgroups of TNBC patients (Sukumar et al, 2021). Activating mutations within *NOTCH1, 2* and *3* genes (Wang et al, 2015) as well as chromosomal deletions and translocations lead to expression of truncated, dominant-active forms of NOTCH1 and 2 receptors (Robinson et al, 2011; Stoeck et al, 2014). Aberrant Notch signaling has been linked to cancer stem cells (CSC) maintenance, tumor progression and chemoresistance in various cancers, (Giuli et al, 2019; BeLow and Osipo, 2020) and NOTCH signatures have been developed to predict the therapeutic response of BC patients, including those of the TNBC subgroups (Omar et al, 2023; Braune et al, 2024). Inhibition of the Notch cascade is achieved with γ-secretase inhibitors (GSI) which inhibit Notch signaling by blocking the proteolytic activity of γ-secretase. This prevents liberation and translocation of the Notch intracellular domain

[1]Ecole Polytechnique Fédérale de Lausanne (EPFL), School of Life Sciences, Swiss Institute for Experimental Cancer Research (ISREC), Swiss Cancer Center Leman (SCCL), Station 19, CH-1015 Lausanne, Switzerland. [2]Translational Data Science Facility, Swiss Institute of Bioinformatics (SIB), AGORA Cancer Research Center, CH-1011 Lausanne, Switzerland. [3]These authors contributed equally: Morgane Fournier, Joaquim Javary. ✉E-mail: Freddy.Radtke@epfl.ch

 

(NICD) to the nucleus and activation of the RBPJ transcription complex (Han and Shen, 2012). Recently, the FDA approved a GSI (Nirogacestat) based on a successful phase 3 clinical study in patients with desmoid tumors (Gounder et al, 2023). With the FDA approval of Nirogacestat, a promising avenue emerges for the use of this class of Notch inhibitors in other cancer indications, including TNBC. Consequently, a phase 2 study of a GSI (NCT04461600, TENACITY) has been initiated in patients with NOTCH-activated recurrent or metastatic TNBC.

Although increasing the arsenal of targeted therapies broadens the therapeutic options of difficult-to-treat cancer patients, it is accompanied by the risk of intrinsic or acquired resistance when used as single agent. Tumor relapse and therapeutic resistance can emerge by various trajectories. First, tumors can be inherently composed of molecularly diverse cancer cells and/or the cells may acquire mutations over time, increasing tumor cell heterogeneity and thus the likelihood of therapy resistance. Second, acquired resistance imposed by drug-mediated selective pressure causes tumor cells to switch between cell states via cellular plasticity. While a switch between cell states can be caused by genetic alterations, they are more often caused by transcriptional or epigenetic changes. Tumor cell plasticity, be it inherent or acquired, contributes to intratumoral heterogeneity and can facilitate multiple aspects of cancer progression, including cell fitness, metastatic potential and resistance to chemo- or targeted therapies (Petrovic et al, 2019; Labrie et al, 2022).

Here, we investigated potential resistance mechanisms in NOTCH-driven, GSI-treated human TNBC using a genome-wide CRISPR-Cas9 screen. We identified a reciprocal inhibitory feedback mechanism between Notch signaling and the pluripotent associated stem cell (SC) transcription factor (TF) SOX2, which shapes tumor cell plasticity and associated therapeutic response to GSI. Moreover, we provide an experimentally based rational for therapeutic options of combination therapies for GSI-sensitive and resistant TNBC.

## Results

### Chronic exposure of NOTCH-driven TNBC cells to GSI induces drug resistance associated with EMT and CSC features

Next-generation sequencing efforts have identified mutations and dysregulation of NOTCH receptors in subgroups of TNBC patients, which can be employed for targeted therapy (Sukumar et al, 2021). BC patient dataset analysis using The Cancer Genome Atlas (TCGA) and the Molecular Taxonomy of Breast Cancer International Consortium (METABRIC) revealed an enrichment for activating mutations and gene amplifications of NOTCH receptors in TNBC compared to luminal and HER2+ patients (Fig. EV1A,B). The identification of genetic NOTCH alterations might be indicative for NOTCH-driven tumor growth, independently of the different BC subgroups (luminal, HER2+ or TNBC). These NOTCH variants, however, were not predictive for OS in TNBC (Fig. EV1C,D). To assess whether active Notch signaling is predictive for disease outcome of TNBC patients, we established a novel NOTCH signature. We combined previously published RNAseq datasets based on GSI-washout and NOTCH gain-of-function experiments in TNBC cell lines (Stoeck et al, 2014; Petrovic et al, 2019) to establish a TNBC-specific NOTCH signature

consisting of 77 genes (Table EV1). Using this newly established NOTCH signature, we investigated BC patient data from the TCGA and METABRIC databases across subgroups (luminal, HER2+ and TNBC) and categorized patients into three groups based on expression scores of the signature (NOTCH$^{High}$–NOTCH$^{Int}$–NOTCH$^{Low}$). We observed a notable enrichment of NOTCH$^{High}$ signature among TNBC patients compared to other subgroups, comprising ~50% of the cohort (Figs. 1A and EV1E).

Moreover, a NOTCH$^{High}$ signature within TNBC patients associated with poor OS in the two independent TCGA and METABRIC patient cohorts and decreased recurrence-free survival (RFS) in the METABRIC cohort, compared to patients with a NOTCH$^{Low}$ signature (Figs. 1B and EV1F,G). Efficient targeting of the Notch pathway could, therefore, be beneficial for NOTCH$^{High}$ signature TNBC patients.

The long-term efficacy of targeted monotherapies in cancer patients, however, is often thwarted by the development of drug resistance (Labrie et al, 2022). To predict mechanisms by which NOTCH-driven TNBC cells overcome NOTCH inhibition, we used the patient-derived TNBC cell line MB157 as model system. Conflicting reports describing NOTCH1 chromosomal aberration prompted us to perform transcriptomic analysis of naive MB157 cells (Stoeck et al, 2014; Paroni et al, 2020) to reassess the NOTCH1 gene locus. Mapping of RNAseq reads for NOTCH1 to the genome revealed that exons 2–27 were absent, indicating an intergenic deletion within NOTCH1 causing loss of the entire extracellular domain of the NOTCH1 receptor (Fig. 1C). MB157 cells thus express a ligand-independent constitutively active form of NOTCH1 (N1-ICD), which still harbors the transmembrane domain and is thus sensitive to GSI treatment (Fig. EV1H). Short-term GSI treatment of MB157 resulted in cell growth inhibition and downregulation of NOTCH target genes including MYC, CCND1, and HES1 (Fig. 1D–F). Chronic exposure of human TNBC cells (MB157) over 25 weeks to GSI resulted in initial cell growth inhibition for ~4 weeks, after which the cells became resistant and started to grow despite the presence of GSI (Fig. 1G). The GSI-resistant MB157 (MB157R) cells were not responsive to GSI, even at 10 μM (Fig. 1H). To characterize GSI-sensitive versus resistant cells, we examined N1-ICD levels. N1-ICD was strongly expressed in MB157 cells but drastically reduced in MB157R cells, while MYC expression was only slightly diminished, indicating that MYC expression is maintained or compensated for in a NOTCH1-independent manner (Fig. 1I). Furthermore, MB157 cells lost their cobblestone phenotypic appearance over the time course of GSI treatment adapting an elongated and fibroblast-like phenotype by the end of the treatment (Fig. 1J).

Gene expression analysis of MB157 and MB157R cells provided insight about the differences between drug-sensitive and resistant cells. Hallmark gene set enrichment analysis (GSEA) showed that MB157R cells were enriched for genes associated with EMT (Fig. EV1I; Dataset EV1), ranking among the top three upregulated signatures. Further analysis focused on BC signatures indicated a downregulation of the NOTCH signature, along with upregulation of EMT and SC signatures in MB157R cells. Moreover, a comparative analysis of luminal, basal, and mesenchymal signatures revealed a mesenchymal transcriptomic signature of GSI-resistant cells compared to their GSI-sensitive counterparts (Fig. 1K). We confirmed increased expression of multiple EMT and SC markers (Mannello, 2013; Ye et al, 2017), including N-Cadherin, Vimentin, SLUG, TWIST and CD49f in MB157R versus

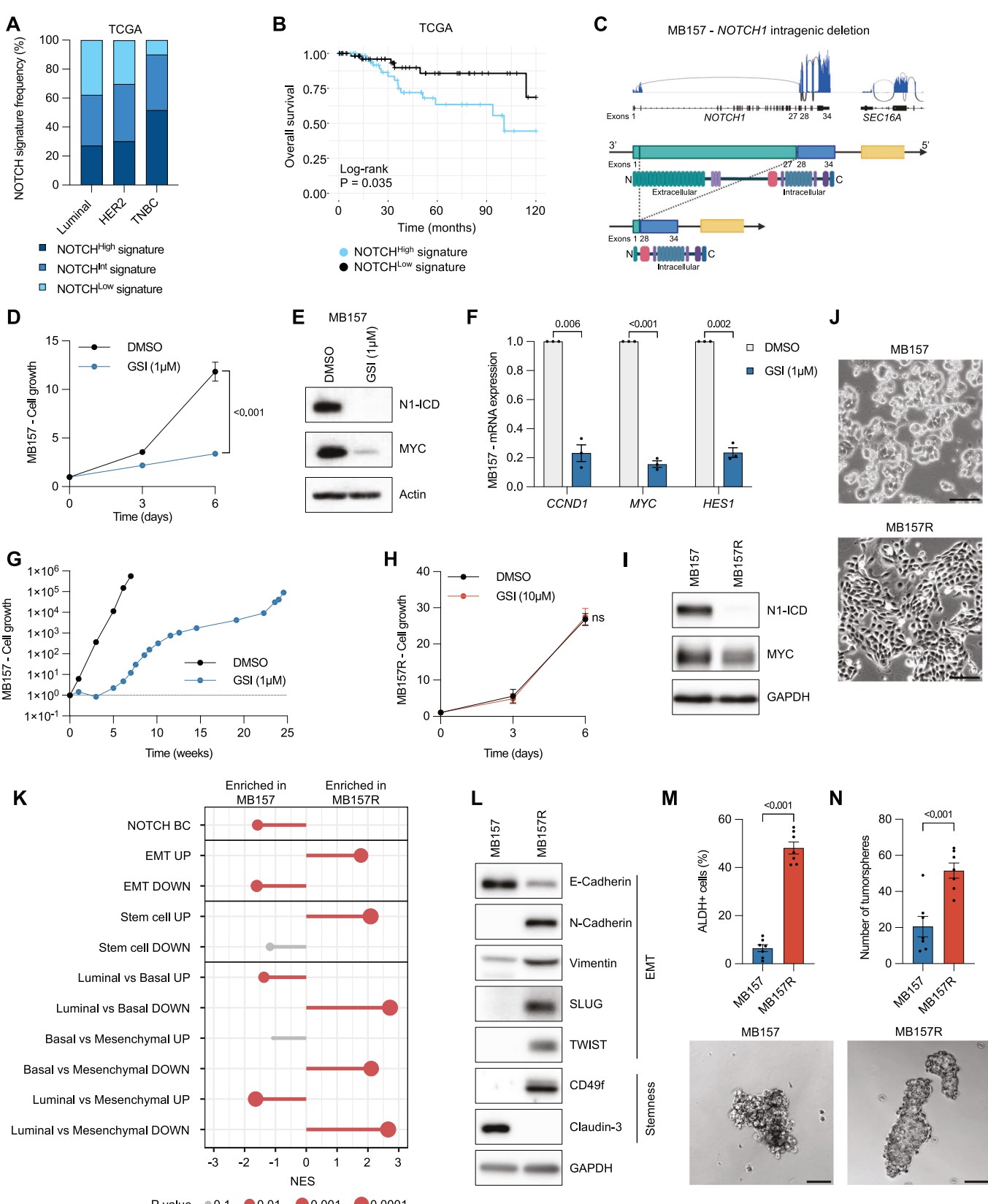

**Figure 1. Chronic exposure of NOTCH-driven TNBC cells to GSI induces drug resistance associated with EMT and CSC features.**

(A) Frequency of NOTCH$^{High/Int/Low}$ signature in luminal ($n = 780$), HER2 ($n = 116$) and TNBC ($n = 191$) patients from TCGA dataset ($n = 1087$). (B) OS of TNBC patients from TCGA dataset with NOTCH$^{High}$ ($n = 65$) or NOTCH$^{Low}$ ($n = 60$) signature. (C) Sashimi plot of *NOTCH1* mRNA expression (blue: reads, gray: splicing events), scheme of *NOTCH1* intergenic deletion and protein domain structure of NOTCH1 in MB157 TNBC cells. (D) Cell proliferation assay of MB157 cells as indicated, $n = 3$. (E) Representative immunoblotting of N1-ICD and MYC, and (F) Relative mRNA expression of *CCND1*, *MYC* and *HES1* in MB157 cells as indicated, 24 h post treatment, $n = 3$. (G) Relative cell growth of MB157 under continuous GSI exposure. (H) Cell proliferation assay of MB157R cells as indicated, $n = 3$. (I) Representative immunoblotting of N1-ICD and MYC derived from MB157 and MB157R cells. (J) Representative images of MB157 and MB157R cells in culture, scale = 50 µm. (K) GSEA of BC-specific signatures for Notch signaling, EMT, SC, luminal, basal and mesenchymal from RNAseq analysis of MB157R compared to MB157 cells, $n = 3$. (L) Representative immunoblotting of EMT and stemness markers derived from MB157 and MB157R cells. (M) Proportion of ALDH$^+$ cells analyzed by flow cytometry in MB157 and MB157R cells, $n = 7$. (N) Quantification and images of tumorspheres derived from MB157 and MB157R cells, $n = 7$. Scale = 100 µm. Data from biological replicates are represented as mean ± SEM. Log-rank test (B), Student *t* test (F, M, N), two-way ANOVA (D, H) or permutation test (K) were used to determine *P* values (ns, not significant). Source data are available online for this figure.

MB157 cells at the mRNA and protein level (Figs. 1L and EV1J). Note that Claudin-3 expression is downregulated in MB157R, in agreement with its expression being inversely correlated to stemness (Pommier et al, 2020). To further validate an upregulated SC signature in MB157R cells, we analyzed the expression of an additional CSC marker Aldehyde dehydrogenase (ALDH) (Panigoro et al, 2020; Khoury et al, 2012) and performed tumorsphere assays as a surrogate for CSC capacity. There were 7.4-fold more ALDH$^+$ MB157R cells compared to naive MB157 cells, and thus were more able to form tumorspheres (Fig. 1M,N). Taken together, these results show that the Notch inhibitor-resistant MB157R cells have undergone EMT and exhibit increased CSC features.

## SOX2 mediates resistance to GSI in TNBC inhibiting Notch signaling, promoting EMT and CSC features

To identify genes implicated in mediating resistance to pharmacological NOTCH inhibition, we performed a genome-wide loss-of-function CRISPR/Cas9 screen (Shalem et al, 2014) using the GSI-resistant MB157R cells. MB157R cells, stably expressing Cas9, were infected with the human GeCKOv2 CRISPR libraries, and treated with either vehicle (VHC) or GSI for 14 days (Fig. 2A). sgRNAs targeting 21 genes were identified as significantly depleted ($P < 0.01$, log2Fc<-1) in GSI versus VHC-treated MB157R cells (Fig. 2B; Dataset EV2). The robust rank aggregation (RRA) method was used to identify genes preferentially lost in response to GSI treatment. We focused on *SOX2* as candidate gene for mediating resistance to NOTCH inhibition as it ranked among the top three genes that were negatively selected in GSI versus VHC (Fig. EV2A). Moreover, SOX2 is a known pluripotency-associated SC TF linked to a multitude of cancer types and cancer cell traits such as proliferation, migration, resistance to established cancer therapies and expression in CSCs (Novak et al, 2020; Liu et al, 2018). To characterize the potential role of SOX2 in GSI-resistant cells we first examined protein expression of SOX2 and N1-ICD in MB157 and MB157R cells. While SOX2 expression is low in N1-ICD positive MB157 cells, SOX2 is strongly expressed in MB157R cells, which coincides with low N1-ICD expression (Figs. 2C and EV2B). To assess the function of SOX2 in MB157R cells, we performed siRNA-mediated knockdown experiments. SOX2 knockdown resulted in reduced cell growth compared to siCtrl (Fig. 2D), characterized by increased cell numbers in G0/G1 and reduced cell numbers in S-phase (Fig. EV2C), indicating that proliferation of MB157R cells is in part dependent on SOX2. Importantly, we observed increased expression of N1-ICD in SOX2 knockdown

MB157R cells, which correlated with reduced expression of EMT and SC markers (Fig. 2E) and reduced numbers of tumorspheres (Fig. EV2D). Together, these results suggest that SOX2 expression in MB157R cells can directly or indirectly inhibit NOTCH1 expression thereby influencing proliferation, plasticity of TNBC and sensitivity to GSI. To test this hypothesis, we performed complementary gain-of-function experiments and induced expression of SOX2 (iSOX2) or empty vector control (iEV) in N1-ICD positive MB157 cells. SOX2 overexpression for 72 h was sufficient to reduce N1-ICD and Claudin-3 expression, while expression of the EMT marker N-Cadherin was increased (Fig. 2F). Interestingly, SOX2 washout revealed that SOX2-mediated effects were reversible. Moreover, SOX2 expression is sufficient to render cell growth of naive MB157 cells resistant to GSI treatment (Fig. 2G). GSEA showed that genes upregulated by SOX2 induction in MB157 were associated with EMT, stemness and a mesenchymal BC cell state while genes associated with Notch signaling were reduced (Figs. 2H and EV2E; Dataset EV1). To determine whether the enhanced capacity of MB157R cells to form tumorspheres compared to MB157 (Fig. 1N) is partially mediated through SOX2, we compared tumorsphere formation and size of iSOX2 or iEV MB157 cells. Tumorsphere numbers and size were significantly increased in SOX2-expressing MB157 compared to control cells (Fig. 2I), indicating that SOX2 is sufficient to induce SC traits.

These findings imply that SOX2-mediated inhibition of Notch signaling and associated induction of EMT and SC traits play an important role in inducing resistance to Notch inhibition in naive MB157 TNBC. To ensure that these observations are not specific to one particular TNBC cell line, we verified the NOTCH-SOX2 interplay and associated phenotypes in two additional TNBC cell lines. First, we used the NOTCH1-driven HCC1599 TNBC cells, which also express a truncated dominant-active NOTCH1 receptor, and verified that cell growth and N1-ICD itself, as well as the expression of some of its downstream target genes are sensitive to GSI-mediated NOTCH inhibition (Fig. EV2F–I and as previously shown (Robinson et al, 2011; Stoeck et al, 2014)). Similarly, expression of SOX2 for 72hrs in HCC1599 TNBC was sufficient to downregulate N1-ICD and Claudin-3, as well as to upregulate the EMT marker N-Cadherin (Fig. EV2J,K). Moreover, SOX2 expression rendered HCC1599 cells resistant to GSI-mediated growth inhibition (Fig. EV2L). We wanted to further confirm the ability of SOX2 to inhibit Notch signaling using a loss-of-function approach. We therefore selected the TNBC cell line HCC1806, which expresses SOX2 and low levels of N1-ICD, thus being insensitive to GSI (Fig. EV2M). siRNA-mediated knockdown

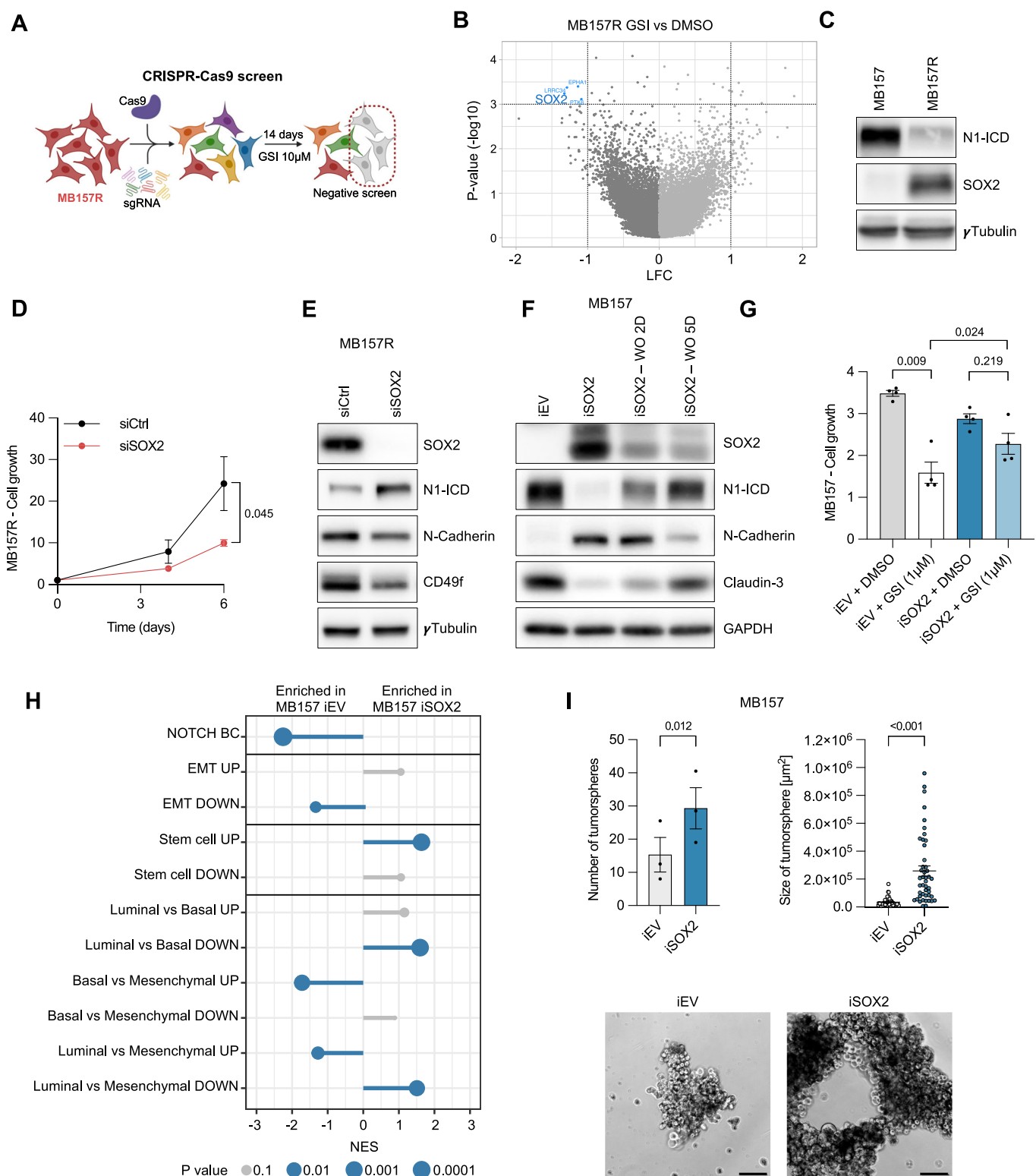

of SOX2 in HCC1806 cells resulted in reduced cell growth, associated with increased N1-ICD and reduced SLUG expression, as well as reduced numbers of tumorspheres (Fig. EV2N–P). The combined findings of the reciprocal loss- and gain-of-function experiments using different TNBC cell lines strongly suggest that the ability of SOX2 to inhibit Notch signaling is not a specific feature of MB157 cells, but is likely to be a more general phenomenon in TNBC cells.

◄

**Figure 2.   SOX2 mediates resistance to GSI in TNBC inhibiting Notch signaling, promoting EMT and CSC features.**

(**A**) Scheme of genome-wide CRISPR-Cas9 screen in MB157R cells. (**B**) Volcano plot of genes differentially expressed in MB157R cells from the CRISPR-Cas9 screen, $n = 3$. (**C**) Representative immunoblotting of N1-ICD and SOX2 derived from MB157 and MB157R cells. (**D**) Cell proliferation assay of MB157R cells transfected with siRNA SOX2 or control, $n = 3$. (**E**) Representative immunoblotting of N1-ICD, SOX2, and EMT/Stemness markers derived from MB157R cells as indicated or (**F**) iSOX2 or iEV MB157 cells (72 h), or SOX2 washout (WO) after 2 or 5 days. (**G**) Cell proliferation assay of MB157 cells as indicated, at day 6, $n = 4$. (**H**) GSEA of BC-specific signatures for Notch signaling, EMT, SC, luminal, basal, and mesenchymal from RNAseq analysis of iSOX2 compared to iEV MB157 cells, $n = 3$. (**I**) Quantification and size of tumorspheres with representative images derived from iSOX2 or iEV MB157 cells, $n = 3$. Scale $= 100$ μm. Data from biological replicates are represented as mean ± SEM. MAGeCK test (**B**), two-way ANOVA (**D**), one-way ANOVA (**G**), permutation test (**H**) or Student $t$ test (**I**) were used to determine $P$ values. Source data are available online for this figure.

## SOX2 interferes with the RBPJ activation complex and represses *NOTCH1* and NOTCH target genes

To investigate the mechanism by which SOX2 can repress Notch signaling, we performed ChIP-seq for RBPJ and SOX2 in iEV MB157, iSOX2 MB157, iEV MB157R, and inducible N1-ICD-expressing (iN1-ICD) MB157R cells. ChIP-seq for N1-ICD was not performed due to the limited quality of the commercially available anti-N1-ICD antibodies. Interestingly, in MB157R cells, 47% of the RBPJ peaks overlapped genome-wide with SOX2 peaks (within a range of 500 bps), and reciprocally, 57% of the SOX2 peaks overlapped with RBPJ peaks. Within this overlap of 21755 peaks, 71% of them contained both SOX2 and RBPJ BSs. The RBPJ-SOX2 peak overlap was significantly higher compared to a peak overlap between RBPJ and H3K27ac, indicating that the RBPJ-SOX2 peak overlap is nonrandom (Fig. 3A,B). The peak overlap was particularly evident in the *NOTCH1* promoter region, and in promoters and enhancers of NOTCH target genes, including *HES1*, *CCND1* and *MYC* (Fig. 3C,D; Appendix Figs. S1 and 2). While expression of SOX2 in MB157 cells induced SOX2 peaks similar to the ones in GSI-resistant MB157R cells, expression of N1-ICD in MB157R cells blunted SOX2 peaks to the same level as the ones identified in MB157 cells (Fig. 3C,D; Appendix Figs. S1 and 2). The close vicinity of SOX2 and RBPJ TF BSs in promoters and enhancers of *NOTCH1* and NOTCH target genes raises the possibility that SOX2 and RBPJ may physically interact, and thereby interfere with the generation of a productive N1-ICD-induced RBPJ TF complex. To test this hypothesis, we performed co-immunoprecipitation (IP) experiments for RBPJ and HA-tagged SOX2 in SOX2-expressing MB157 cells as well as for endogenous RBPJ and SOX2 in MB157R cells. IP of RBPJ was able to retrieve SOX2 and reciprocally, HA-tagged or endogenous SOX2 IP was able to retrieve RBPJ, indicating that SOX2 and RBPJ can physically interact in these settings (Fig. 3E). To further test that this physical interaction may inhibit N1-ICD induced transcription, we performed luciferase reporter assays using an artificial promoter with RBPJ BSs (Lehal et al, 2020). MB157 cells produced a strong GSI-sensitive luciferase signal, while iSOX2 expression resulted in significantly reduced RBPJ/N1-ICD driven luciferase signal (Fig. 3F). These findings strongly corroborate a working model where SOX2 interaction with RBPJ interferes with the formation of a functional N1-ICD/RBPJ transactivation complex thus repressing Notch signaling.

## Reciprocal SOX2 inhibition is mediated through downstream Notch transcriptional repressors of the HEY family

While SOX2 appears sufficient to inhibit Notch signaling and induce EMT and SC traits, we observed a reciprocal expression pattern for N1-ICD and SOX2 across all four cell lines tested.

Specifically, cells expressing high levels of N1-ICD (MB157 and HCC1599) have low levels of SOX2, EMT, and SC markers. In contrast, MB157R and HCC1806 cells show high expression of SOX2, EMT and SC markers, but low levels of N1-ICD (Fig. EV3A). Furthermore, N1-ICD induction in MB157R markedly attenuated SOX2 ChIP-seq peaks, suggesting that N1-ICD can inversely inhibit SOX2 expression (Fig. 3C,D; Appendix Figs. S1 and 2) and mitigate the induction of EMT and stemness features.

To test this hypothesis, we induced the expression of N1-ICD in MB157R and HCC1806 cells. This resulted in reduced expression of SOX2, EMT and SC markers, at both the transcriptional and protein level, as well as reduced cell growth in both TNBC cell lines (Figs. 4A,B and EV3B,F). Interestingly, N1-ICD washout experiments revealed that N1-ICD mediated effects were reversible. SOX2 expression and upregulation of EMT and SC markers were fully rescued within 5 days after washout of N1-ICD induction (Fig. 4A). Expression of N1-ICD in both MB157R and HCC1806 cells also reduced the number and size of tumorspheres (Figs. 4C and EV3G). These results were recapitulated in another independent SOX2-expressing TNBC cell line BT-549, (Fig. EV3H–J). N1-ICD expression correlated with decreased *SOX2* transcripts (Fig. EV3B,D,I), suggesting that N1-ICD inhibits *SOX2* expression at the transcriptional level. We next investigated how N1-ICD could repress transcription of *SOX2*. Although N1-ICD is known for its ability to activate gene expression, some of its direct downstream target genes are transcriptional repressors of the *HES* and *HEY* family (Weber et al, 2014; Fischer and Gessler, 2007). Thus, we assessed the expression of *HES* and *HEY* gene family members in MB157 and MB157R cells to identify candidates for *SOX2* repression. We hypothesized that *SOX2* repressors would be severely repressed in MB157R cells compared to MB157 cells. Accordingly, levels of *HEYL*, *HEY2* and *HES5* were dramatically reduced in MB157R, while *HES1* and *HES4* transcripts were only reduced by 60 and 50% (Fig. EV3K). While SOX2 was downregulated by expression of N1-ICD in MB157R and HCC1806 cells, siRNA-mediated knockdown of HEY2 and HEYL but not HES5 rescued SOX2 transcription and protein expression (Figs. 4D,E and EV3L). These results indicate that HEY2 and/or HEYL are implicated in the transcriptional repression of *SOX2*.

## Escape of GSI-mediated in vivo tumor growth control by TNBC tumor cell plasticity

To assess whether the finely tuned balance between NOTCH and SOX2 shaping tumor cell plasticity that we observed in vitro also occurs in vivo and potentially limits tumor growth control by GSI, two GSI-sensitive (HCC1599 and MB157) and two GSI-resistant (MB157R and HCC1806) GFP-luciferase expressing TNBC cell

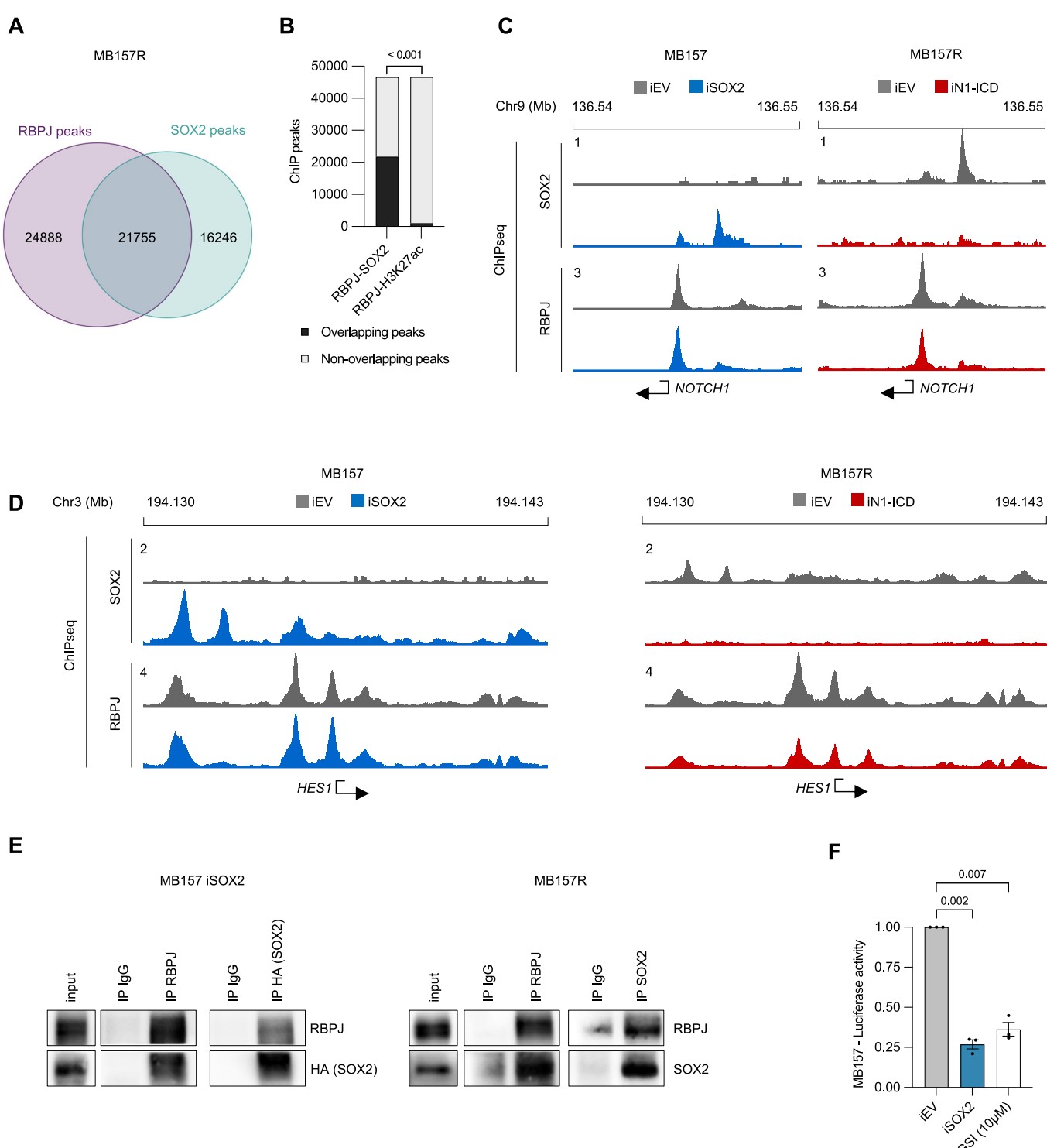

**Figure 3. SOX2 interferes with the RBPJ activation complex and represses *NOTCH1* and NOTCH target genes.**

(A) Overlap of ChIP peaks for RBPJ and SOX2 in MB157R cells ($n = 3$). (B) Genome-wide ChIP overlapping peaks for RBPJ-SOX2 compared to RBPJ-H3K27ac. (C) ChIP peaks for HA-tagged SOX2 or SOX2 and RBPJ on *NOTCH1* and (D) *HES1* promoter in MB157 and MB157R cells as indicated. The y-axis represents reads per million mapped reads. (E) Representative immunoblotting of IP using anti-HA (SOX2) or anti-SOX2 antibodies, and IP of HA (SOX2) or SOX2 using anti-RBPJ antibodies in iSOX2 MB157 and MB157R cells. (F) Luciferase reporter assay for RBPJ reporter in MB157 cells as indicated, 24 h after iSOX2 or GSI treatment, $n = 3$. Data from biological replicates are represented as mean ± SEM. Fisher test (B) and one-way ANOVA (F) were used to determine *P* values. Source data are available online for this figure.

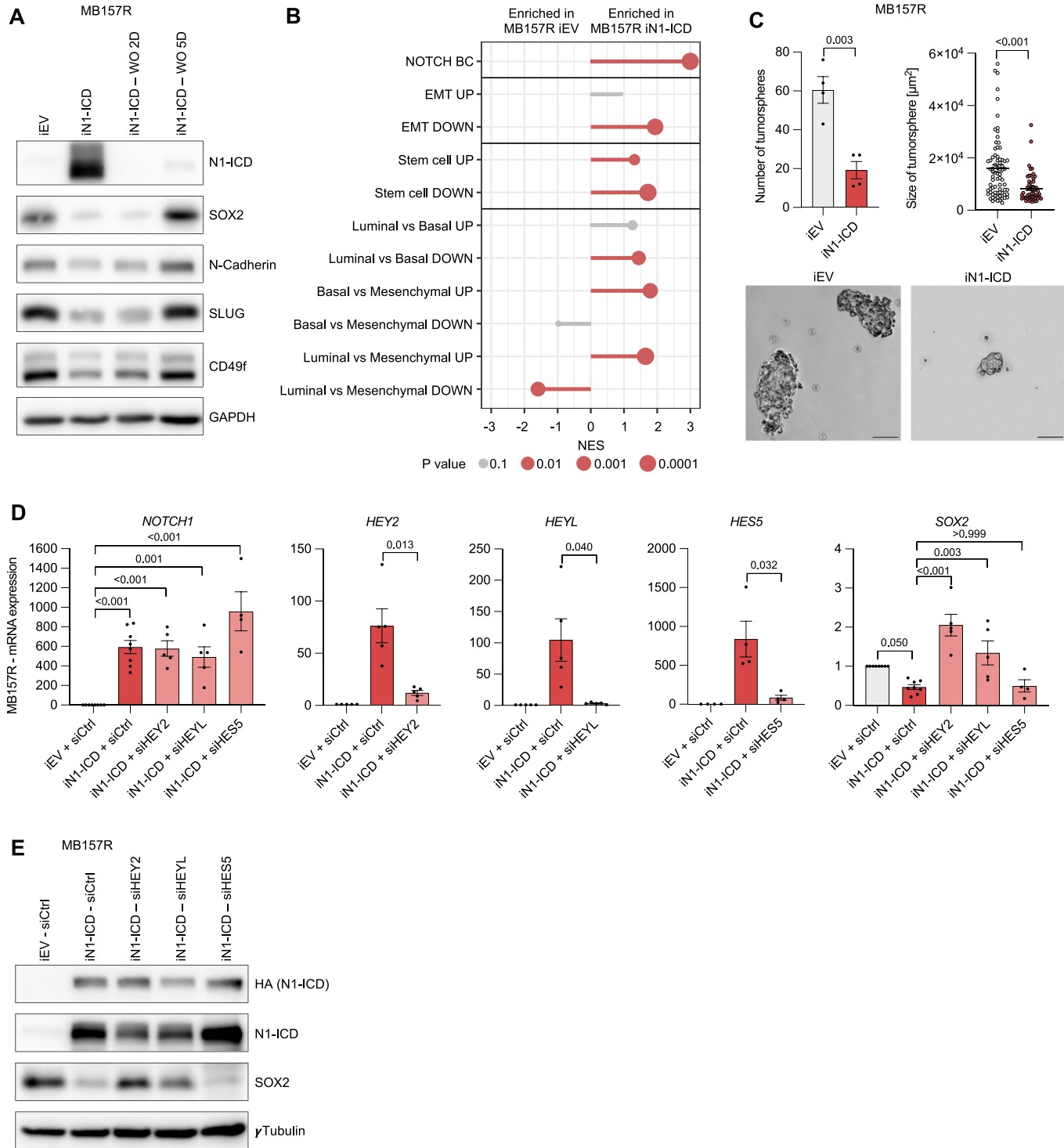

**Figure 4. Reciprocal SOX2 inhibition is mediated through Notch downstream transcriptional repressors of the HEY family.**

(A) Representative immunoblotting of N1-ICD, SOX2 and EMT/Stemness markers derived from iN1-ICD or iEV MB157R cells (72 h), or N1-ICD washout (WO) after 2 or 5 days. (B) GSEA of BC-specific signatures for Notch signaling, EMT, Stem Cell, luminal, basal and mesenchymal from RNAseq analysis of iN1-ICD compared to iEV MB157R cells, $n = 3$. (C) Number and size of tumorspheres with representative pictures derived from iN1-ICD or iEV MB157R cells, $n = 4$. Scale $= 100\,\mu m$. (D) Relative mRNA expression of *NOTCH1*, *HEY2*, *HEYL*, *HES5*, and *SOX2* in MB157R cells as indicated, $n = 4$–5. (E) Representative immunoblotting of HA (N1-ICD), N1-ICD, and SOX2 derived from MB157R cells as indicated. Data from biological replicates are represented as mean ± SEM. Permutation test (B), Student $t$ test (C) or one-way ANOVA (D) were used to determine $P$ values. Source data are available online for this figure.

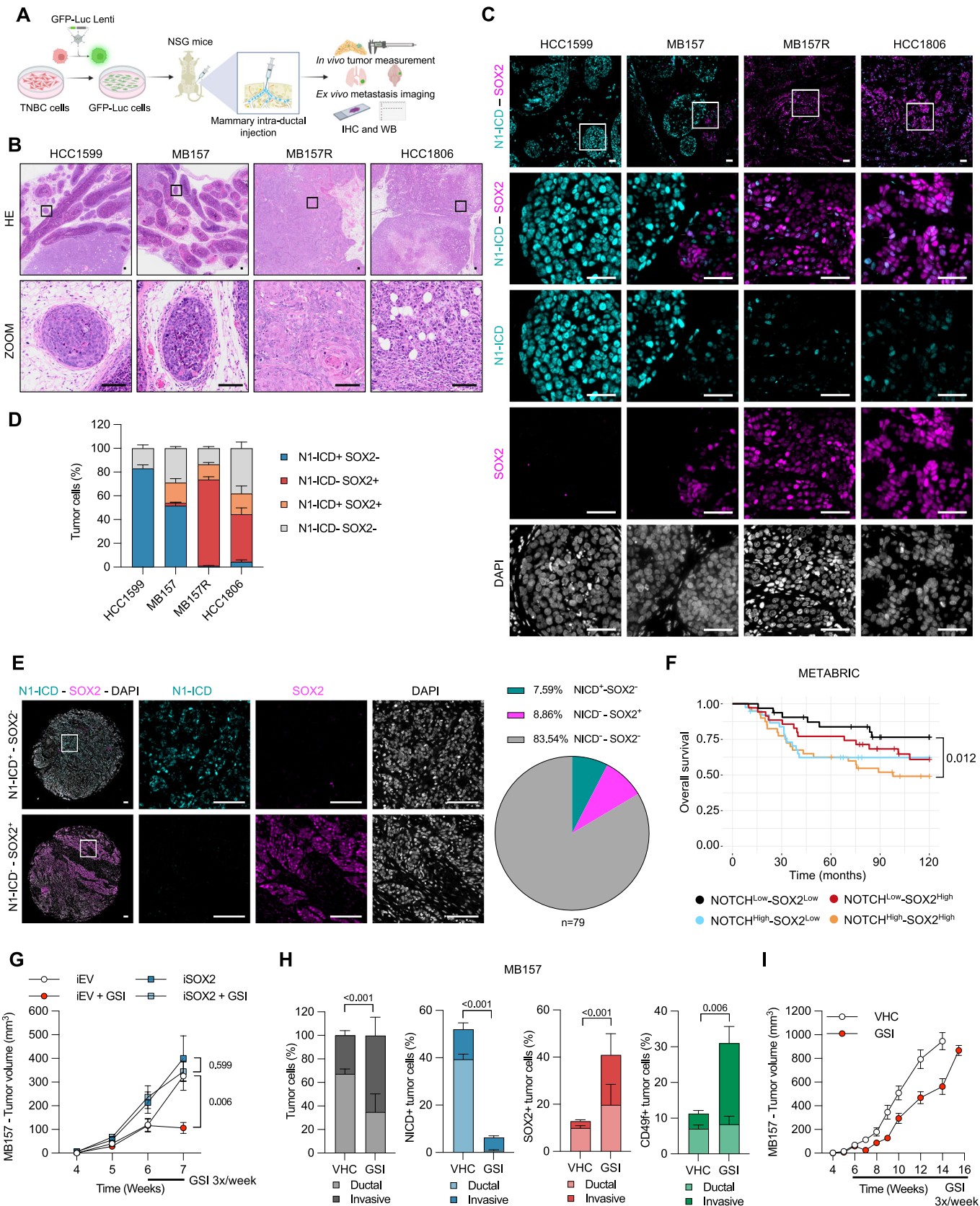

**Figure 5.  Escape of GSI-mediated in vivo tumor growth control due to TNBC tumor cell plasticity.**

(A) Scheme of MIND xenograft model setup in NSG mice, adapted from Sflomos et al (Sflomos et al, 2016). (B) Representative pictures of hematoxylin–eosin coloration for indicated MIND xenograft tumors at endpoint, scale = 100 µm. (C) Representative images of co-immunofluorescence N1-ICD–SOX2, scale = 50 µm, and (D) Quantification of N1-ICD and/or SOX2-positive tumor cells from co-immunofluorescence N1-ICD–SOX2 for indicated xenograft tumors at endpoint, n = 6. (E) Representative images of co-immunofluorescence N1-ICD– SOX2 in TMA of human TNBC samples, scale = 100 µm, with proportion of N1-ICD+ or SOX2+ TNBC samples, n = 79 (F) OS of TNBC patients from METABRIC dataset with NOTCH$^{High/Low}$ signature and SOX2$^{High/Low}$ expression: NOTCH$^{High}$/SOX2$^{High}$ (n = 40), NOTCH$^{High}$/SOX2$^{Low}$ (n = 38), NOTCH$^{Low}$/SOX2$^{High}$ (n = 35) and NOTCH$^{Low}$/SOX2$^{Low}$ (n = 32). (G) Tumor growth of iSOX2 or iEV MB157 xenografts treated with GSI (8 mg/kg) for 1 week, n = 10–11. (H) Quantification of N1-ICD, SOX2 and CD49f positive tumor cells in invasive or ductal areas from HE coloration and IHC staining in MB157 xenografts treated with GSI (8 mg/kg), n = 3. (I) Tumor growth of MB157 xenografts treated with GSI (8 mg/kg) for 10 weeks, n = 10–11. Data from biological replicates are represented as mean ± SEM. Log-rank test (F), two-way ANOVA (G), Cochran–Mantel–Haenszel test (H, left) or Student t test (H) were used to determine P values. Source data are available online for this figure.

lines were engrafted into the milk duct of immune-compromised mice (Fig. 5A). The mouse intraductal (MIND) model represents a more physiologically accurate approach compared to fat pad xenografts as TNBC cells engraft and grow in the milk ducts, providing the normal microenvironment (Kittrell et al, 2016; Sflomos et al, 2016). Tumor growth and dissemination of the xenograft tumors were measured. All four TNBC cell lines showed an engraftment rate >90%, with exponential tumor growth. The time to development of palpable tumors, however, varied between cell lines (Fig. EV4A). Histological analysis at experimental endpoints (~1000 mm³) and quantification of tumor cell growth showed that NOTCH-dependent and -independent tumors have divergent growth patterns. NOTCH-driven tumors, MB157 and HCC1599, grew initially intraductally and became invasive over time, with 53% and 70% of invasive growth at endpoint, respectively. In contrast, NOTCH-independent tumors (MB157R and HCC1806) grew exclusively extraductally, as non-encapsulated, invasive masses, fully replacing the preexisting parenchyma. HCC1599 and MB157 tumors were histopathologically classified as high-grade invasive ductal carcinomas, MB157R as high-grade adenosquamous carcinoma and HCC1806 as high-grade invasive lobular carcinoma (Figs. 5B and EV4B). At endpoint, xenografts of all four TNBC cell lines bore lung metastasis (Fig. EV4C).

To evaluate tumor heterogeneity, we performed immunohisto-chemistry staining for N1-ICD, SOX2, the SC marker CD49f and the EMT marker SLUG on tumor sections of the four TNBC xenograft models. Consistent with our in vitro findings, NOTCH-driven tumors showed high N1-ICD staining but weak staining for SOX2, CD49f and SLUG, whereas NOTCH-independent tumors showed lower levels of N1-ICD staining but higher levels of SOX2, CD49f and SLUG staining (Fig. EV4D). Altogether, NOTCH-independent tumors were more invasive and showed more stemness and EMT features compared to NOTCH-driven tumors. Interestingly, although active N1-ICD signaling was challenging to detect in MB157R and HCC1806 in vitro, certain cells within MB157R and HCC1806 tumor sections revealed detectable N1-ICD staining presumably induced by the ductal microenvironment (Fig. EV4D). To investigate expression of N1-ICD and SOX2 within the same tumor and/or cell, we performed immunofluor-escence co-staining for N1-ICD and SOX2 in all four TNBC xenograft models. Quantitative image analysis of tumor sections revealed expression of N1-ICD and SOX2 in different areas within the same tumor. Moreover, MB157, MB57R and HCC1806 showed tumor cells co-expressing N1-ICD and SOX2 (17%, 13%, and 17%, respectively) (Fig. 5C,D). These results are consistent with the heterogeneity observed in vitro for the MB157 cell line composed of

63.8% of N1-ICD positive cells, 7.1% of SOX2$^{Low}$ cells and 18.9% of N1-ICD-SOX2 co-expressing cells, while MB157R cells are largely negative for N1-ICD expression (1%) and mostly composed of SOX2-positive cells (58%) (Fig. EV4E). Interestingly, HCC1599 tumors expressed the highest number of N1-ICD-positive cells (83% of tumor cells positive for N1-ICD), while being completely negative for SOX2 staining (Fig. 5C,D). To validate the negative correlation between N1-ICD and SOX2 in human TNBC speci-mens, we performed immunofluorescence co-staining for N1-ICD and SOX2 and quantified N1-ICD and SOX2-positive samples in a TMA containing 79 patient-derived TNBCs. 16.5% of the TMA samples showed positive staining for either N1-ICD (7.6%) or SOX2 (8.9%). The staining was mutually exclusive, N1-ICD positive samples were negative for SOX2 and SOX2-positive samples were negative for N1-ICD staining (Fig. 5E), supporting the reciprocal negative regulation between NOTCH and SOX2 in human patients. We did not observe samples that co-stained for both N1-ICD and SOX2 within this limited number of TMAs. This prompted us to investigate larger public datasets of TNBC patients with regard to NOTCH$^{High}$ versus NOTCH$^{Low}$ signature, SOX2 expression and its association with patient survival. TNBC patients of the METABRIC database were classified into four different subgroups based on NOTCH signature and SOX2 expression levels. Among these subgroups, patients characterized by NOTCH$^{High}$ signature and SOX2$^{High}$ expression had the worst OS and RFS rates, whereas those with NOTCH$^{Low}$ signature and SOX2$^{Low}$ expression showed the most favorable prognosis (Figs. 5F and EV4F). This analysis shows that active Notch signaling and high levels of SOX2 can coexist within the same tumor, and associates with poor prognosis, similar to patients whose TNBC tumors express a Notch$^{High}$ signature only or SOX2 only.

Taken together, these results show an inverse correlation between N1-ICD and SOX2 expression, yet in three out of four TNBC models, these proteins can be co-expressed within the same tumor and the same cell. In agreement with the stratification of TNBC patients, HCC1599 cells can be seen as a model for NOTCH$^{High}$ signature/SOX2$^{Low}$ patients, while the MB157 TNBC cells could represent a model for the patient population expressing both NOTCH$^{High}$ signature and SOX2.

To assess the outcome of Notch inhibition in vivo for tumor growth, cell state and tumor cell heterogeneity, we performed MIND xenografts of iEV MB157 and iSOX2 MB157. Tumor-bearing mice were treated with GSI once tumors reached 100 mm³, three times a week (every other day) to avoid intestinal toxicity. Short-term GSI treatment (1 week) was sufficient to show a significant reduction in primary tumor volume of GSI-treated

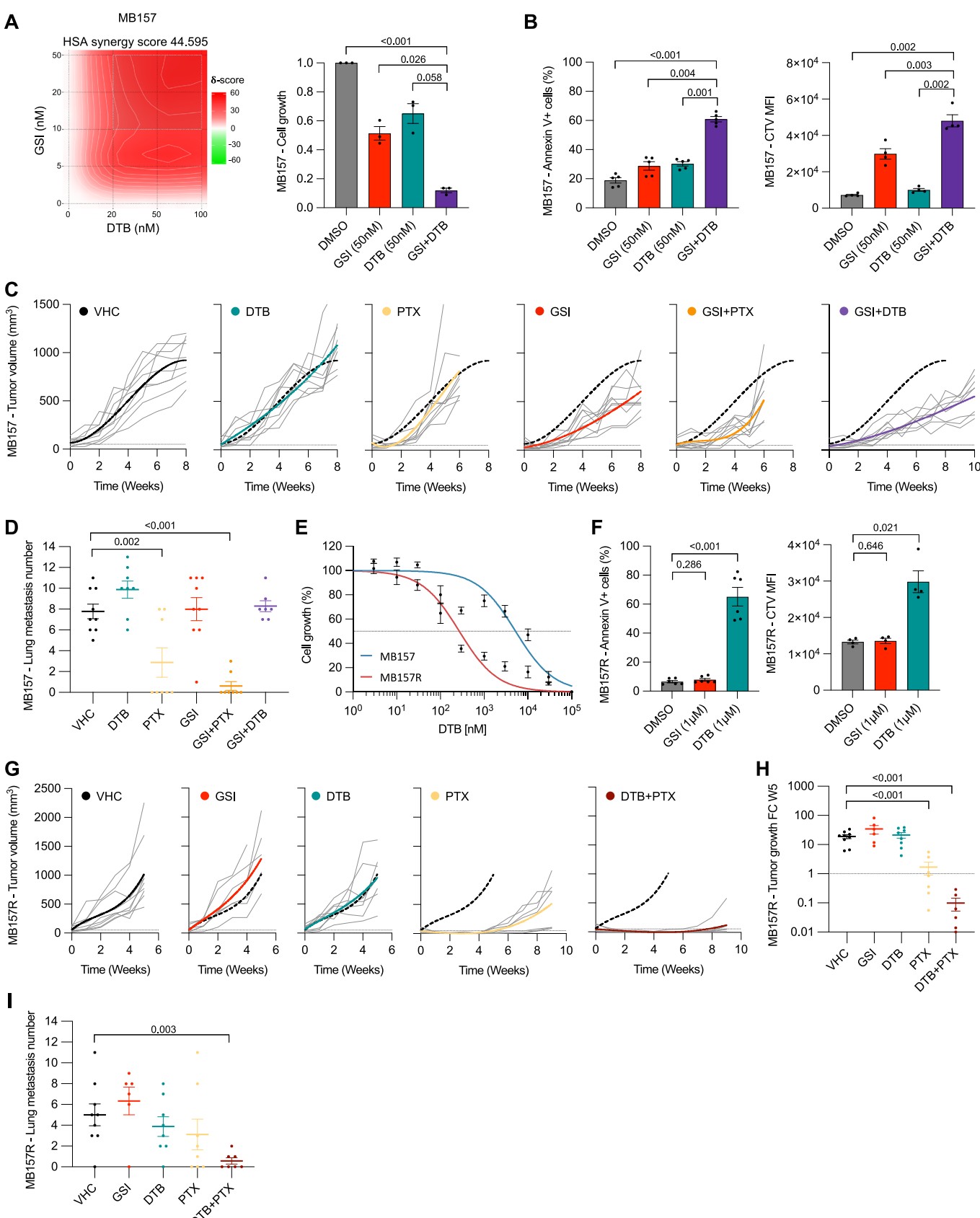

**Figure 6. Combination therapies to treat GSI-sensitive and -resistant TNBC xenografts.**

(**A**) HSA-synergy heatmap and cell growth inhibition of MB157 cells as indicated, after 6 days, $n = 3$. (**B**) The proportion of Annexin V+ cells ($n = 5$) and Cell Trace Violet MFI ($n = 4$) by flow cytometry analysis in MB157 cells as indicated, after 6 days. (**C**) Tumor growth and (**D**) Lung metastasis number in MB157 MIND xenografts treated with GSI (8 mg/kg, 3×/week), DTB (15 mg/kg, 5×/week), PTX (15 mg/kg, 1×/week) or VHC, $n = 7$–9. (**E**) Cell growth inhibition of MB157 and MB157R cells treated with DTB for 6 days, $n = 3$. (**F**) The proportion of Annexin V+ cells ($n = 6$) and Cell Trace Violet MFI ($n = 4$) by flow cytometry analysis in MB157R cells as indicated for 3 days. (**G**) Tumor growth, (**H**) Tumor growth fold change (5 weeks post treatment) and (**I**) Lung metastasis number in MB157R xenografts treated with GSI (8 mg/kg, 3×/week), DTB (15 mg/kg, 5×/week), PTX (15 mg/kg, 1×/week) or VHC, $n = 6$–9. Data from biological replicates are represented as mean ± SEM. One-way ANOVA (**A, B, D, F, H, I**) was used to determine $P$ values. Source data are available online for this figure.

MB157 xenografts compared to VHC-treated mice or GSI-treated animals bearing iSOX2 MB157 tumor cells (Fig. 5G). Thus, SOX2 expression in MB157 cells rendered tumor cells resistant to GSI in vivo, which correlated with an increased number of invasive tumor cells (Fig. EV4G). Lung metastasis between the different groups showed no significant differences (Fig. EV4H). Primary tumors were harvested and analyzed by immunohistochemistry to characterize changes of growth behavior and tumor cell state under drug treatment. GSI-treated MB157 xenografts were negative for N1-ICD staining indicating on-target activity of GSI (Figs. 5H and EV4I). Although tumor volume was reduced, we noticed changes within the proportions of invasive versus ductal tumor areas as well as increased numbers of SOX2 and CD49f positive tumor cells within the remaining tumor (Figs. 5H and EV4I). Thus, in vivo, GSI-mediated inhibition of Notch signaling seems to shift the balance of the tumor cells towards more invasive SOX2+ tumors, which are resistant to GSI. As expected, iSOX2 MB157 tumor cell xenografts showed decreased N1-ICD levels, but increased CD49f positive cells in particular in the invasive growing tumor mass and were resistant to GSI-mediated tumor growth inhibition (Figs. EV4I and Fig. 5G). Long-term treatment of MB157 xenografts showed that tumor growth control was transient, lasting 2–3 weeks before TNBC MB157 cells resumed growth despite continuous GSI treatment (Fig. 5I), recapitulating the in vitro findings (Fig. 1G). GSI treatment of HCC1599 xenografts likewise demonstrated transient and limited tumor growth control, while SOX2 expression rendered tumor cells insensitive to NOTCH inhibition, similar to iSOX2 MB157 xenografts. No significant differences were observed with regard to lung metastasis (Fig. EV4J–L). These results show that SOX2 is sufficient to induce GSI resistance in Notch-driven TNBC, associated with increased EMT/Stemness and invasiveness.

## Combination therapies to treat GSI-sensitive and -resistant TNBC xenografts

One of the major limitations of targeted therapies when used as monotherapy over longer time periods is that they cause tumors resistance. The relapsing tumor is no longer responsive to the original therapy and the therapeutic strategy requires adjustments. One way of trying to avoid adaptation of tumor cells to monotherapy is to use combination therapies and or develop second-line treatment options in case the tumor has become resistant to primary therapy. In light of this argument and the fact that tumor growth regression of GSI-treated TNBC xenografts was only limited and transient, we assessed combination therapies. Since no specific small molecule inhibitors for SOX2 are available, we performed an in vitro drug screen with both GSI-sensitive

(MB157 and HCC1599) and resistant (MB157R and HCC1806) TNBC models to discover drugs that may synergize with GSI-mediated Notch inhibition and/or standard of care. Initially, we performed a high throughput combination drug screen using 1536 FDA-approved compound libraries on NOTCH1-driven MB157 and HCC1599 TNBC cells, and identified 140 compounds efficient at 10 µM in NOTCH-driven TNBC cell lines (Dataset EV3 and 4). The screen was repeated with the 140 compounds at a concentration of 1 µM, identifying 41 compounds. The $IC_{50}$ of these 41 single agents were established and the top 13 compounds were tested for synergy in a concentration matrix with GSI. The Src family kinase inhibitor Dasatinib (DTB) scored best in the GSI matrix screen within MB157 cells with a high single-agent (HSA) synergy score of 44.6, at nM-scale concentrations for both compounds. Combination treatment of GSI plus DTB in vitro at 50 nM inhibited cell growth, through increased apoptosis and reduced cell proliferation, significantly better compared to single compound treatment at the same concentration (Fig. 6A,B). In HCC1599 cells, the HSA-synergy score for DTB and GSI was 6.8, which is indicative for additive but not synergistic action. HCC1599 cells were very sensitive to growth inhibition through induction of increased apoptosis and reduced proliferation by single compound treatment of GSI (10 nM). The combination of GSI plus DTB showed only a modest increase in in vitro growth inhibition (Fig. EV5A,B).

Next, we assessed the in vivo efficacy of DTB, paclitaxel (PTX, as standard of care), and GSI as monotherapy compared to combination treatment of GSI plus PTX and GSI plus DTB in MB157 and HCC1599 MIND xenografts (Figs. 6C and EV5C). Tumor-bearing NSG mice were treated at a tumor size of 100 mm³. Interestingly, both MB157 and HCC1599 xenografts were resistant to PTX treatment, and neither DTB nor GSI showed any substantial long-term tumor growth control as single agents. However, combination therapy of GSI plus PTX and GSI plus DTB, showed increased growth tumor control in both MB157 and HCC1599 xenografts compared to monotherapy. Although GSI plus DTB controlled primary tumor growth in MB157 cells and—to a somewhat lesser extent—HCC1599 xenografts, the number of lung metastasis was comparable to VHC in MB157 or even increased in the HCC1599 model (Figs. 6C,D and EV5C,D). Considering metastatic burden, the best combination therapy appears to be GSI plus PTX as it effectively controlled primary tumor growth and significantly reduced the number of lung metastasis at endpoint in both MB157 and HCC1599 xenografts suggesting that GSI sensitized tumor cells of both xenograft models to PTX.

We next aimed to identify small molecule inhibitors to assess second-line treatment for Notch inhibitor-resistant cells. A second screen for the GSI-resistant MB157R and HCC1806 TNBC cells was performed using the same commercially available libraries.

Surprisingly, DTB again scored best with an $IC_{50}$ of 271 nM in MB157R cells, which is a 20-fold increase in sensitivity compared to its $IC_{50}$ (5.2 μM) in MB157 cells (Fig. 6E; Dataset EV4). In contrast to GSI-sensitive MB157 and HCC1599 cells, both GSI-resistant MB157R and HCC1806 cell lines were highly sensitive to DTB, which inhibited the Src family kinase pathway in vitro, causing induction of apoptosis and reduction of cell proliferation, (Figs. 6F and EV5E–G). Although DTB was effective in vitro, it did not control tumor growth in MB157R and HCC1806 xenografts. As Paclitaxel is used as standard-of-care therapy in TNBC patients we next assessed DTB in combination with PTX. Interestingly, this combination afforded tumor regression in MB157R xenografts, while single-agent PTX only controlled tumor growth (Figs. 6G,H and EV5H,I). In addition, combination treatment reduced lung metastasis in MB157R at endpoint. Although the number of lung metastasis at endpoint were not reduced by combination of PTX plus DTB compared to VHC in HCC1806, it delayed the accumulation of metastatic lesions for more than 5 weeks, which corresponds to their different endpoint analysis (Figs. 6I and EV5J). Taken together, these results point to therapeutic strategies for NOTCH-driven and NOTCH-independent TNBC to avoid and overcome monotherapy-induced resistance.

## Discussion

Cell state and lineage plasticity in cancer is a major source of tumor heterogeneity and a major cause of treatment failure. Understanding tumor cell plasticity at the molecular level before and during response to therapy is important to foresee and counteract tumor cell resistance or relapse (Pérez-González et al, 2023; Yuan et al, 2019). Here, we identify and mechanistically describe a reciprocal inhibitory feedback loop between Notch signaling and the SC-associated TF SOX2, that regulates tumor cell plasticity and associated therapeutic response in NOTCH-driven TNBC. Moreover, our data provide a rational for possible combination therapies in NOTCH-driven TNBC and second-line treatment options for TNBC that demonstrate resistance to pharmacological Notch inhibitors.

Next-generation sequencing efforts have identified activating mutations, gene amplifications and chromosomal rearrangement in NOTCH receptor genes in all subpopulations of BC patients. Genetic NOTCH aberrations are most frequent in TNBC patients (Robinson et al, 2011; Stoeck et al, 2014; Wang et al, 2015). Although activating NOTCH gene mutations have been associated with a reduced OS in diverse cancers (Ferrarotto et al, 2017; Puente et al, 2011; Rossi et al, 2012; Kridel et al, 2012), our analysis of two independent public BC datasets (TCGA and METABRIC) did not confirm this correlation in TNBC. By defining a NOTCH-specific gene expression signature for TNBC, however, we found that NOTCH[High] signature TNBC patients have significantly reduced OS. This result is consistent with independent recent reports wherein NOTCH signatures were developed to predict response to neoadjuvant chemotherapy for TNBC patients (Omar et al, 2023; Braune et al, 2024).

The recent approval of GSI by the FDA for patients with desmoid tumors (Gounder et al, 2023) and the increasing number of clinical trials assessing GSI in various cancers, including TNBC, may eventually lead to its approval for targeted therapy of NOTCH-driven TNBC. Within this context, we investigated possible resistance mechanisms to GSI-mediated Notch inhibition in TNBC and identified SOX2 as a target. We identified a reciprocal negative feedback loop between SOX2 and N1-ICD which drove divergent cell states such as EMT and increased CSC features in TNBC. Mechanistically, SOX2 inhibits Notch signaling by binding to NOTCH1 and NOTCH target gene promoters and/or enhancers and through direct interaction with RBPJ, suggesting that it interferes with the formation of a functional N1-ICD/RBPJ transactivation complex. Meanwhile, Notch signaling inhibits SOX2 expression through the activation of downstream target genes of the HEY family of transcriptional repressors. Thus, Notch signaling and SOX2 seem to be implicated in a Yin and Yang relationship promoting distinct tumor cell states. Several studies associated the Notch pathway as well as SOX2 with features of cancer stem cells and resistance to chemotherapy in breast cancer (Qiu et al, 2013; Azzam et al, 2013; Yu et al, 2019; Das et al, 2019). The reciprocal negative regulation between Notch signaling and SOX2 described herein does not question their association with cancer stem cell features. But it highlights a regulatory circuit that allow cells within TNBC to adopt or favor divergent cell states to escape therapeutic treatment.

Interestingly, recent reports showed that a similar negative relationship between Notch signaling and SOX2 has been inferred for neuroendocrine transformation of lung and prostate cancer, in which SOX2 induces or maintains lineage plasticity towards the neuroendocrine cell state, while Notch signaling would prevent neuroendocrine tumor growth or transformation (Quintanal-Villalonga et al, 2021, 2023; Mu et al, 2017; Puca et al, 2019; Ku et al, 2024). This suggests that the antagonistic relationship between Notch signaling and SOX2 in regulating divergent cell states might apply to several tumor entities. Whether the molecular mechanism of reciprocal inhibition as described above for TNBC is also conserved in prostate, lung or other cancers needs to be addressed in future investigations.

Although frequently discussed in the context of acquired therapeutic resistance, tumor cell plasticity also occurs independent of drug selection, for example, through spatial localization of the tumor cells and the tumor microenvironment (Pérez-González et al, 2023). We assessed tumor cell heterogeneity with respect to N1-ICD and SOX2 protein expression in 4 different xenograft models, two GSI-sensitive (MB157 and HCC1599) and two NOTCH-independent (MB157R and HCC1806). Interestingly, all TNBC xenograft models revealed intratumoral heterogeneity for N1-ICD and/or SOX2 without treatment. With the exception of HCC1599 xenografts, which expressed N1-ICD but were SOX2 negative, tumors of all other models contained tumor cells expressing both N1-ICD and SOX2. Within these tumors, cells with high N1-ICD expression were negative or low for SOX2, while SOX2-positive tumor cells were low for N1-ICD, which is in agreement with the model of reciprocal inhibition we identified. Knowledge about tumor cell heterogeneity before treatment is crucial to improve therapeutic outcome (Bianchini et al, 2022). Treating TNBC patients with heterogenous tumors expressing both N1-ICD and SOX2 with GSI as monotherapy is predicted to elicit a transient therapeutic response coupled with a shift in tumor cell state and ultimately tumor relapse. Blocking Notch signaling would result in derepressed SOX2 expression, driving a more invasive

and GSI-resistant form of tumor growth, as shown in the MB157 model.

The transient GSI-mediated tumor growth control of HCC1599 xenografts cannot be explained by derepressed SOX2 expression as they are largely SOX2 negative. Other cell line xenograft studies have shown better response to GSI treatment than that shown here (Wang et al, 2015; Robinson et al, 2011; Stoeck et al, 2014). This could be for two reasons. First, our treatment strategy of intermittent Notch inhibition (to avoid GSI-mediated gut toxicity (van Es et al, 2005)) may have allowed for reduced tumor growth control. Second, we grafted our cell lines into the more physiologically accurate milk ducts (Sflomos et al, 2016) as compared to the fat pat used in the aforementioned studies. Further experiments would be required to determine the source of this divergent finding.

Our in vitro drug screens with both GSI-sensitive (MB157 and HCC1599) and resistant (MB157R and HCC1806) TNBC models designed to discover FDA-approved drugs that may synergize with GSI-mediated NOTCH inhibition led to the implication of the Src family kinase inhibitor DTB. The combination treatment of GSI plus DTB was well-tolerated and resulted in a better growth tumor control of primary NOTCH-driven tumors compared to single-agent treatment. In fact, DTB as a single agent did not show any tumor growth control. These results agree with two phase II clinical trials that assessed DTB as a single agent in patients with metastatic BC or TNBC. In both trials, DTB showed only limited or no significant anti-tumor activity (Herold et al, 2011; Finn et al, 2011). Although GSI plus DTB showed some tumor growth control, it did not reduce or prevent lung metastasis. In contrast, GSI plus PTX, which was less well-tolerated, reduced lung metastasis dramatically. Primary tumor growth of the NOTCH-driven xenografts was largely unaffected by single-agent PTX treatment, suggesting that GSI sensitized our NOTCH-driven xenograft models to PTX. Similar synergistic effects between GSI and PTX have been reported in other preclinical studies of ovarian and non-small lung cancer (Groeneweg et al, 2014; Kang et al, 2016; He et al, 2017), suggesting that this might not be restricted to only one tumor type. How GSI might mechanistically sensitize multiple cancer types to PTX is currently unclear. Taken together the above-mentioned studies and our findings support the rational to test GSI plus PTX in NOTCH-driven TNBC.

Interestingly, DTB was also the prime hit in our drug screen with GSI-resistant TNBC cells. In vitro DTB inhibited cell growth potently, but was ineffective as a single agent in controlling tumor growth. Nevertheless, the combination of DTB plus PTX induced tumor control in both of our GSI-resistant xenograft models indicating that this combination should be considered for NOTCH-independent or GSI-resistant TNBC. Our results are consistent with a previous preclinical report showing that DTB sensitizes TNBC cells to chemotherapy by targeting BC SCs (Tian et al, 2018). Notably, this combination has been assessed in a single-arm phase II clinical trial in patients with HER2- metastatic BCs (Morris et al, 2018). The majority of recruited patients were estrogen receptor-positive. Despite early termination due to slow accrual, the objective response rate was 23%. A single patient, however, had a complete response, and their BC was TNBC. We could not find any reports of clinical phase II trials assessing the combination of DTB and PTX focusing on patients with TNBC. Our data, however, would support such a rationale.

Taken together, our study identified a novel mechanism of reciprocal inhibition between Notch signaling and SOX2 that shapes tumor cell plasticity and therapeutic escape in NOTCH-driven TNBC. Moreover, our data provide rationale for therapeutic options of combination therapies for GSI-sensitive and -resistant TNBC.

# Methods

## Methods and protocols

### Plasmid generation

Full-length *SOX2* cDNA (pENTR221.SOX2) was obtained from ORFeome Collaboration Collection (GECF, EPFL). DeltaE-*NOTCH1* (Human NOTCH1 lacking extracellular domain, namely N1-ICD) cDNA was a gift from Prof. Mark Chiang, University of Michigan.

Doxycycline (DOX) inducible constructs were generated by cloning cDNAs of *SOX2* or *N1-ICD* into lentiviral (LV) vector pSIN_TRE-GW_3xHA-IRES-PURO (gift from Didier Trono) using Gateway technology (Thermo Fisher Scientific, Cat#11791020).

Luciferase reporter pGL4.26_12xCSL_Luc was previously generated in the laboratory of Freddy Radtke (Lehal et al, 2020). SV40-Renilla vector was obtained from Promega (Cat# E6911).

Plasmid for in vivo imaging mPGK-Luciferase-hPGK-eGFP was a gift from Richard Iggo.

The LentiCas9-Bast plasmid (Addgene, Cat#52962) used to generate Cas9-expressing cells, was deposited on Addgene by Feng Zhang (Sanjana et al, 2014).

## Cell lines and tissue culture conditions

MB157 (CRL-7721), HCC1599 (CRL-2331), HCC1806 (CRL-2335), BT-549 (HTB-122), and HEK293T (CRL-3216) cells were purchased from the American Type Culture Collection and were authenticated by STR profiling and tested negative for mycoplasma. MB157 and HEK293T were cultured in DMEM (Gibco, Cat# 11965092), HCC1806, BT-549, and HCC1599 in RPMI 1640 (Gibco, Cat# 11875093) supplemented with 10% fetal calf serum (FCS) at 37 °C in a 5% $CO_2$ humidified atmosphere. For siRNA transfections, cells were transfected with Lipofectamine RNAiMAX transfection reagent (Invitrogen, Cat# 13778150) according to the manufacturer's instructions. For plasmid transfections, cell lines were transfected using Fugene HD transfection reagent (Promega, Cat# E2311) according to the manufacturer's protocol.

Drug-resistant cells were generated by treating the MB157 cells for 25 weeks with 1 μM γ-secretase inhibitor (GSI, Spirochem, LY3039478), replenishing the inhibitor every 3–4 days. Cells were maintained at a minimal density of $2.5–3 \times 10^6$ cells/10-cm dish. Cell count was performed by trypan blue exclusion at every cell passage.

Tumor cells were transduced with DOX-inducible empty vector (iEV), SOX2 (iSOX2) or DeltaE-NOTCH1 (iN1-ICD) constructs. Briefly, LV particles were produced using calcium-phosphate transfection of the construct of interest and packaging plasmids (gift from Richard Iggo). Tumor cells were infected with LV

**Reagents and tools table**

| Reagent/resource | Source | Identifier |
|---|---|---|
| **Experimental models** | | |
| NSG mice | The Jackson Laboratory | Cat# 005557 |
| MB157 cells | ATCC | Cat# CRL-7721 |
| HCC1599 cells | ATCC | Cat#CRL-2331 |
| HCC1806 cells | ATCC | Cat#CRL-2335 |
| HEK293T cells | ATCC | Cat#CRL-3216 |
| BT-549 cells | ATCC | Cat#HTB-122 |
| Human TNBC TMA BR931 | TissueArray | Cat#BR931 |
| **Recombinant DNA** | | |
| Human GeCKOv2 library A | Sanjana et al, 2014 | Addgene Plasmid #1000000048 |
| Human GeCKOv2 library B | Sanjana et al, 2014 | Addgene Plasmid #1000000049 |
| LentiCas9-Blast | Sanjana et al, 2014 | Addgene Plasmid #52962; RRID:Addgene_52962 |
| pSIN_TRE_GW_3xHA_IRES_PURO | Gift from Didier Trono | N/A |
| pSIN_TRE_SOX2_3xHA_IRES_PURO | | N/A |
| pSIN_TRE_DeltaE-NOTCH1_3xHA_IRES_PURO | | N/A |
| pSD16 | Gift from Richard Iggo | N/A |
| pSD11 | Gift from Richard Iggo | N/A |
| SV40-Renilla | Promega | Cat#E6911 |
| pGL4.26_2xHH_Luc | Lehal et al, 2020 | N/A |
| mPGK-Luciferase-hPGK-eGFP | Gift from Richard Iggo | N/A |
| **Antibodies** | | |
| Rabbit monoclonal anti-Cleaved Notch1 (Val1744) (clone D3B8) _ WB:1/1000 – IHC/IF: 1/50 | Cell Signaling Technology | Cat#4147; RRID:AB_2153348 |
| Rabbit monoclonal anti-RBPSUH XP (clone D10A4) _WB: 1/1000_ChIP: 1/50 | Cell Signaling Technology | Cat#5313; RRID:AB_2665555 |
| Rabbit polyclonal anti-RBPJ _ IP: 1/50 | Proteintech | Cat#30044-1-AP; RRID:AB_3086217 |
| Rabbit monoclonal anti-SOX2 (clone D9B8N) _ WB: 1/1000_ChIP: 1/50_IP: 1/100 | Cell Signaling Technology | Cat#23064; RRID:AB_2714146 |
| Goat polyclonal anti-SOX2 _ IHC/IF: 1/100 | R&D Systems | Cat#AF2018; RRID:AB_355110 |
| Rabbit polyclonal anti-SOX2 _ WB: 1/2000 | Merck Millipore | Cat#AB5603; RRID:AB_2286686 |
| Rabbit monoclonal anti-Claudin-3 (clone D7A3O) _ WB: 1/1000 | Cell Signaling Technology | Cat#83609; RRID:AB_2800021 |
| Rabbit monoclonal anti-SLUG (clone C19G7) WB: 1/1000_IHC: 1/50 | Cell Signaling Technology | Cat#9585; RRID:AB_2239535 |
| Rabbit monoclonal anti-HA (clone C29F4)) WB: 1/1000_ChIP/IP: 1/50 | Cell Signaling Technology | Cat#3724; RRID:AB_10693385 |
| Rabbit monoclonal anti-E-Cadherin (clone 24E10) _ WB: 1/1000 | Cell Signaling Technology | Cat#3195; RRID:AB_2291471 |
| Rabbit monoclonal anti-Integrin α6 _ WB: 1/1000_IHC: 1/50 | Cell Signaling Technology | Cat#3750; RRID:AB_2249263 |
| Mouse monoclonal anti-N-Cadherin (clone 13A9) _ WB: 1/1000 | Santa Cruz | Cat#sc-59987; RRID:AB_781744 |
| Mouse monoclonal anti-HA.11 (Clone 16B12)_WB: 1/1000 | Biolegend | Cat#901516 |
| Mouse monoclonal anti-TWIST (clone Twist2C1a) _WB: 1/500 | Santa Cruz | Cat#sc-81417; RRID:AB_1130910 |

| Reagent/resource | Source | Identifier |
|---|---|---|
| Rabbit monoclonal anti-Vimentin XP (Clone D21H3) _ WB: 1/1000 | Cell Signaling Technology | Cat#574; RRID:AB_2798037 |
| Rabbit polyclonal anti-Phospho-Src Family (Tyr416) _ WB: 1/1000 | Cell Signaling Technology | Cat#2101; RRID: AB_331697 |
| Rabbit monoclonal anti-Phospho-AKT (Ser473) (DE9) _ WB: 1/1000 | Cell Signaling Technology | Cat#4060; RRID: AB_2315049 |
| Mouse monoclonal gamma-Tubulin _ WB: 1/2000 | Sigma-Aldrich | Cat#T6557; RRID:AB_477584 |
| Mouse monoclonal anti-GAPDH (Clone 6C5) _ WB: 1/5000 | Abcam | Cat#ab8245; RRID:AB_2107448 |
| Rabbit monoclonal cMYC (Clone Y69) _ WB: 1/1000 | Abcam | Cat#ab32072; RRID:AB_731658 |
| Mouse monoclonal anti-beta Actin _ WB: 1/2000 | Abcam | Cat#ab8226; RRID:AB_306371 |
| Mouse monoclonal anti-TATA binding protein _ WB: 1/1000 | Abcam | Cat#ab51841; RRID:AB_945758 |
| Rabbit polyclonal anti-HA _ WB: 1/1000_ChIP: 1:150 | Abcam | Cat#ab9110; RRID:AB_307019 |
| Goat anti-rabbit IgG, HRP-linked _ IHC: 1/200 | Cell Signaling Technology | Cat#7074; RRID:AB_2099233 |
| Horse anti-mouse IgG, HRP-linked _ IHC: 1/200 | Cell Signaling Technology | Cat#7076; RRID:AB_330924 |
| Mouse anti-rabbit IgG (Conformation Specific, L27A9), HRP-linked _ IHC: 1/200 | Cell Signaling Technology | Cat#5127; RRID:AB_10892860 |
| ImmPRESS HRP Horse Anti-Rabbit IgG, Polymer Detection Kit | Vector Laboratories | Cat#MP-7401; RRID:AB_2336529 |
| ImmPRESS HRP Horse Anti-Goat IgG, Polymer Detection Kit | Vector Laboratories | Cat#MP-7405; RRID:AB_2336526 |
| ImmPRESS HRP Horse Anti-Mouse IgG, Polymer Detection Kit | Vector Laboratories | Cat#MP-7402; RRID:AB_2336528 |
| Donkey anti-Goat IgG, Alexa Fluor 568 _ IF: 1/500 | Thermo Fisher Scientific | Cat#A-11057; RRID:AB_2534104 |
| Donkey anti-Rabbit IgG, Alexa Fluor 647 _ IF: 1/500 | Thermo Fisher Scientific | Cat#A-31573; RRID:AB_2536183 |
| **Oligonucleotides** | | |
| Primers are listed in Table EV2 | | |
| **Chemicals, peptides, and recombinant proteins** | | |
| RPMI 1640 | Gibco | Cat#11875093 |
| DMEM | Gibco | Cat#11965092 |
| Puromycin | InvivoGen | Cat#ant-pr-1 |
| Blasticidin | InvivoGen | Cat#ant-bl-1 |
| Doxycycline Hyclate | Merck | Cat#D9891 |
| Polybrene | Santa Cruz | Cat#sc-134220 |
| Advanced DMEM/F-12 medium | Gibco | Cat#11540446 |
| Recombinant Human EGF | PreproTech | Cat#AF-100-15 |
| Recombinant Human FGF-basic | PreproTech | Cat#100-18B |
| B27 | Gibco | Cat# 17504044 |
| GlutaMAX Supplement | Gibco | Cat#35050061 |
| D-Luciferin Firefly | Biosynth | Cat#L-8220 |
| Paclitaxel (CAS no 33069-62-4) | Selleckchem | Cat#S1150 |
| Dasatinib (CAS no 302962-49-8) | Apollo Scientific | Cat#OR302638 |
| Crenigacestat (CAS no 1421438-81-4) | SpiroChem | LY3039478 |
| DMSO | AppliChem | Cat#A3672.0100 |
| PrimeScript RT Master Mix (Perfect real Time) | Takara | Cat#RR036A |
| RNeasy Mini Kit | Qiagen | Cat#74104 |
| TRIzol Reagent | Invitrogen | Cat#15596026 |

| Reagent/resource | Source | Identifier |
|---|---|---|
| PRImezol Reagent | Canvax | Cat#AN1100 |
| Power SYBR Green PCR Master Mix | Applied Biosystems | Cat#43-676-59 |
| WesternBright ECL Spray | Advansta | Cat#K-12049-D50 |
| nProtein A Sepharose 4 Fast Flow | GE Healthcare | Cat#17-5280-04 |
| PrestoBlue Cell Viability Reagent | Invitrogen | Cat#A13261 |
| Herculase II Fusion DNA polymerase | Agilent Technologies | Cat#600679 |
| Lipofectamine | Invitrogen | Cat#13778-075 |
| FuGENE HD Transfection Reagent | Promega | Cat#E2311 |
| Lipofectamine RNAiMAX Transfection Reagent | Qiagen | Cat#13778150 |
| FlexiTube GeneSolution GS6657 for SOX2 | Qiagen | Cat#1027416 |
| FlexiTube GeneSolution GS26508 for HEYL | Qiagen | Cat#1027416 |
| FlexiTube GeneSolution GS388585 for HES5 | Qiagen | Cat#1027416 |
| FlexiTube GeneSolution GS23493 for HEY2 | Qiagen | Cat#1027416 |
| Fluoromount-G | SouthernBiotech | Cat#0100-01 |
| DAB | Sigma-Aldrich | Cat#D5905 |
| Disuccinimidyl glutarate (CAS 79642-50-5) | Santa Cruz | Cat#sc-285455 |
| Formaldehyde | Pierce Life Technologies | Cat#28906 |
| Protease inhibitor cocktail | Sigma | Cat#P8340 |
| PhosSTOP EASYpack | Roche | Cat#PHOSS-RO |
| Protease Inhibitor Mini Tablets | Pierce | Cat#A32953 |
| Xylasol | Graeub | |
| Ketasol 100 | Graeub | |
| **Commercial assays** | | |
| FITC Annexin V Apoptosis Detection kit with 7-AAD | Biolegend | Cat#640922 |
| Cell Trace Violet Proliferation Kits | Invitrogen | Cat#C34571 |
| BD Pharmingen BrDU Flow kits | BD Biosciences | Cat#2617060 |
| ALDEFLUOR Kit | Stem Cell Technologies | Cat#01700 |
| iDeal ChIP-seq kit for Transcription Factors Kit | Diagenode | Cat#C01010170 |
| Dual-Luciferase Reporter Assay | Promega | Cat#E1910 |
| ZymoResearch Quick-gDNA MidiPrep Plus Kit | ZymoResearch | Cat#ZYM-D4075-25TST |
| Qiagen gel purification kit | Qiagen | Cat#28706 |
| Qiagen PCR purification kit | Qiagen | Cat#28106 |
| Gateway LR Clonase II Enzyme mix | Thermo Fisher Scientific | Cat#11791020 |
| NEBNext Ultra II DNA Library Prep Kit for Illumina | New England Biolabs | Cat#7103 |
| Illumina Stranded mRNA Prep, Ligation | Illumina | Cat#20040534 |
| **Software and algorithms** | | |
| QuPath | Bankhead et al, 2017 | https://qupath.github.io/ |
| FIMO (MEME suite v.5.5.5) | Bailey et al, 2015 | https://meme-suite.org/meme/ |
| bclconvert (v.3.9.3) | Illumina | https://emea.support.illumina.com/sequencing/sequencing_software/bcl-convert.html |
| STAR and Salmon (nf-core/rnaseq version 3.9) | https://nf-co.re | STAR RRID:SCR_004463; Salmon RRID:SCR_017036 |
| EdgeR (v.3.42.4) | http://bioconductor.org/packages/edgeR/ | RRID:SCR_012802 |
| voom from limma (v.3.56.2) | http://bioinf.wehi.edu.au/limma/ | RRID:SCR_010943 |

| Reagent/resource | Source | Identifier |
|---|---|---|
| clusterProfiler (v.4.8.3) | http://bioconductor.org/packages/release/bioc/html/clusterProfiler.html | RRID:SCR_016884 |
| msigdbr v.7.5.1 | https://cran.r-project.org/package=msigdbr; | RRID:SCR_022870 |
| IGV (v. 2.16.2) | http://www.broadinstitute.org/igv/ | RRID:SCR_011793 |
| MAGeCK (v.0.5.9.2) | https://sourceforge.net/p/mageck/wiki/Home/ | RRID:SCR_025016 |
| bcl2fastq (v2.19) | https://support.illumina.com/sequencing/sequencing_software/bcl2fastq-conversion-software.html | RRID:SCR_015058 |
| fastQC (v0.11.5) | http://www.bioinformatics.babraham.ac.uk/projects/fastqc/ | RRID:SCR_014583 |
| MAGeCKFlute (v2.2.0) | https://bioconductor.org/packages/MAGeCKFlute | DOI: 10.18129/B9.bioc.MAGeCKFlute |
| BWA and MACS2 (nf-core/chip-seq version 2.0.0) | https://nf-co.re | BWA RRID:SCR_010910; MACS2 RRID:SCR_013291 |
| survminer (v0.4.9) | https://rdocumentation.org/packages/survminer/versions/0.4.9 | RRID:SCR_021094 |
| survival (v3.2-7) | https://CRAN.R-project.org/package=survival | RRID:SCR_021137 |
| FlowJo v10 software (TreeStar) | https://www.flowjo.com/ | RRID:SCR_008520 |
| JASPAR | http://jaspar.genereg.net | RRID:SCR_003030 |
| GraphPad Prism (v.10.2.1) | http://www.graphpad.com/ | RRID:SCR_002798 |
| **Deposited data** | | |
| RNA-Seq and ChIP-Seq | This paper | GEO: GSE262009 |
| Raw and analyzed data related to RNA-Seq | This paper | GEO: GSE262001 |
| Raw and analyzed data related to ChIP-Seq | This paper | GEO: GSE262007 |

particles performing spininfection at 800×*g* for 2 h at 37 °C using 1 µg/mL polybrene (Santa Cruz, Cat#sc-134220). Expression of SOX2 or N1-ICD was induced using 0.5–1 µg/mL DOX (Merck, Cat#D9891) for 72 h. Washout experiments were performed by inducing SOX2 or N1-ICD for 48 h and then removing DOX for 2 to 5 days.

To generate GFP-luciferase-expressing cells, tumor cells were transduced with mPGK-Luciferase-hPGK-eGFP plasmid by spininfection.

Cells were incubated with GSI or Dasatinib (DTB, Apollo Scientific, Cat#OR302638) at indicated concentrations and times. Dimethyl sulfoxide (DMSO, AppliChem, Cat#A3672.0100) was used as a control.

## Immunoblot and immunoprecipitation

Proteins were extracted in RIPA lysis buffer (50 mM Tris-HCl pH 7.5, 150 mM NaCl, 1% Triton X-100, 0.5% sodium deoxycholate, 0.1% SDS) supplemented with protease (Pierce, Cat# A32953) and phosphatase (Roche, Cat#PHOSS-RO) inhibitors and short cycles of sonication. Protein lysates were immunoblotted with the indicated antibodies (Reagents and Tools Table) and revealed with ECL (Advansta, Cat#K-12049-D50). GAPDH (Abcam, Cat#8245, (1/5000)), gamma-Tubulin (Sigma-Aldrich, Cat#T6557, (1/2000)), TBP (Abcam, Cat#51841 (1/1000)) or Actin (Abcam, Cat#ab8226, (1/2000)) were used as loading controls. For immunoprecipitation, proteins were extracted in lysis buffer (20 mM Tris pH 8.0, 150 mM NaCl, 1% NP-40, 10% glycerol) supplemented with protease inhibitor. Per immunoprecipitation, 2 mg of proteins were

incubated overnight at 4 °C in 500 µl lysis buffer with indicated antibodies and 50 µl Protein A Sepharose beads (GE Healthcare, Cat#17-5280-04). Beads were washed 5× in lysis buffer and proteins were eluted with SDS sample buffer (50 mM Tris-HCl pH 6.8, 10% glycerol, 2% SDS, 0.1% bromophenol, 100 mM Dithiothreitol) before immunoblotting.

## RNA extraction and quantitative real-time PCR

Total RNA was extracted with the RNeasy Mini Kit (Qiagen, Cat#74104) or Trizol/chloroform method (Invitrogen, Cat#15596026), then reverse transcribed into cDNA with 5× PrimeScript RT Master Mix Kit (Takara, Cat#RR036A) using manufacturer's instructions. Quantitative real-time PCR was performed using SYBR green (Applied Biosystems, Cat#43-676-59) on QuantStudio 7 machine (Applied Biosystems). The threshold cycle (Ct) value for each gene was normalized to the Ct value of the respective reference gene (*HPRT*, *GAPDH* or *TBP*) and the relative levels of expression were calculated. Primers are listed in Table EV2.

## RNAseq

Illumina stranded mRNA ligation (ISML) prep (Illumina, Cat# 20040534) starting from 800 ng of RNA was performed according to Illumina protocol 1000000124518 v01. The libraries were quantified by qubit DNA HS (Thermo Fisher Scientific, Cat# Q32851) and their profile analyzed by TapeStation TS4200 (Agilent). The libraries were sequenced on Illumina NovaSeq 6000 (Illumina) in a PE60. The reads for ISML (Netxera) adapters

were trimmed using bclconvert (v.3.9.3). Mapping and quantification were performed with STAR and Salmon (nf-core/rnaseq version 3.9) on the human genome hg38 with reverse strand-specificity setting and default parameters. Raw counts were normalized in 3 batches of 6 samples each (batch1 MB157R-iEV +MB157R-iN1-ICD; batch2 MB157-iEV+MB157-iSOX2; batch3 MB157R-iEV+MB157-iEV) using TMM method from EdgeR (v.3.42.4) and voom from limma (v.3.56.2). Genes were filtered out if average TPM < 2 or average counts <5 per sample. Uncharacterized, predicted and pseudogenes were also filtered out to keep 3 matrices of counts (batch1 $n = 11{,}210$; batch2 $n = 10{,}904$; batch3 $n = 11{,}435$ genes). In each batch, differential expression was computed with limma. GSEA was performed with clusterProfiler (v.4.8.3) applying GSEA with default parameters, 100,000 permutations to obtain $P$ values. The Hallmark collection and BC-specific gene sets found in Chemical and Genetic Perturbations Curated collection (CGP, C2) from msigdbr (v.7.5.1) were used, namely EMT (Hollern et al, 2018), Stem cell (Lim et al, 2010), Luminal, Basal, Mesenchymal (Stoeck et al, 2014; Petrovic et al, 2019; Charafe-Jauffret et al, 2006), as wells as a custom curated NOTCH signature was also used.

### Sashimi plot

Sashimi plot was generated with IGV (v. 2.16.2) using the reads from EV control MB157.

## Cell growth assay

For 2D proliferation assays, 2000–10,000 cells were seeded onto 96-well plates and incubated at indicated treatment and concentrations for up to 6 days. At day 0, 3 and 6, cell growth was measured using PrestoBlue reagent (Invitrogen, Cat#A13261) as described the manufacturer's instructions.

## Flow cytometry analysis

For cell cycle analysis, cells were labeled with 10 µM BrdU for 2 h at 37 °C. After labeling, cells were fixed with and stained with anti-BrdU-FITC and 7-AAD following the manufacturer's instructions (BD Pharmingen BrdU flow kit, Cat# 2617060).

ALDEFLUOR Assay Kit (Stemcell Technologies, Cat# 01700) was used to detect ALDH+ cells according to the manufacturer's instructions. Briefly, cells were trypsinized, washed with 1× PBS, and resuspended in ALDEFLUOR Assay Buffer containing ALDH substrate. A fraction of cells was incubated with the ALDH inhibitor diethylaminobenzaldehyde as a negative control. Subsequently, cells were incubated at 37 °C for 45 min to allow ALDH substrate conversion. Following incubation, cells were washed and resuspended in ALDEFLUOR Assay Buffer for flow cytometric analysis.

Cell Trace Violet Cell Proliferation Kit (Invitrogen, Cat# C34571) was used to label cells according to the manufacturer's instructions. Briefly, cells were harvested and resuspended in 1× PBS at a concentration of $1 \times 10^6$ cells/mL. Cell Trace Violet dye was added to the cell suspension at a final concentration of 5 µM and incubated for 20 min at 37 °C. Staining was quenched by adding five volumes of complete growth medium containing 10% FCS. Labeled cells were then washed with 1× PBS and resuspended in fresh growth medium for subsequent plating. Cell Trace Violet

Mean Fluorescence Intensity (CTV MFI) was measured to determine the proliferation index. A low CTV MFI indicates a higher proliferation rate, and vice versa.

Apoptosis was assessed using the Annexin V-FITC Apoptosis Detection Kit (Biolegend, Cat#640922) according to the manufacturer's instructions. Briefly, cells were trypsinized, harvested, washed with cold 1× PBS, and resuspended in 1× Annexin-V binding buffer. Cells were then incubated with Annexin V-FITC and 7-AAD for 15 min at room temperature (RT) in the dark.

Stained cells were acquired on a Gallios (Beckman Coulter) or BD Fortessa LSR II flow cytometer, and data were analyzed with FlowJo v10 software (TreeStar).

## Tumorsphere assay

Cells were plated at very low confluence (5000 cells/well) in ultra-low adherent six-well plates (Corning, Cat#3471) in Advanced DMEM/F-12 medium (Gibco, Cat#11540446) without FCS by adding B27 (Gibco, Cat# 17504044), 200 nM GlutaMAX Supplement (Gibco, Cat#35050061), 20 ng/mL EGF (PreproTech, Cat#AF-100-15) and 10 ng/mL FGF2 (PreproTech, Cat#100-18B) for 2 weeks. Tumorspheres with a diameter greater than 50 µm were counted and measured using the Olympus Cell xCellence Microscope (Leica).

## CRISPR screen

CRISPR-Cas9 screens were performed using Human GeCKOv2 library (Deposited on Addgene by Feng Zhang (Sanjana et al, 2014), Cat# 1000000048 and # 1000000049). A minimum of $6 \times 10^7$ Cas9-expressing MB157R cells (cultured with 8 µg/mL blasticidin) were transduced at an MOI of 0.3–0.4 with the sgRNA library for minimum 200-fold coverage, in the presence of 1 µg/mL polybrene (Santa Cruz, Cat#sc-134220) and seeded onto 10-cm dishes. Forty-eight hours post-transduction, medium was changed and replaced with fresh DMEM media containing 8 µg/mL blasticidin (resistance for Cas9 plasmid) and 1 µg/mL puromycin (resistance for sgRNA plasmid). After 4 days of puromycin selection, cells were combined and two T0 samples ($6 \times 10^7$ cells per sample) were collected and stored at $-80$ °C for later processing. The remaining cells were divided into triplicates and cultured with DMSO (AppliChem, Cat#A3672.0100) or GSI (Spirochem, LY3039478) at 10 µM (Spirochem, LY3039478) for an additional 14 days. At this point, cells were collected in $6 \times 10^7$ cell aliquots (200-fold coverage) for gDNA extraction. Genomic DNA was extracted using the using ZymoResearch Quick-gDNA MidiPrep Plus Kit (ZymoResearch, Cat#ZYM-D4075-25TST) according to the manufacturer's protocol. sgRNA sequences were amplified using Herculase II Fusion DNA polymerase (Agilent Technologies, Cat#600679) in two rounds: first round by GeCKO-F1/R1; second round by NGS-Lib_Fwd/Rev 1-10. PCR products were purified (Qiagen gel purification kit, Cat#28706, and Qiagen PCR purification kit, Cat#28106), quantified by Qubit (Thermo Fisher Scientific), and profiled by Fragment Analyzer (Advanced Analytical).

Samples were sequenced on an Illumina NextSeq 500 Instrument (Illumina) according to the manufacturer's instructions, yielding 24 to 36 million single-end 75 nucleotide reads per sample. Model-based Analysis of Genome-wide CRISPR/Cas9 Knockout (MAGeCK, v.0.5.9.2) was used to analyze the data. Reads were

trimmed of their adapters with bcl2fastq (v2.19) and quality-controlled with fastQC (v0.11.5). Read count table was obtained using the MAGeCK count command with default parameters on the Human GeCKOv2 combined library of A and B. Gene summary table was filtered to remove non-targeting controls, genes with less than four guide RNAs and non-annotated genes. GSI and DMSO-treated samples were compared using the MAGeCK test command with default parameters, producing Robust Rank Aggretion (RRA) tables. Further analysis was performed with MAGeCKFlute (v2.2.0). Genes with $P < 0.001$ and a log2 fold change $> 1$ or $< -1$ were considered as positive and negative significant hits, respectively (Dataset EV2).

## ChIP-seq

Chromatin immunoprecipitation followed by sequencing (ChIP-seq) was performed on the chromatin of MB157-iEV, MB157-iSOX2, MB157R-iEV, and MB157R-iN1-ICD cells treated with 1 µg/mL DOX (Merck, Cat#D9891) for 72 h. A double cross-linking fixation method was used. Briefly, adherent cells were directly fixed into the dishes with 2 mM Disuccinimidyl glutarate (DSG, Santa Cruz, Cat# sc-285455) and 1 mM $MgCl_2$ in 8 mL 1× PBS for 45 min at RT. Then cells were cross-linked with 1% methanol-free formaldehyde (Pierce Life Technologies, Cat# 28906) and quenched with 0.125 M glycine. Cells were lysed with lysis buffer LB1 (50 mM HEPES-KOH pH 7.4, 140 mM NaCl, 1 mM EDTA, 0.5 mM EGTA, 10% glycerol, 0.5% NP-40, 0.25% Triton X-100, Protease inhibitor cocktail (Sigma, Cat# P8340)) twice during 10 min at 4 °C. Nuclear lysis was performed using LB2 buffer (10 mM Tris-HCl pH 8.0, 200 mM NaCl, 1 mM EDTA, 0.5 mM EGTA, Protease inhibitor cocktail (Sigma, Cat# P8340)) for 10 min at 4 °C. The cell pellet was rinsed twice with SDS shearing buffer (10 mM Tris-HCl pH 8.0, 1 mM EDTA, 0.15% SDS, Protease inhibitor cocktail), resuspended in 1 mL SDS shearing buffer and transferred to a milliTUBE 1 mL AFA fiber. Chromatin was sonicated on a E220 Focused-ultrasonicator (Covaris) using the following settings: 18 min, 200 cycles, 5% duty, 140 W.

The IP reactions were performed following the protocols of the Diagenode iDeal ChIP-seq kit for Transcription Factors (Diagenode, Cat# C01010170). For each IP reaction, 4 million cells were used. Sheared chromatin was immunoprecipitated overnight at 4 °C with HA (Abcam Cat#ab9110, 2 µg), RBPJ (CST Cat#5313, 1:50) or SOX2 (CST Cat#23064, 1:50) antibodies. Eluted DNA was quantified with Qubit (Thermo Fisher Scientific) and Agilent Fragment Analyzer (Advanced Analytical).

The libraries were prepared from ChIP and input DNA by Gene Expression Core Facility of EPFL. NEBNext Ultra II DNA library prep (New England Biolabs, Cat#E7103) starting from 3 ng of DNA for "Chip" samples and 10 ng of DNA for "input" samples was performed according to the New England Biolabs' protocol NEBNext Ultra II DNA Library Prep Kit (Version 6.1_5/20, internal exp code RPX030). The libraries were quantified by qubit DNA HS and performed profile analysis by TapeStation TS4200. The libraries were sequenced using the Illumina novaseq PE60 platform, with paired-end sequencing and 7–28 million reads per sample. Reads were trimmed to remove NEBNext Ultra II DNA (TruSeq) adaptors with bclconvert (v.3.9.3). Mapping and peak calling were performed with BWA and MACS2 (nf-core/chip-seq version 2.0.0) on the human genome hg38 with the following parameters: macs_gsize = 2.7e9, macs_fdr = 0.05, narrow_peak = true, all other parameters were left to default. Peak calling was performed on pooled replicates to increase sequencing depth.

### Peak co-localization

We computed the minimum genomic distances between peak summit (highest scoring position) pairs RBPJ/SOX2. Co-localizing peaks are defined as pairs with summits distant by less than 500 bp. Randomly sampled positions in enhancer regions in the MB157 cell line (Petrovic et al, 2019) were used as background control. $P$ values were then computed by applying Fisher test on the proportions of co-localizing peaks in experimental versus background cases. Co-localizing peak pairs were scanned for the presence of RPBJ and SOX2 BS using FIMO (MEME suite v.5.5.5) (Bailey et al, 2009) and the matrices in JASPAR/CISBP database.

### Peak representation

IGV was used to visualize HA (SOX2), SOX2 and RBPJ peaks in the different regions of interest. HES1 promoter, MYC enhancer et CCND1 enhancer regions were defined according the genomic positions published by Petrovic et al (Petrovic et al, 2019).

### ChIP-qPCR validation

The two major peaks on *HES1* and *NOTCH1* promoters were validated by ChIP-qPCR using specific primers indicated in Table EV2.

## Luciferase assay

Cells were plated in 96-well plates (15,000 cells/well) and were co-transfected 24 h later with pGL4.26_MycE1_Luc or pGL4.26_12xCSL_Luc and SV40_Renilla plasmids using Fugene HD (Promega, Cat#E2311). Two hours post-transfection, DOX at 1 µg/mL (Merck, Cat#D9891) or GSI at 1 µM (Spirochem, LY3039478) was added to the cells. Cells were incubated for 24–72 h at 37 °C and bioluminescence signal was measured using Dual-luciferase kit (Promega, Cat#E1910) as recommended by the manufacturer's instructions.

## Drug screens

The drug synergy screen was performed in several steps using a library of 1536 FDA-approved drugs (Dataset EV3) pulsed in 384-well plates. MB157 and HCC1599 cells were automatically plated at 1400 cells/well and incubated for 6 days, before viability readout using PrestoBlue (Invitrogen, Cat# A13261). A Z' score >0.5 was used to technically validate the drug screen and then compounds were selected based on the best viability inhibition compared to control. In the first step, the full library of 1536 drugs were tested at 10 µM, and then the 140 best compounds were filtered at 1 µM. From this second screen, the 41 best compounds were tested in a range of concentration from 8 nM to 100 µM. Based on the IC50 and the Hill slope ($0.5 < x < 5$) of the 3rd screen, the best 13 compounds were tested for synergy with GSI (Spirochem, LY3039478) using a matrix of concentration from 8 nM to 100 µM for drug X and 0.08 nM to 1 µM for GSI. The synergy of the different combinations was analyzed using the classical Highest single-agent model. Finally, the combination GSI-DTB (Apollo

Scientific, Cat#OR302638) was validated manually in 96-well plates with 5000 MB157 or 10,000 HCC1599 cells/well for 6 days.

The drug screen for the GSI-resistant MB157R and HCC1806 cells was performed with the full library of 1536 FDA-approved drugs at 10 μM. MB157R and HCC1806 cells were automatically plated at 1500 (MB157R) or 700 (HCC1806) cells/well and incubated for 3 days, before viability readout using PrestoBlue. Subsequently, the 26 common hits between the two GSI-sensitive MB157 and HCC1599 and the two GSI-resistant MB157R and HCC1806 cells were tested in a range of concentration from 8 nM to 100 μM and DTB was the best hit with higher sensitivity in GSI-resistant cells (Dataset EV4).

## Animal models

Animal experiments were performed in accordance with protocols approved by the Service de la Consommation et des Affaires Vétérinaires of Canton de Vaud (VD3822). NOD.Cg-*Prkdc^scid Il2rg^tm1Wjl*/SzJ mice (NSG) were purchased from Jackson Laboratories. Mice were housed in groups of 2–5 under pathogen-free conditions in the animal facility of EPFL under a 12-h light/12-h dark cycle (from 7 a.m. to 7p.m.) at $21 \pm 1$ °C. Female mice (8–12 weeks) were anesthetized by intraperitoneal injection using 10 mg/kg xylazine (Graeub) and 75 mg/kg ketamine (Graeub). The sample size was estimated by power analysis, according to http://www.3rs-reduction.co.uk, using a significance level of $P = 0.05$ and a statistical power of 80%. Intraductal injections of single-cell suspensions were performed as described (Sflomos et al, 2016). Briefly, 100,000 (MB157, HCC1806, HCC1599) or 500,000 (MB157R) GFP-luciferase-expressing tumor cells were injected intraductally in 1× PBS using a 33-gauge Hamilton pipette (Hamilton, Cat# HA-80508-22) under binocular loop in NSG mice.

Tumor growth was monitored once or twice a week by blinded caliper measurement. Tumor size was estimated using the following formula: length * width$^2$ *1/2. Images were acquired and analyzed using Living Image Software version 4.4 (Caliper Life Sciences, Inc.). Mice were euthanized according to the legal endpoint for animal experimentation, at the latest when the mean tumor size of a group reached 1000 mm$^3$.

Engrafted mammary glands were harvested at endpoint (4–20 weeks after intraductal injections), fixed in 4% formaldehyde for HE and IHC, or snap-frozen in liquid nitrogen for RNA and protein extraction. For ex vivo bioluminescence measurements, mice were first injected with 300 mg/kg luciferin (Biosynth AG, Cat# L-8220), for 6 min, then injected with 150 mg/kg pentobarbital. Resected lungs were then imaged by IVIS 14 min after luciferin injection. Images were acquired and analyzed using Living Image Software version 4.4 (Caliper Life Sciences, Inc.). Metastasis were counted based on the foci detected on the IVIS images.

For overexpression models, DOX (Merck, Cat#D9891) was given to the mice in the drinking water at 0.5 mg/mL together with saccharine at 0.5 mg/mL (changed once a week), starting 1 week post-injection.

For drug treatment, mice were randomized according to tumor size and treated with vehicle, GSI (Spirochem, LY3039478), DTB (Apollo Scientific, Cat#OR302638) or Paclitaxel (Selleckchem, Cat#S1150) once tumors were palpable (around 100 mm$^3$). GSI: IP, 8 mg/kg, 3×/week in 10% DMSO (AppliChem, Cat#A3672.0100), 90% Corn oil. DTB: gavage, 15 mg/kg, 5×/week in 44% propylene glycol, 0.5% acetic acid. Paclitaxel: IP, 8 mg/kg, 1×/week in 5% DMSO, 40% PEG300, 5% Tween80. Mice without tumors or with tumors <20 mm$^3$ at the start of the treatment were excluded from the study. Tumor growth was monitored until the endpoint.

## Histology

Mammary tumors were fixed in 4% formaldehyde overnight at 4 °C, washed with 70% ethanol, paraffin-embedded (Tissue-Tek VIP® 6AI; Sakura) and sectioned at 4 μm. Sections were de-waxed, re-hydrated, and endogenous peroxidases were quenched using 1% H$_2$O$_2$. Human TNBC TMA Br931 was purchased from TissueArray. Heat-induced epitope retrieval was performed (10 mM trisodium citrate buffer, pH 6, 20 min at 95 °C or 10 mM Tris-EDTA buffer, pH 9, 20 min at 95 °C). Primary antibodies (Reagents and Tools Table) were incubated at 4 °C overnight. After three washes in 1× PBS, secondary antibodies ImmPRESS HRP anti-Rabbit IgG (Vector laboratories, Cat#MP-7401), or ImmPRESS HRP anti-Goat IgG (Vector laboratories, Cat#MP-7405) were incubated for 1 h at RT followed by 3,3'-Diaminbenzidine (DAB, Sigma-Aldrich, Cat#D5905) revelation. Sections were counterstained with Harris hematoxylin. For IF, sections were incubated overnight at 4 °C with N1-ICD and SOX2 primary antibodies. Fluorescent secondary antibodies anti-Goat (Thermo Fisher Scientific, Cat# A-11057) or anti-Rabbit (Thermo Fisher Scientific, Cat# A-31573), were incubated for 1 h at RT before being mounted using Fluoromount-G (SouthernBiotech, Cat# 0100-01) and imaged using the Olympus VS200 whole-slide scanner. For quantification, images were analyzed with QuPath 0.5.0 (Bankhead et al, 2017). Briefly, using a custom script, tissues were annotated, and different tissue regions were detected using a pixel classifier. Then, cells were detected within regions of interest, namely invasive and ductal tumor regions, and classified according to DAB or fluorescence intensity of the staining.

## Patient data analysis

### Data collection

RNA sequencing, whole exome sequencing (WES), microarray, copy number alterations and survival data were obtained from the TCGA_BRCA (https://gdac.broadinstitute.org/runs/stddata__2016_01_28/data/BRCA/20160128/) and METABRIC (https://www.nature.com/articles/nature10983) cohorts using cBioPortalData package (v2.2.11). While the METABRIC dataset cohort only contains primary tumors, a low number of metastatic samples ($n = 7$) are available in the TCGA dataset cohort. However, they were excluded to focus the analysis on primary tumors only. Moreover, METABRIC provides recurrence-free survival data in addition to overall survival. Estrogen receptor (ER), Progesterone receptor (PR), and HER2 receptor calls reported in Lehmann et al, 2021 (Lehmann et al, 2021) were used to stratify patients in TNBC (i.e., ER−/PR−/HER2−), HER2 (HER2 +) or luminal (ER + /HER2−) subtypes.

### NOTCH alterations

Within each patient, NOTCH was classified as mutated if a missense mutation was reported for any of NOTCH1, NOTCH2, NOTCH3 or NOTCH4, while it was classified as amplified if a "cna" value of 2 (indicating high-level amplification) was reported for any of them.

## NOTCH signature

We curated a novel NOTCH signature by integrating data from prior studies by Petrovic et al (Stoeck et al, 2014; Petrovic et al, 2019), which focused on gene regulation upon GSI-washout. We used a fold-change threshold of 1 while considering the union across the various cell lines. Subsequently, we intersected this regulatory signature with genes identified in our N1-ICD gain-of-function RNAseq analysis (MB157R-iN1-ICD vs iEV), examining the impact of N1-ICD gain-of-function in MB157R cells, applying a fold-change threshold of 1 and an adjusted $p$ value of 0.05. Finally, we integrated this intersected set with a canonical collection of manually curated targets. Our refined signature encompasses a total of 77 genes ("ACTBL2", "ADAMTSL4", "ANKRD1", "ARRDC4", "BMP6", "CCND1", "CDK5R1", "CERS1", "CPNE7", "CXCL2", "CYP1B1", "DSG3", "DUSP5", "E2F2", "ENC1", "FAT2", "FBLN7", "FGFBP1", "FLT1", "GADD45A", "GAS5", "HES1", "HES2", "HES4", "HES5", "HEY1", "HEY2", "HEYL", "ID3", "IER3", "IFFO2", "IGFBP3", "IL12A", "IL20RA", "INHBB", "JAG1", "JPH2", "KCNG1", "KCNK5", "KRT5", "KRT6A", "KRT6B", "KRT6C", "KRT75", "LAMA4", "LURAP1L", "LYPD5", "MMP7", "MT1X", "MYC", "MYCL", "MYEOV", "NCR3LG1", "NOTCH3", "NRARP", "OLFM4", "P2RY6", "PAPPA", "PDZD2", "PI3", "PLK2", "PMAIP1", "PRR5L", "RAB11FIP1", "RHOV", "SAT1", "SCD5", "SERPINA3", "SERPINB9", "SNAI1", "STARD4", "TFRC", "TGFB2", "XK", "YPEL2", "ZNF469", "ZNF750").

To calculate NOTCH signature scores, the median expression of each signature genes was computed and a HIGH or LOW value to the patient was attributed for each of them, next the total count of signature HIGH values was tallied for each patient. Finally, the patients were stratified in three equally sized groups from the lowest to the highest signature scores (NOTCH signature LOW, INT, and HIGH).

### Survival analysis

Survival analysis has been performed using survminer (v0.4.9) and survival (v3.2-7) packages. Briefly, the Surv function was used to construct a survival object for time-to-event data, and the Survfit function fitted a survival curve to the data. Kaplan–Meier survival estimates were then used for visualization of survival curves in relation to different categorical variables. In our case, patients were stratified into NOTCH signature HIGH or LOW as described above, as well as SOX2 HIGH or LOW based on 33% lowest or highest expression of SOX2. The observation period for survival data was capped at a maximum duration of 10 years. Log-rank statistical tests were performed to assess the significance of the comparison results.

## Quantification and statistical analysis

Statistical tests were performed using GraphPad Prism (v.10.2.1). Pairwise Student $t$ test was used for comparison of data in paired observations with two unmatched groups. For most of the experiments, a one-way ANOVA with Bonferroni post hoc test was used for comparison of data in paired observation experiments containing three unmatched groups or more. For proliferation curves, two-way ANOVA was used to compute the $P$ value. For proportions of overlapping ChIP peaks, Fisher test was used to determine the $P$ value. For percentages of tissue types on tissue sections, Cochran–Mantel–Haenszel test (CMH) was used to determine the $P$ value. The results are expressed as average $\pm$ SEM and $P$ values are indicated for comparisons of interest.

## Graphics

The visual abstract and schemes graphics were created with BioRender.com.

## Data availability

Primary datasets generated in this study: The CRISPR-Cas9 screen, RNAseq and ChIP-seq datasets are submitted to the GEO database under the accession code GSE270368, GSE262001, and GSE262007, respectively.

The source data of this paper are collected in the following database record: biostudies:S-SCDT-10_1038-S44321-024-00161-8.

## Peer review information

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

## Acknowledgements

The authors acknowledge Thomas Zwahlen for bioinformatic analysis, and thank Ute Koch and Amber Bowler, for critical reading of the manuscript, Jelena Zaric for scientific and technical advice, Christine Göpfert for pathological analysis of xenografts, the staff from the Histology Core Facility, Gene Expression Core Facility, Flow Cytometry Core Facility, Biomolecular Screening Facility, BioImaging and Optics Platform and Center of Phenogenomics at École Polytechnique Fédérale de Lausanne for excellent technical support. This work was supported in part by the "FOND'ACTION contre le cancer" to FR.

## Author contributions

**Morgane Fournier**: Data curation; Formal analysis; Validation; Investigation; Visualization; Methodology; Writing—review and editing. **Joaquim Javary**: Data curation; Formal analysis; Validation; Investigation; Visualization; Methodology; Writing—review and editing. **Vincent Roh**: Data curation; Formal analysis; Investigation; Writing—review and editing. **Nadine Fournier**: Data curation; Formal analysis; Investigation; Writing—review and editing. **Freddy Radtke**: Conceptualization; Supervision; Funding acquisition; Writing—original draft; Project administration.

Source data underlying figure panels in this paper may have individual authorship assigned. Where available, figure panel/source data authorship is

listed in the following database record: biostudies:S-SCDT-10_1038-S44321-024-00161-8.

## Disclosure and competing interests statement

The authors declare no competing interests.

# Expanded View Figures

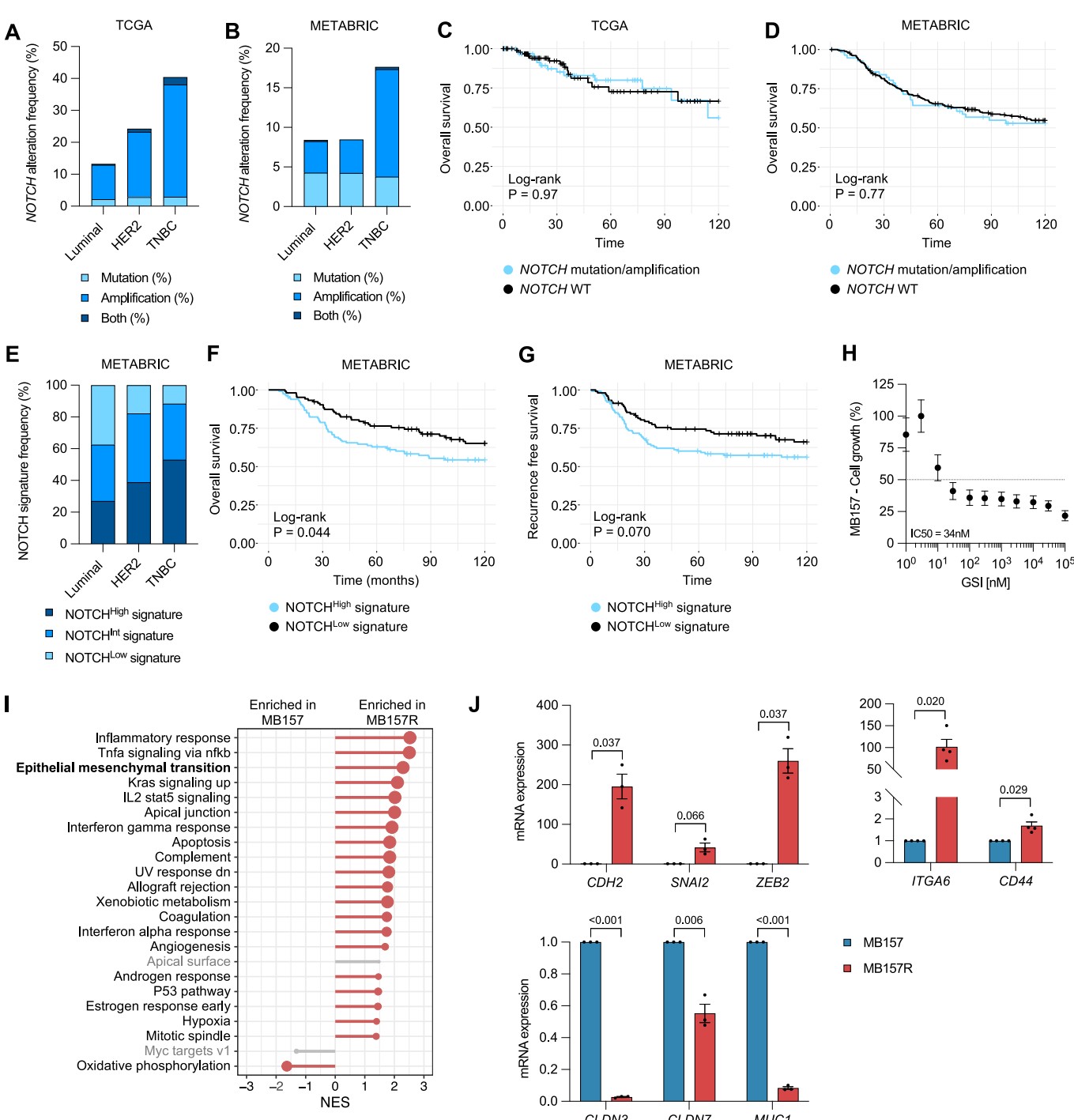

**Figure EV1.  Chronic exposure of NOTCH-driven TNBC cells to GSI induces drug resistance which is associated with EMT and CSC features.**

(A) *NOTCH* mutation and amplification frequency in luminal ($n = 686$), HER2 ($n = 103$) and TNBC ($n = 168$) patients from TCGA dataset ($n = 957$). (B) *NOTCH* mutation and amplification frequency in luminal ($n = 1332$), HER2 ($n = 212$) and TNBC ($n = 317$) patients from METABRIC dataset ($n = 1861$). (C) OS of TNBC patients from TCGA dataset with *NOTCH* wild-type ($n = 100$) or altered ($n = 68$). (D) OS of TNBC patients from METABRIC dataset with *NOTCH* wild-type ($n = 261$) or altered ($n = 56$). (E) Frequency of NOTCH$^{High}$, NOTCH$^{Int}$ or NOTCH$^{Low}$ signature in luminal ($n = 1395$), HER2 ($n = 26$) and TNBC ($n = 347$) patients from METABRIC dataset ($n = 1968$). (F) OS of TNBC patients from METABRIC dataset with NOTCH$^{High}$ ($n = 114$) or NOTCH$^{Low}$ ($n = 103$) signature. (G) RFS of TNBC patients from METABRIC dataset with NOTCH$^{High}$ ($n = 114$) or NOTCH$^{Low}$ ($n = 103$) signature. (H) Cell growth inhibition of MB157 cells treated with GSI (1 μM) for 6 days, $n = 4$. (I) Hallmark GSEA from RNAseq analysis of MB157R compared to MB157 cells, $n = 3$. (J) Relative mRNA expression of EMT and stemness markers in MB157 and MB157R cells, $n = 3$. Data from biological replicates are represented as mean ± SEM. Log-rank test (C, D, F, G), permutation test (I) or Student $t$ test (J) were used to determine $P$ values.

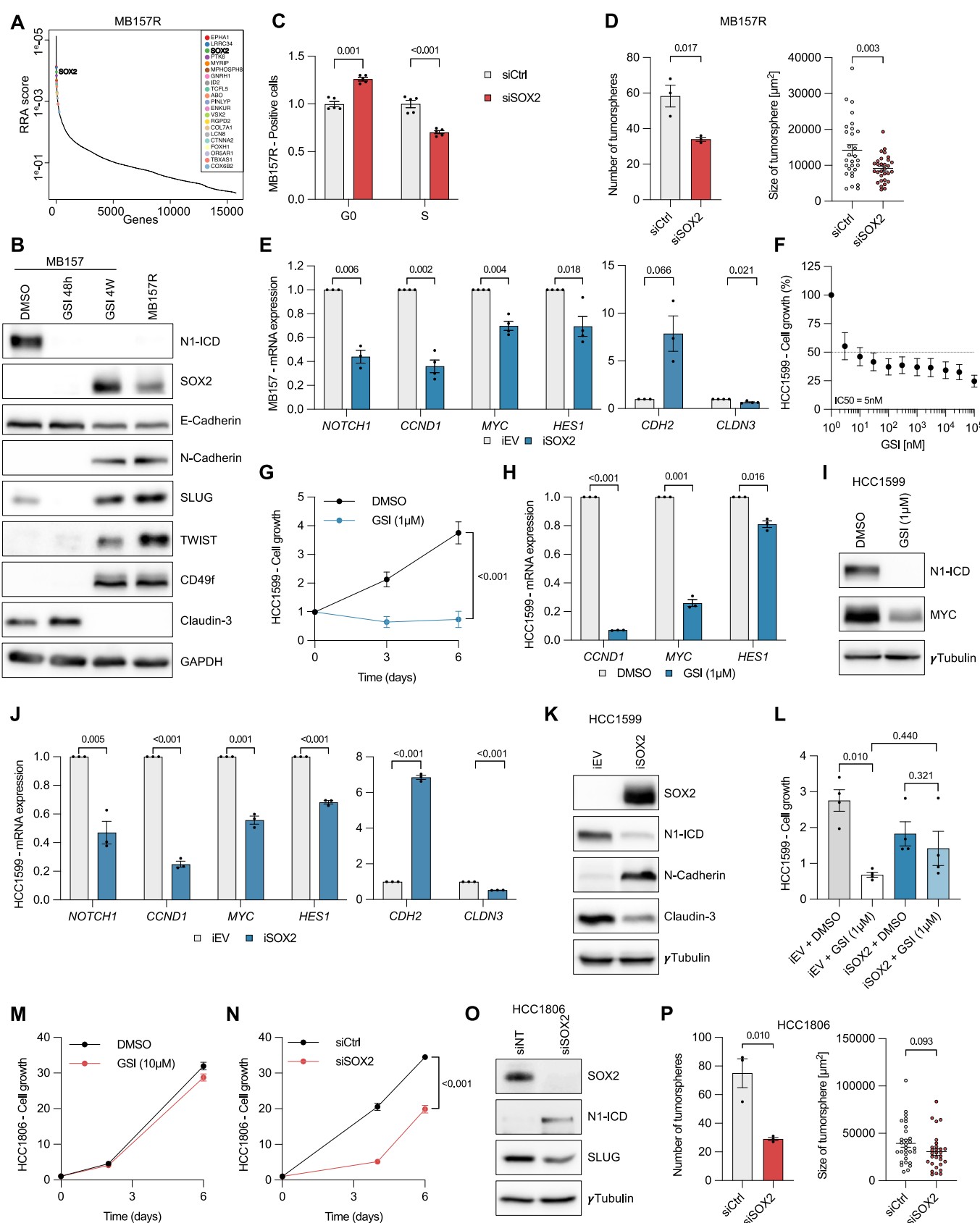

◀ **Figure EV2. SOX2 mediates resistance to GSI in TNBC inhibiting Notch signaling, promoting EMT and CSC features.**

(A) Robust rank aggregation (RRA) of genes negatively selected in CRISPR-Cas9 screen of MB157R cells treated with GSI (10 μM) for 14 days. (B) Representative immunoblotting of N1-ICD, SOX2 and EMT/Stemness markers derived from MB157 or MB157R cells as indicated. (C) Cell cycle analysis of MB157R cells 72 h after transfection with siRNA SOX2 or Ctrl, $n = 5$. (D) Number and size of tumorspheres derived from MB157R cells with siRNA SOX2 or Ctrl, $n = 3$. (E) Relative mRNA expression of *NOTCH1* and its target genes (*CCND1*, *MYC*, *HES1*) and EMT/Stemness markers in iSOX2 or iEV control MB157 cells, 72 h after DOX induction, $n = 3$. (F) Cell growth inhibition of HCC1599 treated with GSI (1 μM) for 6 days, $n = 4$. (G) Cell proliferation assay of HCC1599 cells treated with GSI (1 μM) for 6 days, normalized to day 0, $n = 3$. (H) Relative mRNA expression of *CCND1*, *MYC* and *HES1* in HCC1599 treated with GSI (1 μM) for 24 h, $n = 3$. (I) Representative immunoblotting of N1-ICD and MYC derived from HCC1599 cells treated with GSI (1 μM) for 24 h, $n = 3$. (J) Relative mRNA expression of *NOTCH1* and its target genes (*CCND1*, *MYC*, *HES1*) and EMT/Stemness markers, $n = 3$ and (K) Representative immunoblotting of N1-ICD, SOX2, N-Cadherin and Claudin-3 derived from iSOX2 or iEV control HCC1599 cells, 72 h after DOX induction, $n = 3$. (L) Cell proliferation assay of iSOX2 or iEV control HCC1599 cells treated with GSI (1 μM) or VHC. Cell growth was assessed 6 days post treatment and normalized to day 0, $n = 4$. (M) Cell proliferation assay of HCC1806 cells treated with GSI (10 μM) for 6 days, normalized to day 0, $n = 3$. (N) Cell proliferation assay of HCC1806 cells 72 h after transfection with siRNA SOX2 or Ctrl, normalized to day 0, $n = 3$. (O) Representative immunoblotting of N1-ICD, SOX2 and SLUG derived from HCC1806 cells 72 h after transfection with siRNA SOX2 or Ctrl, $n = 3$. (P) Number and size of tumorspheres derived from HCC1806 cells with siRNA SOX2 or Ctrl, $n = 3$. Data from biological replicates are represented as mean ± SEM. Student *t* test (**C–E**, **H**, **J**, **P**) two-way ANOVA (**G**, **N**) or one-way ANOVA (**L**) were used to determine *P* value.

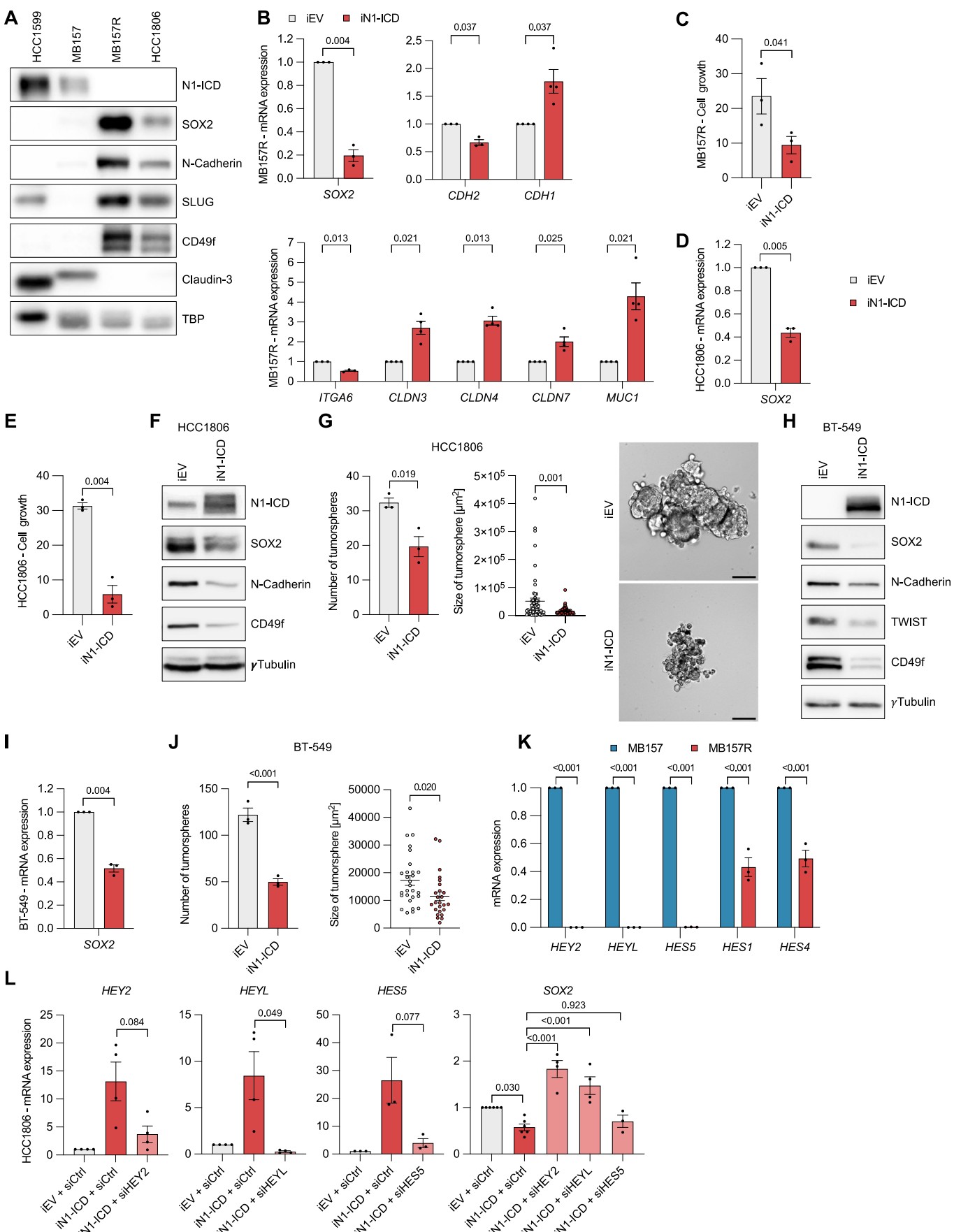

◀ **Figure EV3. Reciprocal SOX2 inhibition is mediated through Notch downstream transcriptional repressors of the HEY family.**

(A) Representative immunoblotting of N1-ICD, SOX2 and EMT/Stemness markers derived from HCC1599, MB157, MB157R and HCC1806 cells. (B) Relative mRNA expression of *SOX2* and EMT/Stemness markers in iN1-ICD or iEV control MB157R cells, 72 h after DOX induction $n = 3$–4. (C) Cell proliferation assay of iN1-ICD or iEV control MB157R cells. Cell growth was assessed 6 days after DOX induction and normalized to day 0, $n = 4$. (D) Relative mRNA expression of *SOX2* in iN1-ICD or iEV control HCC1806 cells, 72 h after DOX induction, $n = 3$. (E) Cell proliferation assay of iN1-ICD or iEV control HCC1806 cells. Cell growth was assessed 6 days after DOX induction and normalized to day 0, $n = 3$. (F) Representative immunoblotting of N1-ICD, SOX2 and EMT/Stemness markers derived from iN1-ICD or iEV control HCC1806 cells, 72 h after DOX induction. (G) Number and size of tumorspheres with representative pictures derived from iN1-ICD compared to iEV control HCC1806 cells after 14 days, $n = 3$. Scale = 100 μm. (H) Representative immunoblotting of N1-ICD, SOX2 and EMT/Stemness markers and (I) Relative mRNA expression of *SOX2* in iN1-ICD or iEV control BT-549 cells, 72 h after DOX induction $n = 3$. (J) Number and size of tumorspheres derived from iN1-ICD compared to iEV control BT-549 cells after 14 days, $n = 3$. (K) Relative mRNA expression of *HEY/HES* family genes in MB157 and MB157R cells, $n = 3$. (L) Relative mRNA expression of *HEY2*, *HEYL*, *HES5* and *SOX2* in iN1-ICD or iEV control HCC1806 cells 72 h after transfection with siRNA HEY2, HEYL, HES5 or Ctrl $n = 3$–4. Data from biological replicates are represented as mean ± SEM. Student *t* test (B–E, G, I–L) or one-way ANOVA (L) were used to determine *P* value.

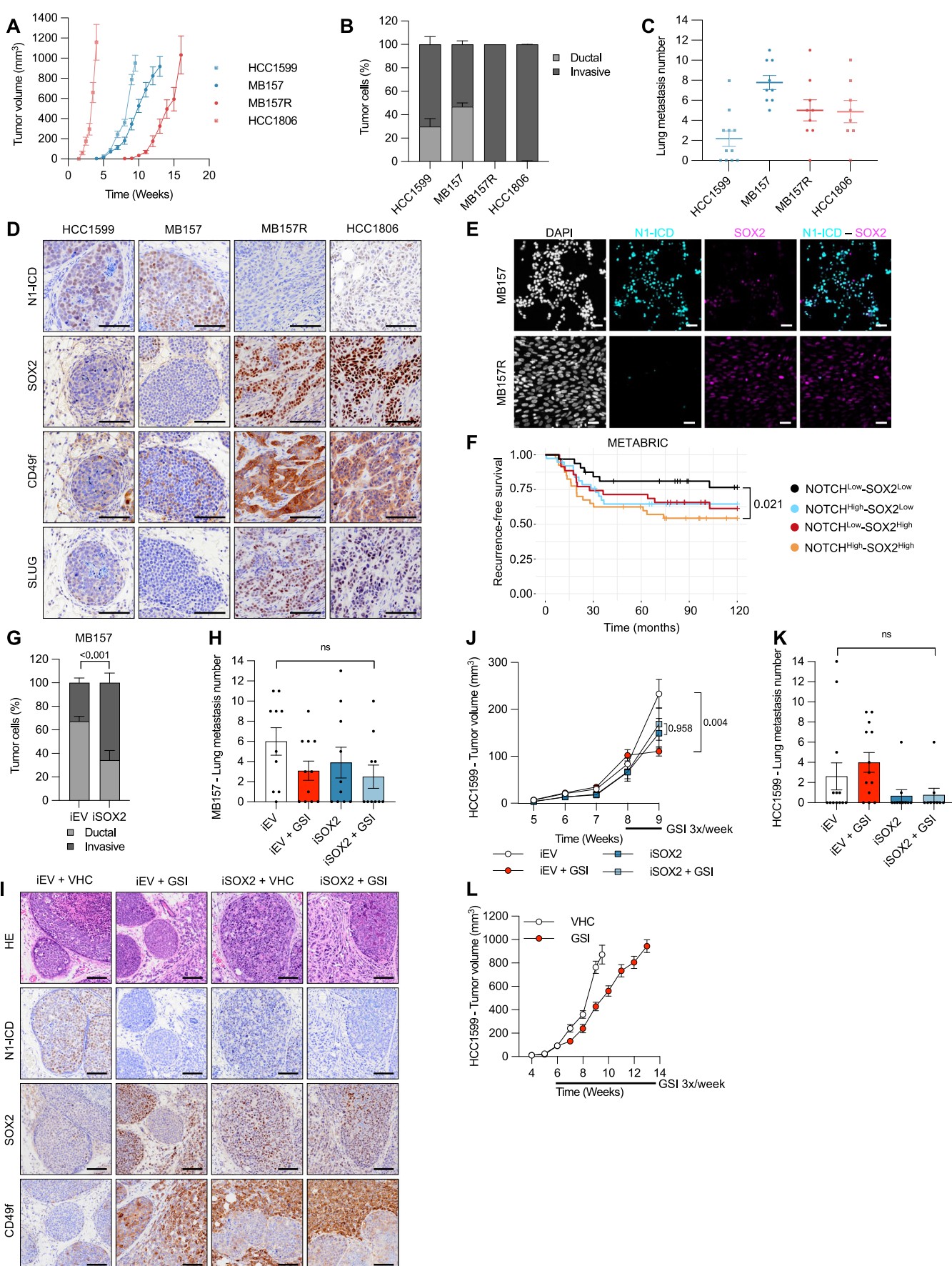

◄ **Figure EV4.  Escape of GSI-mediated in vivo tumor growth control due to TNBC tumor cell plasticity.**

(A) Tumor growth of HCC1599, MB157, MB157R and HCC1806 MIND xenografts ($n = 9$–11). (B) Quantification of tumor cells in invasive or ductal areas from HE coloration in HCC1599, MB157, MB157R and HCC1806 MIND xenografts at endpoint ( ~ 1000 mm³) ($n = 3$). (C) Lung metastasis number in HCC1599, MB157, MB157R and HCC1806 MIND xenografts at endpoint, $n = 9$–11. (D) Representative pictures of N1-ICD, SOX2, CD49f and SLUG IHC stainings for HCC1599, MB157, MB157R and HCC1806 MIND xenograft tumors at endpoint, scale $= 100$ μm. (E) Representative images of co-immunofluorescence staining of N1-ICD – SOX2 for MB157 and MB157R cell lines in vitro, scale $= 50$ μm. (F) RFS of TNBC patients from METABRIC dataset with NOTCH$^{High/Low}$ signature and SOX2$^{High/Low}$ expression. TNBC patients are divided in 4 groups: NOTCH$^{High}$/SOX2$^{High}$ ($n = 40$), NOTCH$^{High}$/SOX2$^{Low}$ ($n = 38$), NOTCH$^{Low}$/SOX2$^{High}$ ($n = 35$) and NOTCH$^{Low}$/SOX2$^{Low}$ ($n = 32$). (G) Quantification of tumor cells in invasive or ductal areas from HE coloration in MB157-iSOX2 xenografts, $n = 3$. (H) Lung metastasis number in SOX2-expressing or EV control MB157 MIND xenografts treated with GSI (8 mg/kg, 3×/week) or VHC for 1 week, $n = 10$–11. (I) Representative pictures of HE coloration, N1-ICD, SOX2 and CD49f IHC stainings for iSOX2 or iEV control MB157 MIND xenografts treated with GSI (8 mg/kg, 3×/week) or VHC for 1 week, scale $= 100$ μm. (J) Tumor growth of iSOX2 or iEV control HCC1599 MIND xenografts treated with GSI (8 mg/kg, 3×/week) or VHC for 1 week, $n = 9$–13. (K) Lung metastasis number in iSOX2 or iEV control HCC1599 MIND xenografts treated with GSI (8 mg/kg, 3×/week) or VHC for 1 week, $n = 9$–13. (L) Tumor growth of HCC1599 xenografts treated with GSI (8 mg/kg) or VHC for 8 weeks, $n = 9$–13. Data from biological replicates are represented as mean ± SEM. Log-rank test (F), Cochran–Mantel–Haenszel test (G), one-way ANOVA (H, K) or two-way ANOVA (J), were used to determine $P$ value.

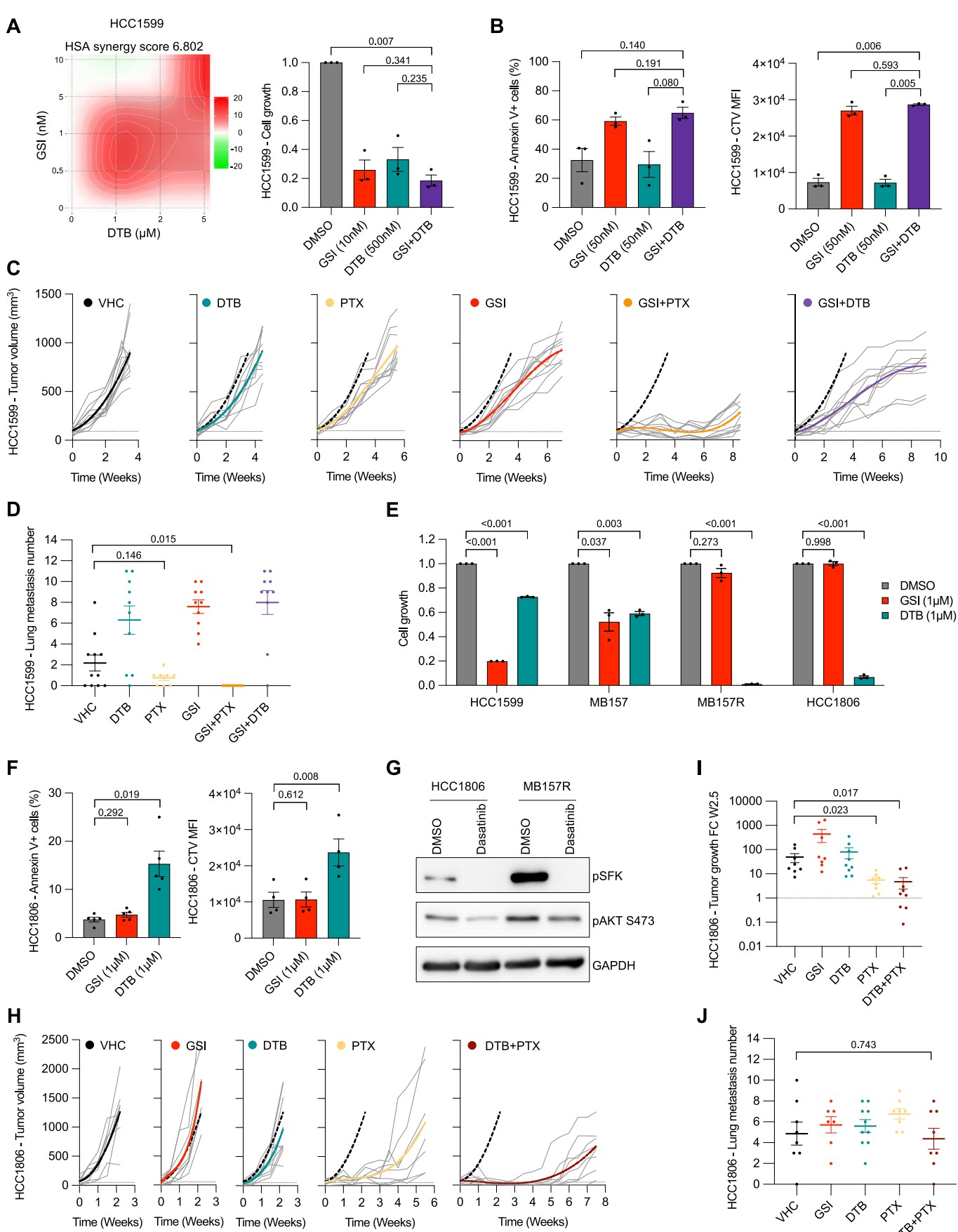

◀ **Figure EV5. Combination therapies to treat GSI-sensitive and -resistant TNBC xenografts.**

(A) HAS-synergy heatmap derived from HCC1599 cells treated with a concentration matrix of GSI-DTB for 6 days and cell growth inhibition of HCC1599 cells treated with GSI (10 nM) and/or DTB (500 nM) for 6 days, $n = 3$. (B) Proportion of Annexin V+ cells ($n = 4$) and Cell Trace Violet mean fluorescence intensity (MFI) ($n = 3$) in HCC1599 cells treated with GSI (50 nM) and/or DTB (50 nM) for 6 days, by flow cytometry analysis. (C) Tumor growth of HCC1599 xenografts treated with GSI (8 mg/kg, 3×/week), DTB (15 mg/kg, 5×/week), PTX (15 mg/kg, 1×/week) or VHC, $n = 9$–11. (D) Lung metastasis number in HCC1599 xenografts treated with GSI (8 mg/kg, 3x/week), DTB (15 mg/kg, 5x/week), PTX (15 mg/kg, 1×/week) or VHC, $n = 8$-11. (E) Cell growth inhibition of HCC1599, MB157, MB157R and HCC1806 cells treated with GSI and/or DTB for 6 days, $n = 3$. (F) Proportion of Annexin V+ cells ($n = 5$) and Cell Trace Violet MFI ($n = 4$) in HCC1806 cells treated with GSI (1 μM) or DTB (1 μM) for 3 days, by flow cytometry analysis. (G) Representative immunoblotting of pSFK and pAKT S473 derived from MB157R and HCC1806 cells treated with Dasatinib 1 μM for 18 h (H) Tumor growth of HCC1806 MIND xenografts treated with GSI (8 mg/kg, 3x/week), DTB (15 mg/kg, 5×/week), PTX (15 mg/kg, 1×/week) or VHC, $n = 8$–10. (I) Tumor growth fold change (2.5 weeks after treatment) of HCC1806 MIND xenografts treated with GSI (8 mg/kg, 3×/week), DTB (15 mg/kg, 5x/week), PTX (15 mg/kg, 1×/week) or VHC, $n = 6$–9. (J) Lung metastasis number in HCC1806 MIND xenografts treated with GSI (8 mg/kg, 3×/week), DTB (15 mg/kg, 5×/week), PTX (15 mg/kg, 1×/week) or VHC, $n = 8$-10. Data from biological replicates are represented as mean ± SEM. One-way ANOVA (A, B, D–F, I, J) was used to determine $P$ value.

