## [Peer Review File · EMBO Molecular Medicine]

Reciprocal inhibition of NOTCH and SOX2 shapes tumor cell plasticity and therapeutic escape in TNBC

Morgane Fournier, Joaquim Javary, Vincent Roh, Nadine Fournier, and Freddy Radtke

Corresponding author: Freddy Radtke (Freddy.Radtke@epfl.ch)

Review Timeline:

Submission Date:	7th May 24
Editorial Decision:	18th Jun 24
Revision Received:	5th Sep 24
Editorial Decision:	7th Oct 24
Revision Received:	15th Oct 24
Accepted:	16th Oct 24

Editor: Lise Roth

Transaction Report:

18th Jun 2024

Dear Prof. Radtke,

Thank you for the submission of your manuscript to EMBO Molecular Medicine, and please accept my apologies for the unusual delay in getting back to you, as one referee needed more time to complete his/her report.

We have now received feedback from the three reviewers, and as you will see from the reports below, they acknowledge the interest of the study and are overall supporting publication of your work pending appropriate revisions. In particular, concerns on the choice of cell line / use of a single cell line, contradicting results with existing literature, and mechanism of reciprocal interaction Sox2/Notch will need to be adequately addressed.

If you feel you can satisfactorily address these points and those listed by the referees, you may wish to submit a revised version of your manuscript.

Please note that upon further cross-commenting with the referees, we agreed that 'other comments' 4, 18 and 21 from referee #3 would NOT need to be addressed.

Please attach a covering letter giving details of the way in which you have handled each of the points raised by the referees. A revised manuscript will once again be subject to review, and we cannot guarantee at this stage that the eventual outcome will be favorable.

We require:

- 1) A .docx formatted version of the manuscript text (including legends for main figures, EV figures and tables). Please make sure that the changes are highlighted to be clearly visible.
- 2) Individual production quality figure files as .eps, .tif, .jpg (one file per figure). For guidance, download the 'Figure Guide PDF' (<https://www.embopress.org/page/journal/17574684/authorguide#figureformat>).
- 3) At EMBO Press we ask authors to provide source data for the main figures. Our source data coordinator will contact you to discuss which figure panels we would need source data for and will also provide you with helpful tips on how to upload and organize the files.
- 4) A .docx formatted letter INCLUDING the reviewers' reports and your detailed point-by-point responses to their comments. As part of the EMBO Press transparent editorial process, the point-by-point response is part of the Review Process File (RPF), which will be published alongside your paper.
- 5) A complete author checklist, which you can download from our author guidelines (<https://www.embopress.org/page/journal/17574684/authorguide#submissionofrevisions>). Please insert information in the checklist that is also reflected in the manuscript. The completed author checklist will also be part of the RPF.
- 6) Please note that all corresponding authors are required to supply an ORCID ID for their name upon submission of a revised manuscript.
- 7) It is mandatory to include a 'Data Availability' section after the Materials and Methods. Before submitting your revision, primary datasets produced in this study need to be deposited in an appropriate public database, and the accession numbers and database listed under 'Data Availability'. Please remember to provide a reviewer password if the datasets are not yet public (see <https://www.embopress.org/page/journal/17574684/authorguide#dataavailability>). In case you have no data that requires deposition in a public database, please state so in this section. Note that the Data Availability Section is restricted to new primary data that are part of this study.
- 8) For data quantification: please specify the name of the statistical test used to generate error bars and P values, the number (n) of independent experiments (specify technical or biological replicates) underlying each data point and the test used to calculate p-values in each figure legend. The figure legends should contain a basic description of n, P and the test applied. Graphs must include a description of the bars and the error bars (s.d., s.e.m.). Please provide exact p values.
- 9) Our journal encourages inclusion of *data citations in the reference list* to directly cite datasets that were re-used and

obtained from public databases. Data citations in the article text are distinct from normal bibliographical citations and should directly link to the database records from which the data can be accessed. In the main text, data citations are formatted as follows: "Data ref: Smith et al, 2001" or "Data ref: NCBI Sequence Read Archive PRJNA342805, 2017". In the Reference list, data citations must be labeled with "[DATASET]". A data reference must provide the database name, accession number/identifiers and a resolvable link to the landing page from which the data can be accessed at the end of the reference. Further instructions are available at .

13) Author contributions: CRediT has replaced the traditional author contributions section because it offers a systematic machine readable author contributions format that allows for more effective research assessment. Please remove the Authors Contributions from the manuscript and use the free text boxes beneath each contributing author's name in our system to add specific details on the author's contribution. More information is available in our guide to authors.

16) As part of the EMBO Publications transparent editorial process initiative (see our Editorial at <http://embomolmed.embopress.org/content/2/9/329>), EMBO Molecular Medicine will publish online a Review Process File (RPF) to accompany accepted manuscripts.

In the event of acceptance, this file will be published in conjunction with your paper and will include the anonymous referee reports, your point-by-point response and all pertinent correspondence relating to the manuscript. Let us know whether you agree with the publication of the RPF and as here, if you want to remove or not any figures from it prior to publication. Please note that the Authors checklist will be published at the end of the RPF.

I look forward to receiving your revised manuscript.

Yours sincerely,

Lise Roth

***** Reviewer's comments *****

Referee #1 (Remarks for Author):

TNBC remains a substantial therapeutic challenge. Activating mutations and a Notch activation signature are linked to poor outcome in TNBC, reported here and by others. This manuscript examines the roles of Notch signaling and the SOX2 transcription factor in shaping the cancer cell phenotype. As NOTCH inhibition via Gamma secretase inhibitor (GSI) blockade is being explored in a clinical trial in TNBC, understanding how resistance may develop in this setting is an important problem, investigated here.

The study begins with developing a GSI resistant MB157 line. Resistance is linked to an EMT type phenotype. A genome wide CRISPR screen with 21 hits is used to identify candidate genes driving resistance. One of these is SOX2. A series of experiments in naïve MB157 makes a compelling case that SOX2 drives the EMT phenotype, downregulation of Notch and GSI resistance, findings confirmed in two further cell lines by loss and over expression. To investigate the mechanism of SOX2 repressing NOTCH, ChIPseq is performed using SOX2 and RBPJk, finding an impressive overlap between peaks. The proteins co-immunoprecipitate and interact at a synthetic Notch reporter.

NOTCH1 ICD and SOX2 reciprocally inhibit each other via downstream target genes.

Cell lines are then engrafted into milk ducts of immune suppressed mice and shown to recapitulate the in vitro findings.

Impressively, the authors then perform a 1536 FDA approved drug screen, identifying dasastinib as a compound that had in vitro activity in GSI resistant cells as a sole agent and in vivo activity in combination with paclitaxel.

Overall this is an impressive study, with well executed experiments and a formidable quantity of data. The conclusions are well supported. It may be worth rewriting the abstract to bring out the findings more clearly, but otherwise there is little to criticise here.

Referee #2 (Remarks for Author):

In this paper, the authors investigated escape mechanisms of NOTCH-driven TNBC when treated with a γ -Secretase inhibitor GSI. Upon gaining resistance to GSI, the authors observed an increase in SOX2, which increased EMT and stemness. They described a reciprocal inhibition between Notch 1 signaling (N1-ICD) and SOX2. SOX2 was shown to inhibit Notch 1 signaling by directly interacting with RBPJ, whereas Notch 1 inhibited SOX2 through the HEY transcriptional repressors (HEY2 and/or HEYL). The authors also identified different drug combinations for GSI-resistant and sensitive cells. Overall, the manuscript addresses an important problem and is very well written. However, the authors need to address the following concerns before it can be accepted.

Potential Limitations/Suggestions:

- One of the major concerns is that, according to the authors, Sox2 and Notch1 negatively regulate each other. Therefore, should it not be expected that upon GSI treatment, Sox2 levels would go up quickly rather than taking 4-5 weeks to acquire that resistance? The authors show in EV5G that one week after GSI treatment, N1-ICD levels decrease while Sox2 increases (H&E). How would that discrepancy with in vitro results be explained?

• There is extensive literature showing how N1-ICD positively regulates stemness (PMID: 23041621) and how N1-ICD positively regulates SOX2 expression (PMID: 23982961). However, this paper argues against this set of literature and uses one cell line to

demonstrate most of their finding.

- The papers did not use the same markers to be consistent across various models; for example, they used GAPDH as a control in one cell model and g-tubulin in the same figure panel on a different cell line. There is limited consistency regarding the EMT and stem cell markers across various experiments. The authors chose the markers based on their interests
- In Fig1-N, There is no significant difference in mammosphere formation ($p=0.049$) and it is truly concerning.
- The authors need to include a mammosphere formation assay for siSox2
- In line 2 of the introduction, the authors define tnbc as such: "malignancy which is classically defined by expression of hormone receptors (HR, estrogen and progesterone receptors) and human epidermal growth factor receptor 2 (HER2)". They could clarify the expression levels of HR and HER2 (defined by expression of...).
- If their hypothesis is true, then overexpressing N1-ICD in any cell line should decrease SOX2, less EMT, and stem cell properties. The authors should use any other TNBC cells to demonstrate this phenomenon.
-
- The authors should explain their rationale behind choosing a cell line that constitutively expresses NOTCH1 in a ligand-independent manner. In the case that it is to use a cell that is NOTCHhigh, they should also clarify why they chose that one over other cell lines overexpressing NOTCH.
-
- In page 7 line 15, the authors cited Fig. EV1D, F while they probably meant to cite Fig1 D,F.
- Conditions in Figure 1: The authors compared MB157 to MB157R which is resistant to GSI (missing the N1-ICD). When comparing MB157R to the sensitive one (MB157), it would be nice to include a comparison of EMT/Stem markers before and after the GSI treatment for MB157. That could then be compared to MB157R.
- According to their subsequent data, the inhibition of N1-ICD should increase SOX-2 levels. Hence, the impact of GSI inhibition on MB157 (not 157R) SOX-2 levels needs to be shown.
- In expanded figure 2, the labeling in the legend is out of order after D, and a figure (M) is mentioned in the legend but absent from the file.
- In Figure 4, the authors performed a washout experiment for iN1-ICD, showing its reversible effects in MB157R. Washing out after iSox2 in MB157 cells would help to see if Sox2's effects are reversible. The same could also be said for GSI treatments.
- In Figure 5 and EV5, the authors claim that: "SOX2 is sufficient to induce GSI resistance in Notch-driven TNBC, associated with increased EMT/Stemness and invasiveness". Sox2 sufficiency to induce stemness and EMT has been shown in previous figures in vitro. However, this figure has not shown its sufficiency to induce EMT and invasiveness (an increase in invasiveness is shown in MB157 after GSI treatment).
- In EV5F/I no statistical analysis was performed on the number of metastasis.

Referee #3 (Comments on Novelty/Model System for Author):

see major comments #4 and 8

Referee #3 (Remarks for Author):

The manuscript titled "Reciprocal inhibition of NOTCH and SOX2 shapes tumor cell plasticity and therapeutic escape in triple-negative breast cancer" by Morgane Fournier and colleagues takes up a vital topic in cancer therapeutics, specifically triple-negative breast cancer (TNBC). The revelation of a reciprocally inhibitory feedback loop between Notch signaling and the pluripotency-associated transcription factor SOX2 brings forth a noteworthy discovery. This has crucial implications in our understanding of the factors modulating divergent cellular states, epithelial-mesenchymal transition, and characteristics of cancer stem cells. Moreover, evaluating both monotherapies and combination drug therapies in NOTCH-inhibitor sensitive and resistant TNBC xenografts provides valuable insight. This could aid in identifying robust second-line and combination treatment strategies, which can potentially enhance tumor growth control and curtail metastatic burden. Its impact is thus underscored in the context of ongoing research. Consequentially, this manuscript should serve as a valuable resource for a range of readers. However, in its current state, the study has several shortcomings that call for extensive revisions to be considered suitable for publication in EMBO Molecular Medicine.

Major comments:

1) The manuscript relies significantly on the premise of reciprocal regulation between NOTCH and SOX2 signaling directing TNBC tumour cell plasticity. However, the heterogeneity related to SOX2 expression inherent to the MB157 parental cell line (as observed in Figure 5C) forces one to wonder if MB157R is simply a selected variant of the SOX2 positive clones. Thus, authors are encouraged to investigate potential clonal heterogeneity within the parental cell line by using Immunofluorescence. If this turns out to be true, the authors should assess the N1-ICD+ and SOX2+ sub-population's plasticity and their spontaneous phenotype switching capability.

2) A significant observation brought forward is the evidence of reciprocal negative regulation between NOTCH and SOX2 signaling. This finding is particularly intriguing. Nonetheless, the authors seem to overlook instances where NOTCH signaling has been implicated in driving stem cell phenotypes in TNBC (refer to PMID: 31379945 as an exemplar). Therefore, I would encourage the authors to rethink their statements concerning a NOTCH <-> SOX2 driven phenotypic switch among cancer stem cell phenotypes. Such a reorientation would likely make their findings more congruous with their observations in Fig1 - indicating

worse survival rates in patients exhibiting high NOTCH signatures. This adjustment would also align their findings with the subsequent survival analyses, demonstrating worst prognosis in NOTCHhigh/SOX2high patients.

3) The final section of the manuscript (Fig6), "Combination therapies to treat GSI-sensitive and -resistant TNBC xenografts," appears to be insufficiently connected to the mechanistic components detailed in Fig1-5. A significant gap is evidently present without a convincing explanation for the sensitivity of GSI-resistant cells to Dasatinib. Moreover, the abrupt introduction of a combination of Paclitaxel and Dasatinib creates confusion. There needs to be a clear reason for choosing this particular combination over any other potential pairing. I would urge a significant review and revision of this section to enhance narrative clarity and highlight the main takeaway. The authors could also explore and provide insights on SRC-specific signaling activity in SOX2high cells, adding depth to the broader context of SOX2 and SRC regulation. Importantly, for completeness and credibility, consider references such as PMID 23009336 on SRC signaling's role in modulating SOX2.

4) To validate the study, it would be prudent to include a repeat of essential experiments utilizing the HCC1599 cell line.

5) The authors must provide the data underlying high-throughput analysis methods (Crispr-screenings as well as drug screening) in supplementary data, matching common good scientific practice now a day (as it is the case also for NGS data).

6) The findings via ChIP sequencing should be corroborated through different methods; ChIP-qPCR could prove useful for cross-verifying crucial binding sites.

7) To confirm the reciprocal regulation between NOTCH and SOX2 and substantiate their experimental methodology, authors should consider overexpressing transcriptionally deficient forms of N1-ICD and SOX2. Subsequently, they must demonstrate that the cell phenotypes and representative target genes are not affected.

8) To robustly support the manuscript's claims about reciprocal negative regulation between NOTCH and SOX2, clinical staining from a TNBC patient collective should be included.

Other comments:

1) Manuscripts should be provided with line numbers to facilitate the reviewing process

2) Please improve the comprehensibility of the main text and employ a standard normalization method to retain the control conditions' variance across all graphs.

3) The manuscript conflicts with earlier publications mentioning the pro-tumorigenic implications of NOTCH signaling in HCC1806 cells (PMID: 23012411). Primarily, this discrepancy occurs in the context of the observations made in figure EV4D-G. The authors should discuss/state these contradicting results.

4) Fig1H: It would be interesting to also test the transcriptional activity of RBPJ/N1-ICD in luciferase assay prior figure 3F to support Notch signaling loss in MB157R as well as upon GSI treatment in this same system.

5) To substantiate their data further, I encourage the authors to reanalyze publicly available scRNA-seq data of TNBC patients, notably showing negative correlation of SOX2 expression with the NOTCH signature in single cells, and showing negative correlation of HEY2 and HEYL on SOX2 expression.

6) Please make the supplementary file #2 available in its original XLS file format, as the PDF version is not user-friendly.

7) Please provide the marker's size for all Western blots in this study (not the band size); also consider including quantification data of the western blot replicates.

8) The scale bars' size should be presented consistently across all figures, preferably in micrometers.

9) Figure 1C: Alter the color code for splicing events, and amend the figure legend for clarity.

10) Figure 1D: An estimation of IC50 for both MB157 and MB157R should be included in the results.

11) Fig1G: Add the missing error bars and statistics to improve the representation.

12) Fig1K: Slight confusion could arise from the current graph title. Redesign the annotations with "Enriched in MB157R" and "Enriched in MB157" to avoid potential misinterpretation (comment eventually also applicable to other bubble plots of this manuscript).

13) Please revise all instances referring to "Mammosphere" to "tumorsphere" throughout the text and the figures.

- 14) Fig. 2F: The authors should provide RT-qPCR of NOTCH levels upon SOX2 overexpression.
- 15) Fig2I: The images appear to be provided with different zooms, skewing the cell size representation. The iSOX2 image showcases sphere aggregates. To get a more objective illustration, it is recommended for authors to instigate the experiment with conditions that discourage cell aggregation. This could be achieved by using lower cell number, or possibly incorporating methyl cellulose.
- 16) Fig3D: The ChIP-seq tracks require the addition of units and scale for better comprehension.
- 17) Fig3E: Inputs are missing and need to be incorporated.
- 18) To validate the statement pillared on findings, "These findings strongly corroborate a working model where SOX2 binding in close vicinity to RBPJ does interfere with the formation of a functional N1-ICD/RBPJ activation transcription complex thus repressing Notch signaling," it is suggested to execute competition binding assays. This can be achieved by progressively increasing either SOX2 or N1-ICD amounts and evaluating the interaction shift with RBJP.
- 19) "BC patient dataset analysis using The Cancer Genome Atlas (TCGA) and the Molecular Taxonomy of Breast Cancer International Consortium (METABRIC) revealed an enrichment for activating mutations and gene amplifications of NOTCH receptors in TNBC compared to HR+ and HER2+ patients (Fig. EV1A,B), indicating that tumor growth within these subgroups could be Notch-driven.": it is not clear what "subgroups" are meant here.
- 20) Fig4D: the authors should improve consistency of their study: ChIP-seq focused on HES1 but the subsequent analyses important for SOX2 regulation focus on HEY2, HEYL, HES5. The authors should therefore provide the respective tracks and validate their observations via other methods (e.g. ChIP-qPCR).
- 21) Fig4D-E: The data presented would strongly benefit from additional verification of the binding of HEY2 and HEYL to the SOX2 regulatory region. Using a method such as ChIP-qPCR for this analysis is recommended. Furthermore, to evaluate the impact of HEY2 and HEYL, an experiment where these are overexpressed in MB157R and/or HCC1806, with an expected result of a downregulation in SOX2 mRNA levels, should be performed. This would provide further support for your findings.
- 22) Fig5E: The take-home message from the METABRIC analysis needs to be clarified for better understanding and learning.
- 23) Fig5F: Alter the color code to improve comprehension. Differences between light grey and white could be challenging to discern.
- 24) Fig5H seems to be an extension of Fig5F. Hence, authors should consider providing information about what happens with iSOX2 transplanted tumors over longer period of time.
- 25) Fig6B: Please provide an explanation for DTB MFI.
- 26) Fig 6C: Please indicate why the 10-week period for GSI alone is not represented.
- 27) Fig 6F: It is not clear why GSI and DTB concentrations used in MB157R are higher than those for MB157, even though IC50 is lower in this cell line. Please explain or correct the graphs accordingly.
- 28) "Efficient targeting of the Notch pathway could, therefore, be beneficial for NOTCHHigh signature TNBC patients." The authors must respect good scientific practice rules and cite the relevant to date available literature! As example, recent publications related to this topic (PMID: 38172915) has to be mentioned (best already in the introduction) and accordingly discussed!
- 29) Typo: "thwarted"
- 30) The authors should provide the graphs of IC50 estimation for the inhibitors utilized in this study, according to the cell system.
- 31) Typo "CCDN1"
- 32) Consistency "(Fig. S4H)" EV4H

Point by point response to referees' comments and summary of revisions to manuscript: EMM-2024-19945

We would like to thank all reviewers for their overall positive evaluation of our manuscript, as well as their insightful and helpful comments, which have helped to improve our study.

Please find below a detailed point-by-point response to the individual comments:

Reviewer #1

Overall, this is an impressive study, with well executed experiments and a formidable quantity of data. The conclusions are well supported. It may be worth rewriting the abstract to bring out the findings more clearly, but otherwise there is little to criticise here.

We want to thank this reviewer for the very positive and supportive statements regarding our study. We appreciated the comment and reworded the abstract within the given word limit of 175 words to better emphasize our findings as suggested. Thank you for the helpful recommendation.

Reviewer #2 (Remarks for Author):

In this paper, the authors investigated escape mechanisms of NOTCH-driven TNBC when treated with a γ -Secretase inhibitor GSI. Upon gaining resistance to GSI, the authors observed an increase in SOX2, which increased EMT and stemness. They described a reciprocal inhibition between Notch 1 signaling (N1-ICD) and SOX2. SOX2 was shown to inhibit Notch 1 signaling by directly interacting with RBPJ, whereas Notch 1 inhibited SOX2 through the HEY transcriptional repressors (HEY2 and/or HEYL). The authors also identified different drug combinations for GSI-resistant and sensitive cells. Overall, the manuscript addresses an important problem and is very well written. However, the authors need to address the following concerns before it can be accepted.

Thank you for the positive assessment of our manuscript and the importance of the scientific problem. Please find underneath our comments how we tried to address your concerns.

Potential Limitations/Suggestions:

1. One of the major concerns is that, according to the authors, Sox2 and Notch1 negatively regulate each other. Therefore, should it not be expected that upon GSI treatment, Sox2 levels would go up quickly rather than taking 4-5 weeks to acquire that resistance? The authors show in EV5G that one week after GSI treatment, N1-ICD levels decrease while Sox2 increases (H&E). How would that discrepancy with *in vitro* results be explained?

This is a very good observation and a valid point. Indeed, it takes at least three weeks until SOX2 expression, EMT and the stem cell marker CD49f are detectable *in vitro* in GSI-treated MB157 cells, please see the kinetic experiment shown underneath. In contrast in the *in vivo* xenotransplantation experiment, SOX2, as well as CD49f, expression increases within one week post GSI treatment (please see new EV Fig. 4H and for quantification Fig. 5H, former EV5G). The expectation that kinetics of SOX2 expression and those of other markers should be similar as expected by this referee is based on the assumption that growth conditions in *in vitro* and *in vivo* using the MIND model correlate. However, this is

unlikely to be the case. *In vitro* the cells grow on plastic in a defined medium with 10% FCS, while *in vivo* the cells grow intra-ductal and are exposed to the mammary gland microenvironment. The physiological environment seems to be more permissive for faster expression of SOX2 and CD49f and potentially other markers, which were not assessed here. The differences in kinetic of CD49f expression between *in vitro* and *in vivo* is similar as for SOX2 and thus consistent with the possibility that these kinetic differences reflect different *in vitro* and *in vivo* growth conditions. Please refer to Figure below and EV Fig. 4H.

2. There is extensive literature showing how N1-ICD positively regulates stemness (PMID: 23041621) and how N1-ICD positively regulates SOX2 expression (PMID: 23982961). However, this paper argues against this set of literature and uses one cell line to demonstrate most of their finding.

We agree that there is some literature associating NOTCH with stemness and SOX2 expression in TNBC as referenced by the two publications cited by this referee. Although Notch signaling has been implicated robustly with stem cell phenotypes for example in the intestine and the developing brain, however the role of Notch signaling in driving a cancer stem cell phenotype in TNBC is most likely more complex and context dependent. The first paper (PMID: 23041621) by Qui et al published in cancer letters in 2013 used two different TNBC cell lines (SUM149 and MDA-MB-231). The authors produced blocking huN1 antibodies directed against the NOTCH1 NRR domain. This antibody was able to reduce tumor growth in xenotransplantation assays of both cell lines (Figure 3A). The ability of the antibody to reduce NICD in antibody treated animals is only shown for SUM149 cells (Figure 3B), no data are shown for MDA-MB-231 cells. The relationship that NOTCH associates with a cancer stem cell phenotype is shown by sorting single cells from huN1 antibody-treated SUM-149 xenotransplanted mice and showing that they have a reduced capacity to form mammospheres (Fig 4B). Treating SUM149 xenografts with a combination huN1 antibody and docetaxel was superior in tumor growth inhibition compared to single agent treatment. These data largely based on SUM149 led to the interpretation that specific inhibition of Notch1 signaling enhances the antitumor efficacy of chemotherapy in TNBC through reduction of cancer stem cells. The second paper (PMID: 23982961) by Azzam et al published in EMBO Molecular Medicine in 2013 is certainly a very nice paper showing a positive correlation between N1ICD and SOX2 expression in a subfraction of sorted CD44⁺CD24^{low} MDA-MB-231 cells. Only N1ICD overexpression studies in MDA-MB-231 cells resulted in an increase of SOX2 mRNA expression and NICD binding to the SOX2 promoter in ChIP assays. This correlated with increased mammosphere formation and a modest increase in SOX2 protein levels, while N1ICD was not detectable in unsorted

parental MBA-MB-231 cells (Figure 5). Thus, this positive correlation between NOTCH1 and SOX2 seems to occur in a subfraction of MDA-MB-231 cells.

I trust that this referee understands that it is not our intention to discredit any of the above cited papers, but to highlight that the large body literature is not that extensive and certainly context dependent, which may explain certain differences between our and the above-mentioned studies.

Moreover, there are also more recent publications showing that a similar negative relationship between Notch signaling and SOX2 has been inferred for neuroendocrine transformation of lung and prostate cancer. There SOX2 either induces or maintains lineage plasticity towards the neuroendocrine cell state, while Notch signaling would prevent neuroendocrine tumor growth or transformation (Quintanal-Villalonga et al, 2021, 2023; Mu et al, 2017; Puca et al, 2019; Ku et al, 2024, JCI).

We further discussed this point in the manuscript on page 20.

It is not quite clear to us why the referee states that we used only one cell line to demonstrate most of our findings. It is true that the genome wide CRISPR screen was performed in ~~only~~ one cell line, in which we identified SOX2 as a potential candidate mediating resistance to a pharmacological NOTCH inhibitor. However, the reciprocal interaction between NOTCH1 and SOX2 and their consequences was shown using 4 different cell lines (MB157, MB157R, HCC1599 and HCC1806). For the revised version of this manuscript, we performed new experiments using an additional cell line BT549 (please see also point 7), confirming and validating the robustness of our findings. These new data sets are now shown as in Figure EV Fig3 H-J and mentioned in the text on page 12.

3. The papers did not use the same markers to be consistent across various models; for example, they used GAPDH as a control in one cell model and g-tubulin in the same figure panel on a different cell line. There is limited consistency regarding the EMT and stem cell markers across various experiments. The authors chose the markers based on their interests.

We used γ -Tubulin and GAPDH as loading control depending on the size of the proteins of interest. For example, SOX2 (30kDa) is too close in size to GAPDH (35kDa), so we sometimes used γ -Tubulin (50kDa) instead. Please keep also in mind that the project started more than five years ago, and different controls were used at different times. However, this does not impact our results as these loading controls are not differentially expressed under our experimental conditions.

Regarding our EMT/stemness markers, we selected specific markers that are expressed in each setting to highlight the differences. Furthermore, the RNA-seq GSEA signatures of EMT and Stemness are based on multiple makers, and their regulation aligns with the markers detected by Western blotting across all conditions. This consistency supports the validity of our marker selection and experimental results.

4. In Fig1-N, there is no significant difference in mammosphere formation ($p=0.049$) and it is truly concerning.

We do agree that the original dataset had a borderline statistical significance ($p=0.049$). However, we performed additional experiments that allowed us to add additional data points and further enhance the statistical power, making the difference even more significant, which is now shown as Fig 1N in the revised version of our manuscript.

5. The authors need to include a mammosphere formation assay for siSox2

Thank you for the suggestion. As requested, we performed tumorsphere assays in MB157R and HCC1806 after siRNA mediated knockdown of SOX2. SOX2 knockdown in both MB157R and HCC1806 caused a significant reduction in number of tumorspheres. Reduction of tumorsphere size was significant in MB157R cells, and a similar trend was observed for HCC1806 cells. These new data are now shown in Fig. EV2D and P and mentioned in the manuscript on page 9 and 10.

6. In line 2 of the introduction, the authors define tnbc as such: "malignancy which is classically defined by expression of hormone receptors (HR, estrogen and progesterone receptors) and human epidermal growth factor receptor 2 (HER2)". They could clarify the expression levels of HR and HER2 (defined by expression of...).

Thank you for pointing out this mistake. We now corrected this in the new version of the manuscript as follows on page 3: "Triple-negative breast cancer (TNBC) is the most aggressive subtype of breast cancer (BC), classically defined by lack of expression of the hormone receptors (HR, estrogen and progesterone receptors) and human epidermal growth factor receptor 2 (HER2)."

7. If their hypothesis is true, then overexpressing N1-ICD in any cell line should decrease SOX2, less EMT, and stem cell properties. The authors should use any other TNBC cells to demonstrate this phenomenon.

We demonstrated this phenomenon in two cell lines: MB157R (See Figure 4A-C, and Figure EV 3B) and HCC1806 (see Figure EV 3F-G).

To further strengthen our findings and provide additional evidence as suggested, we conducted new experiments in a third TNBC cell line, BT-549. The results are in line with results obtained with MB157R and HCC1806. These new data are now shown as Fig. EV3H-J and mentioned in the text on page 12.

8. The authors should explain their rationale behind choosing a cell line that constitutively expresses NOTCH1 in a ligand-independent manner. In the case that it is to use a cell that is NOTCH^{high}, they should also clarify why they chose that one over other cell lines overexpressing NOTCH.

The rationale of choosing a TNBC expressing a truncated, ligand independent form of NOTCH1 protein, was to use a model system that has a high NOTCH activity and whose cellular growth *in vitro* and *in vivo* has previously been reported to be Notch-driven, evidenced by the fact that they are GSI-sensitive as they still retain their transmembrane domain (Stoeck et al., 2014, Cancer Discovery and Robinson et al., 2011, Nature Medicine). Working with a TNBC cell line that is driven by a strong NOTCH allele instead of a weaker ligand-dependent allele facilitated the approach to generate GSI-resistant clones and to perform a genome-wide CRISPR screen. For example, you do not need to use a coculture system with NOTCH ligand expressing cells.

9. In page 7 line 15, the authors cited Fig. EV1D, F while they probably meant to cite Fig1 D,F.

Thank you for pointing out this mistake. We have corrected it as follows: "(Fig. 1D-F and EV1H)".

10. Conditions in Figure 1: The authors compared MB157 to MB157R which is resistant to GSI (missing the N1-ICD). When comparing MB157R to the sensitive one (MB157), it would be nice to include a comparison of EMT/Stem markers before and after the GSI treatment for MB157. That could then be compared to MB157R.

Thank you for this suggestion. We agree and added a Western blot showing SOX2, as well as EMT and stemness markers, comparing vehicle treated MB157 cells, MB157 treated with GSI for 48 hours, and 4 weeks (when resistance first appears) and MB157R. These new data have been included in the revised version of our manuscript and are now shown in Fig. EV2B and mentioned in the manuscript on page 8.

11. According to their subsequent data, the inhibition of N1-ICD should increase SOX-2 levels. Hence, the impact of GSI inhibition on MB157 (not 157R) SOX-2 levels needs to be shown.

This question is addressed by the data presented in response to questions 1 and 10, please see corresponding Figures showing Western blot analysis comparing vehicle treated MB157 cells, MB157 cells treated with GSI for 48 hours (Figure provided in response 10), and the results of a kinetic study of MB157 cells treated with GSI for 1, 2, 3 and 4 weeks (Figure provided in response 1). While GSI treatment for 48 hours is not sufficient to increase SOX2 expression, 3 weeks of continuous GSI treatment is necessary to detect increased SOX2 protein expression *in vitro*. Please also refer to answer of point 1 and corresponding *in vivo* data showing an increase of SOX2 positive cells in GSI treated MB157 xenotransplants (Fig. EV4H and the quantification is shown in Fig. 5H, page 16 of the manuscript).

12. In expanded figure 2, the labeling in the legend is out of order after D, and a figure (M) is mentioned in the legend but absent from the file.

Thank you for noticing this mistake, we corrected it.

13. In Figure 4, the authors performed a washout experiment for iN1-ICD, showing its reversible effects in MB157R. Washing out after iSox2 in MB157 cells would help to see if Sox2's effects are reversible. The same could also be said for GSI treatments.

We agree and performed the experiment as suggested. We performed a washout experiment for iSOX2 expression in MB157 cells and confirmed that the effects mediated by SOX2 are reversible upon washout. We now added these results and replaced former Figure 2F with this new Figure 2F shown underneath.

Performing the same washout experiments for GSI as suggested is not feasible. Please see Figures to point 1 and 10. GSI treatment in short term washout experiments does not work as SOX2 expression *in vitro* is not observed before 3-4 weeks of continuous GSI treatment.

14. In Figure 5 and EV5, the authors claim that: "SOX2 is sufficient to induce GSI resistance in Notch-driven TNBC, associated with increased EMT/Stemness and invasiveness". Sox2 sufficiency to induce stemness and EMT has been shown in previous figures *in vitro*. However, this figure has not shown its sufficiency to induce EMT and invasiveness (an increase in invasiveness is shown in MB157 after GSI treatment).

Our sentence " SOX2 is sufficient to induce GSI resistance in Notch-driven TNBC, associated with increased EMT/Stemness and invasiveness" (page 17), seems not to have been clear enough and was misunderstood. We did not mean to say that induced expression of SOX2 is sufficient to induce GSI resistance and to induce EMT and stemness. We therefore used the word "associated", but apparently this can still be misunderstood. We now changed the wording of our sentence for clarity and state the following:
 "SOX2 is sufficient to induce GSI resistance in Notch-driven TNBC, **which is** associated with increased EMT/Stemness and invasiveness" (page 17).

Moreover, we added a Fig EV4F, showing a quantification of ductal versus invasive growth in induced vehicle control and induced SOX2 expressing xenograft tumors showing increased invasive growth of SOX2 expressing tumors. This new Figure panel is mention on page 16 of the manuscript.

15. In EV5F/I no statistical analysis was performed on the number of metastasis.

Although we did perform the statistical analysis, we unintentionally omitted the information and did not add it to the graphs since the statistical analysis (one-way ANOVA) did not show significant differences for any of the comparisons. This information has now been added to the graphs, former EV5F/I have now become EV4 G/J.

Reviewer #3 (Remarks for Author):

The manuscript titled "Reciprocal inhibition of NOTCH and SOX2 shapes tumor cell plasticity and therapeutic escape in triple-negative breast cancer" by Morgane Fournier and colleagues takes up a vital topic in cancer therapeutics, specifically triple-negative breast cancer (TNBC). The revelation of a reciprocally inhibitory feedback loop between Notch signaling and the pluripotency-associated transcription factor SOX2 brings forth a noteworthy discovery. This has crucial implications in our understanding of the factors modulating divergent cellular states, epithelial-mesenchymal transition, and characteristics of cancer stem cells. Moreover, evaluating both monotherapies and combination drug therapies in NOTCH-inhibitor sensitive and resistant TNBC xenografts provides valuable insight. This could aid in identifying robust second-line and combination treatment strategies, which can potentially enhance tumor growth control and curtail metastatic burden. Its impact is thus underscored in the context of ongoing research. Consequentially, this manuscript should serve as a valuable resource for a range of readers. However, in its current state, the study has several shortcomings that call for extensive revisions to be considered suitable for publication in EMBO Molecular Medicine.

We want to thank this reviewer for the positive constructive remarks regarding our work. Please see underneath our efforts to address all the major comments and the majority of other comments.

1) The manuscript relies significantly on the premise of reciprocal regulation between NOTCH and SOX2 signaling directing TNBC tumour cell plasticity. However, the heterogeneity related to SOX2 expression inherent to the MB157 parental cell line (as observed in Figure 5C) forces one to wonder if MB157R is simply a selected variant of the SOX2 positive clones. Thus, authors are encouraged to investigate potential clonal heterogeneity within the parental cell line by using Immunofluorescence. If this turns out to be true, the authors should assess the N1-ICD⁺ and SOX2⁺ sub-population's plasticity and their spontaneous phenotype switching capability.

Thank you for this suggestion. We agree with the argument. Since we didn't work at the single cell level, we cannot exclude the possibility of cellular heterogeneity within the MB157 cell line. As suggested, we performed immunofluorescence staining for N1-ICD and SOX2 in *in vitro* cultured MB157 cells (please see Figure underneath). While Western blot analysis for SOX2 showed very low levels of SOX2 expression in MB157 whole cell lysates, immunofluorescence staining revealed that 7% of *in vitro* cultured MB157 cells are positive for SOX2 and 10% are positive for both SOX2 and N1-ICD, which is in agreement and consistent with our *in vivo* staining of MB157 xenograft tumors.

Unfortunately, it is not feasible to sort N1-ICD and SOX2 positive MB157 sub-populations and to assess their spontaneous phenotype switching capacity. Due to the fact that both proteins are intracellular, sorting is not feasible based on either N1-ICD⁺ or SOX2⁺ cells since the permeabilization process to label for these markers would kill the cells. Thus, when considering exclusively the immunofluorescence staining, we cannot formally exclude the possibility that MB157R cells could represent a sub-clone of MB157 cells that have resisted GSI treatment. However, we would argue this does not change the discovery of the reciprocal regulation between NOTCH and SOX2 signaling and associated plasticity phenomenon, which is strongly supported through multiple washout experiments with iNICD and iSOX2 gain-of-function, as well as SOX2 loss-of-function and pharmacological NOTCH inhibition experiments, leading to a reversible reciprocal inhibition of NICD and SOX2 in multiple cell lines *in vitro* and *in vivo*.

2) A significant observation brought forward is the evidence of reciprocal negative regulation between NOTCH and SOX2 signaling. This finding is particularly intriguing. Nonetheless, the authors seem to overlook instances where NOTCH signaling has been implicated in driving stem cell phenotypes in TNBC (refer to PMID: 31379945 as an exemplar). Therefore, I would encourage the authors to rethink their statements concerning a NOTCH <-> SOX2 driven phenotypic switch among cancer stem cell phenotypes. Such a reorientation would likely make their findings more congruous with their observations in Fig1 - indicating worse survival rates in patients exhibiting high NOTCH signatures. This adjustment would also align their findings with the subsequent survival analyses, demonstrating worst prognosis in NOTCHhigh/SOX2high patients.

Thank you for this suggestion. Please see also answer point 2 of reviewer 2.

We do not question previous studies or literature associating NOTCH with a stem cell phenotype in TNBC. One needs to keep in mind that the association of NOTCH signaling and cancer stem cells is complex, context dependent and derived from a limited number of model systems. In fact, we show that MB157 cells whose growth is Notch dependent have the capacity to form tumorspheres (Fig. 2I). Induced expression of SOX2 in MB157 cells results in a higher number and size of tumorspheres (Fig.2I). This is a SOX2 gain-of-function experiment simply indicating that SOX2 *in vitro* results in an increase tumorsphere formation, but it does not question the ability of NOTCH to be associated with cancer stem cells. In fact, SOX2 is also associated with a cancer stem cell phenotype and we show this in multiple TNBC cell line models, which have either lost Notch signaling such as the MB157R cells (EV Fig2 D) or that are SOX2 positive and NOTCH negative such as HCC1806 (EV Fig2 P) and BT-549 (EV Fig3 J). Transient siRNA mediated knockdown of SOX2 reduces tumorsphere

numbers and size of these NOTCH deficient TNBC cells (EV Fig 2D and P). Induced expression of N1-ICD in SOX2 positive TNBC cells such as MB157R, HCC1806 and BT-549 reduced number and size of tumorsphere formation (Fig. 4C, EV Fig 3 G, J). The reduced number of tumorspheres correlates with reduced expression of SOX2, mediated by induced expression of N1-ICD. Also, these results do not question the association between Notch and cancer stem cells. It rather suggests that in this particular context induced expression of N1-ICD is inhibiting SOX2 expression, which is important for tumorsphere formation and this function of NOTCH is dominant over its capacity to drive a cancer stem cell phenotype that has been shown using other TNBC cell lines.

Thus, the question is not about NOTCH <-> SOX2 phenotype switch for cancer stem cells, in fact both Notch signature positive and SOX2 positive TNBC patients show worse overall survival (OS) compared to TNBC patients that are SOX2 negative and have low NOTCH signature. There is no significant difference between the OS of patients with NOTCH^{high}-SOX2^{low}, NOTCH^{low}-SOX2^{high} and NOTCH^{high}-SOX2^{high}, all are bad and worse compared to NOTCH^{low}-SOX2^{low} patients (Fig. 5F).

The comment of the reviewer points indirectly to another interesting question namely whether TNBC patients whose tumors express higher levels of a stem cell signature (Lim stemness signature) or stem cell markers such as ALDH or CD49f may bear prognostic value for survival probability. We took up this question analyzing public patient datasets such as TCGA and METABRIC for correlations between Stemness (Lim Stemness signature or ALDH and ITGA6 stemness markers) and patient survival. We did not find any significant correlation that would support the hypothesis of reduced survival probability of TNBC patients with a high stem cell signature or expression of the investigated stem cell markers. The corresponding survival curves are shown below for the perusal of the referees.

Stemness signature (Lim_UP)

ALDH

TCGA

METABRIC

ITGA6

TCGA

METABRIC

3) The final section of the manuscript (Fig6), "Combination therapies to treat GSI-sensitive and -resistant TNBC xenografts," appears to be insufficiently connected to the mechanistic components detailed in Fig1-5. A significant gap is evidently present without a convincing explanation for the sensitivity of GSI-resistant cells to Dasatinib. Moreover, the abrupt introduction of a combination of Paclitaxel and Dasatinib creates confusion. There needs to be a clear reason for choosing this particular combination over any other potential pairing. I would urge a significant review and revision of this section to enhance narrative clarity and highlight the main takeaway.

We changed our introductory paragraph as follows to better introduce the final section of the manuscript and why and how we performed a drug screen to identify possible combination therapies with GSI and or standard of care therapeutics such as paclitaxel (page 17). Please see underneath.

“Combination therapies to treat GSI-sensitive and -resistant TNBC xenografts

One of the major limitations of targeted therapies when used as monotherapy over longer time periods is that they cause tumors resistance. The relapsing tumor is no longer

responsive to the original therapy and the therapeutic strategy requires adjustments. One way of trying to avoid adaptation of tumor cells to monotherapy is to use combination therapies and or develop second line treatment options in case the tumor has become resistant to primary therapy. In light of this argument and the fact that tumor growth regression of GSI-treated TNBC xenografts was only limited and transient, we assessed combination therapies. Since no specific small molecule inhibitors for SOX2 are available, we performed an *in vitro* drug screen with both GSI-sensitive (MB157 and HCC1599) and resistant (MB157R and HCC1806) TNBC models to discover drugs that may synergize with GSI-mediated Notch inhibition and/or standard of care.

The rational why we performed a drug screen using Notch inhibitor resistant cells and how we identified Dasatinib is described on page 18/19 as follows: please see underneath.

“We next aimed to identify small molecule inhibitors to assess second line treatment for Notch inhibitor resistant cells. A second screen for the GSI-resistant MB157R and HCC1806 TNBC cells was performed using the same commercially available libraries. Surprisingly, DTB again scored best with an IC_{50} of 271 nM in MB157R cells, which is a 20-fold increase in sensitivity compared to its IC_{50} (5.2 μ M) in MB157 cells (Fig. 6E). In contrast to GSI-sensitive MB157 and HCC1599 cells, both GSI-resistant MB157R and HCC1806 cell lines were highly sensitive to DTB *in vitro*, with induction of apoptosis and reduction of cell proliferation, associated with a decrease of the Src family kinase pathway (Fig. 6F and EV5E-G).”

To better introduce the rational of testing Dasatinib in combination with Paclitaxel we changed the wording of our manuscript as follows (page 19):

Although DTB was effective *in vitro*, it did not control tumor growth in MB157R and HCC1806 xenografts. DTB in combination with PTX, however, afforded tumor regression in MB157R xenografts, while single agent PTX only controlled tumor growth (Fig. 6G, H and EV5H, I).

To

Although DTB was effective *in vitro*, it did not control tumor growth in MB157R and HCC1806 xenografts. As Paclitaxel is used as standard of care therapy in TNBC patients we next assessed DTB in combination with PTX. Interestingly this combination afforded tumor regression in MB157R xenografts, while single agent PTX only controlled tumor growth (Fig. 6G, H and EV5H, I).

We hope that by changing the wording we have improved the clarity of our manuscript. Please note that a large-scale drug screen is always empiric and does not necessarily connect to mechanisms described in Figure 1-5. But this part of the manuscript may inspire a therapeutic avenue that might be worth considering for difficult to treat TNBC patients.

The authors could also explore and provide insights on SRC-specific signaling activity in SOX2^{high} cells, adding depth to the broader context of SOX2 and SRC regulation.

Src-family kinase (SFK) signaling is active and totally blocked by dasatinib in SOX2^{high} TNBC cells, as revealed by pSFK and pAKT Western blot, please see figure underneath. The new data has now been added to the revised manuscript as Figure EV5G and is mentioned in the manuscript on page 19.

Importantly, for completeness and credibility, consider references such as PMID: 23009336 on SRC signaling's role in modulating SOX2.

Thank you for pointing out this paper. This and other papers provide correlative evidence that SOX2 expression might be regulated by SRC kinase activities. We considered this possibility a while ago and tested if the expression of SOX2 in our TNBC cell lines might be dependent on SRC kinase activity. This was however not the case, which is the reason we did not continue investigating this possibility and hence have not cited this paper or other papers in that direction.

4) To validate the study, it would be prudent to include a repeat of essential experiments utilizing the HCC1599 cell line.

We have already included data from the HCC1599 cell line in the majority of our experiments:

- GSI IC50 (Fig. EV2F); Cell growth inhibition, NOTCH target genes mRNA expression, N1-ICD and MYC protein expression with GSI treatment (Fig. EV2G, H, I); SOX2, N1-ICD and N-cadherin protein expression after SOX2 gain-of-function (Fig. EV2K); Cell growth inhibition after SOX2 gain-of-function with GSI treatment (Fig. EV2L)
- N1-ICD, SOX2, EMT and stemness markers protein expression (Fig. EV3A)
- HE staining and N1-ICD and SOX2 co-immunofluorescence of MIND xenograft (Fig. 5B,C,D)
- Tumor growth, ductal/invasives tumor cells %, lung metastasis number, N1-ICD, SOX2 and CD49f staining of MIND xenograft (Fig. EV4A, B, C, D); Tumor growth and lung metastasis number of MIND xenograft with SOX2 gain-of-function and GSI treatment, (Fig. EV4I, J); Tumor growth of MIND xenograft after long term GSI treatment (Fig. EV4K)
- Synergy growth inhibition, apoptosis and proliferation analysis after GSI and DTB treatment (Fig. EV5A, B); Tumor growth and lung metastasis number of MIND

xenograft after combination treatments (Fig. EV5C, D); Cell growth after GSI or DTB treatment (Fig. EV5E)

Furthermore, we have now added mRNA expression data for *NOTCH1*, *NOTCH* target genes, and EMT and stemness markers after gain-of-function of *SOX2* in HCC1599 (FIG EV2J).

Additionally, we have included data for stemness (*CD49f*, *Claudin3*) and EMT (*SLUG*) markers at the protein level after gain-of-function of *SOX2* in HCC1599 (FIG EV2K).

Finally, we have also performed a Co-IP experiment for *RBPJ*-*SOX2* to further support the mechanistic aspects of our study for the perusal of the referee.

5) The authors must provide the data underlying high-throughput analysis methods (Crisp-screens as well as drug screening) in supplementary data, matching common good scientific practice now a day (as it is the case also for NGS data).

We agree. The CRISPR screen data have been submitted to GEO (accession number GSE270368) and are included in Expanded View Table 3. Additionally, we have added the drug screening data in Expanded View Table 5.

6) The findings via ChIP sequencing should be corroborated through different methods; ChIP-qPCR could prove useful for cross-verifying crucial binding sites.

We added ChIP-qPCR for cross validation on crucial sites identified in the main figure 3 as suggested. Specifically, we validated peaks on the *NOTCH1* promoter and the *HES1* promoter in MB157 iEV cells, MB157 iSOX2 cells, MB157R iEV cells and MB157R iN1-ICD cells after chromatin immunoprecipitation for HA-tag, *SOX2* or *RBPJ*, please see underneath. These data are now added as Appendix Fig S2.

7) To confirm the reciprocal regulation between NOTCH and SOX2 and substantiate their experimental methodology, authors should consider overexpressing transcriptionally deficient forms of N1-ICD and SOX2. Subsequently, they must demonstrate that the cell phenotypes and representative target genes are not affected.

After consultation with the editor, we mutually agreed that this point does not need to be addressed.

8) To robustly support the manuscript's claims about reciprocal negative regulation between NOTCH and SOX2, clinical staining from a TNBC patient collective should be included.

To this reviewer request we purchased a commercially available Tissue Micro-Array (TMA) derived from TNBC patients. This TMA included 79 TNBC patient samples. We performed immunofluorescence for N1-ICD and SOX2 co-staining and quantified N1-ICD and SOX2 positive samples for all 79 patient samples. 16.5% of the TMA samples showed positive staining for either N1-ICD (7.6%) or SOX2 (8.9%). The staining was mutually exclusive, N1-ICD positive samples were negative for SOX2 and SOX2 positive samples were negative for N1-ICD staining, supporting the reciprocal negative regulation between NOTCH and SOX2. We did not observe samples that co-stained for both N1-ICD and SOX2 within these limited numbers of TMAs. Please see representative pictures of the TMA staining and the quantitative analysis underneath. These new data have now been added as Figure 5E and are mentioned in manuscript on page 15.

Other comments:

1) Manuscripts should be provided with line numbers to facilitate the reviewing process

We added line numbers to the manuscript as requested.

2) Please improve the comprehensibility of the main text and employ a standard normalization method to retain the control conditions' variance across all graphs.

Thank you for pointing this out. We agree and have standardized the names of the control conditions as iEV (induced empty vector control) and VHC (vehicle) to ensure consistency across the entire manuscript.

3) The manuscript conflicts with earlier publications mentioning the pro-tumorigenic implications of NOTCH signaling in HCC1806 cells (PMID: 23012411). Primarily, this discrepancy occurs in the context of the observations made in figure EV4D-G. The authors should discuss/state these contradicting results.

We do not believe that the results are necessarily contradictory to the Lombardo's paper as we have not performed the same experiments. The manuscript by Lombardo et al published

in PNAS in 2012 focuses on the role of Nicastrin in breast cancer and not directly on Notch signaling itself. Nicastrin can regulate NOTCH but also many other targets.

In our view and in our hands HCC1806 TNBC cells are not a good model system for studying NOTCH-driven growth of TNBC compared to MB157 or HCC1599 for following reasons:

1. They do not have genetic aberrations such as point mutations or inter-chromosomal deletions that would result in constitutive expression of N-ICDs, a strong NOTCH allele or Notch signature.
2. Consequently N1-ICD levels are very low and not easy to detect by Western blot analysis (please see Fig. EV2O, and Fig. EV3A and Lombardy et al Fig. 1C and FigS1, N-ICD bands are always faint).
3. *In vitro* growth of HCC1806 is not affected by GSI-mediated inhibition of Notch signaling even at concentration of 10 μ M (please compare Fig EV2 M (HCC1806) with Fig 1D and Fig EV2G (NOTCH-dependent cell line MB157 and HCC1599)).

The potential discrepancy mentioned by the reviewer refers to the tumorsphere assays which are used as surrogate marker for stemness. The tumorsphere assays have been performed by both Lombardo et al and us using HCC1806 cells. Please note that the experimental set up and questions are very different. Lombardo et al performed tumorsphere assays with HCC1806 cells that expressed either sh-Luc as control or sh for Nicastrin. In their Figure 3C and D they show that HCC1806 cells with sh-mediated Nicastrin depletion have reduced sphere-forming efficiency compared to control cells. In previous experiments the authors showed that knockdown of Nicastrin in HCC1806 cells correlates with reduced N-ICD expression and other markers such as p-AKT (Figure 1C). Based on this correlation, the interpretation of this reviewer warrants that the reduced tumorsphere formation is most likely mediated through reduced Notch signaling, which might be one plausible possibility. However, no N-ICD rescue experiments in Nicastrin depleted HCC1806 cells were performed to demonstrate that tumorsphere formation is indeed driven by Notch signaling in this setting. Since Nicastrin regulates additional targets to NOTCH, it could be a NOTCH-independent effect, for example via CD44 which is an important stem cell marker and a target of Nicastrin.

We also performed tumorsphere formation assays using HCC1806 cells and show that these cells have the capacity to form tumorspheres, which is in agreement with the Lombardo paper. Please note that we have not performed any specific loss-of Notch-function experiments for tumorsphere formation and therefore cannot comment on whether tumorsphere formation is Notch dependent. We used HCC1806 cells as an example for SOX2 expressing TNBC cells to show that inducible expression N1-ICD results in reduced SOX2, which is in agreement with our reciprocal negative feedback loop model for Notch and SOX2. Inducible expression of N1-ICD in HCC1806 cells results in a 35% reduction of tumorsphere numbers and a slightly reduced tumorsphere size. Our finding does not contradict the results of the Lombardo paper nor does it rule out the possibility that tumorsphere formation might still be dependent on Nicastrin or maybe even Notch signaling. If HCC1806 cells require active Notch signaling for tumor formation then both results would be compatible with the interpretation that you need an optimal concentration of N-ICD for tumorsphere formation. Consequently, in situations, in which N-ICD levels might be too low or too high you interfere with the ability to form optimal number and size of tumor spheres. Based this line of reasoning we do not agree that our results are necessarily contradictory and that it is suitable to discuss this in our manuscript.

4) Fig1H: It would be interesting to also test the transcriptional activity of RBPJ/N1-ICD in luciferase assay prior figure 3F to support Notch signaling loss in MB157R as well as upon GSI treatment in this same system.

The editor informed us, upon further cross-commenting with the referees, that this point does not need to be addressed.

5) To substantiate their data further, I encourage the authors to reanalyze publicly available scRNA-seq data of TNBC patients, notably showing negative correlation of SOX2 expression with the NOTCH signature in single cells, and showing negative correlation of HEY2 and HEYL on SOX2 expression.

Thank you for the suggestion. To investigate the correlation of SOX2 and NOTCH1 expression at the cellular level, we leveraged publicly available single-cell RNA sequencing (scRNA-Seq) data from various breast cancer studies. The raw count matrices were downloaded from the Gene Expression Omnibus (GEO) database. These matrices were then used to generate Seurat objects, which were normalized using the “NormalizeData” function in Seurat to correct for sequencing depth and other technical variations. Next, we filtered the data based on metadata information to focus exclusively on triple-negative breast cancer (TNBC) cells. In this subset, both NOTCH1 and, particularly, SOX2 had low expression counts. Scatter plots of the data highlighted a significant number of dropouts, where expression is detected for one gene but not the other. While this observation could fit the hypothesis of SOX2 and NOTCH1 expression being negatively correlated (or even mutually exclusive), it is crucial to note that dropouts are a well-known issue in scRNA-Seq data due to the stochastic nature of RNA capture and amplification. Thus, a zero count does not definitively indicate the absence of gene expression but rather could be a result of technical limitations, such as low sequencing coverage per cell. To address this issue, we refined our analysis by including only cells that had at least one read for both SOX2 and NOTCH1. This approach aimed to reduce the impact of technical dropouts on our correlation analysis. We performed a Spearman correlation test, which showed a statistically significant negative correlation between SOX2 and NOTCH1 expression ($p = 0.011$). Nevertheless, caution is advised in interpreting these results due to the low number of cells and read counts, which might affect the robustness of the findings.

As a consequence of the limited robustness of the scRNAseq available data sets and their analysis we chose not to include these data, although they would fit our hypothesis.

However, we included the co-IF data from the TMA as suggested in point 8, showing a negative correlation between SOX2 and N1-ICD protein expression at single cell resolution.

6) Please make the supplementary file #2 available in its original XLS file format, as the PDF version is not user-friendly.

We agree and will upload Supplementary File #2 in XLS file format.

7) Please provide the marker's size for all Western blots in this study (not the band size); also consider including quantification data of the western blot replicates.

We will provide the marker size for all Western blots available by uploading the original western blot as source data, which will be made available as online information.

We performed all Western blots in biological triplicates to ensure the reliability of our results. The results were consistent between the different Western blots of the biological replicates. Since the differences in Western blot band intensities were very clear and distinct, we opted not to include the quantification data of individual bands.

8) The scale bars' size should be presented consistently across all figures, preferably in micrometers.

All scale bars are indeed presented in micrometers (μm). To clarify and increase consistency across the figures, we have standardized the scale bars for each type of results as follows:

- Cell lines $50\mu\text{m}$
- Tumorspheres $100\mu\text{m}$
- IHC and H&E $100\mu\text{m}$
- IF $50\mu\text{m}$
- TMA $100\mu\text{m}$

Moreover, the information regarding the size of the scale bars are mentioned in every corresponding figure legend.

9) Figure 1C: Alter the color code for splicing events, and amend the figure legend for clarity.

For better clarity, we have changed the color code of the splicing events of the Sashimi plot to grey. The reads are depicted in blue for contrast and the legend has been amended accordingly.

10) Figure 1D: An estimation of IC_{50} for both MB157 and MB157R should be included in the results.

We have addressed the IC_{50} of GSI in MB157 in Fig. EV1H. For MB157R, there was no significant growth difference observed at $10\mu\text{M}$ compared to the vehicle control (DMSO) (Fig. 1H). Therefore, the estimated IC_{50} for MB157R is greater than $10\mu\text{M}$, which is typically considered indicative of low sensitivity or resistance to the drug *in vitro*. In more detail we addressed this comment under point 30 with a table of all IC_{50} of the different cell lines for GSI, Dasatinib and Paclitaxel from the drug screen, which are now provided as expanded table 5 in the revised manuscript.

11) Fig1G: Add the missing error bars and statistics to improve the representation.

Fig. 1G is representative of three independent biological replicates.

However, due to different growth kinetics between the replicates (the resistance did not emerge exactly at the same time for all the replicates), the cells were split on different days. As a result, the data points cannot be pooled and statistics cannot be performed. For transparency, we have included the data from the two other replicates (REP1 and REP2) for the perusal of the referee. Unfortunately, REP1 cells died after 13 weeks in culture, likely due to their fragility from continuous GSI treatment as they began to develop resistance.

12) Fig1K: Slight confusion could arise from the current graph title. Redesign the annotations with "Enriched in MB157R" and "Enriched in MB157" to avoid potential misinterpretation (comment eventually also applicable to other bubble plots of this manuscript).

Thank you for pointing this out. We agree with your suggestion and have updated the annotations. The left side of the graph now reads "Enriched in MB157" and the right-side reads "Enriched in MB157R" to clearly differentiate the two groups and avoid any potential misinterpretation. We have applied this change to the other lollipop plots in the manuscript as well.

13) Please revise all instances referring to "Mammosphere" to "tumorsphere" throughout the text and the figures.

We have changed all instances of "mammosphere" to "tumorsphere" throughout the text and figures, as requested.

14) Fig. 2F: The authors should provide RT-qPCR of NOTCH levels upon SOX2 overexpression.

This information has already been provided in our first submission in Fig. EV2C and is now again shown in the revised version of our manuscript in Fig EV2E.

15) Fig2I: The images appear to be provided with different zooms, skewing the cell size representation. The iSOX2 image showcases sphere aggregates. To get a more objective illustration, it is recommended for authors to instigate the experiment with conditions that discourage cell aggregation. This could be achieved by using lower cell number, or possibly incorporating methyl cellulose.

Thank you for this observation and your comment. We would like to clarify that the images were not taken at different zoom levels. We followed established protocols for this experiment, which have been documented in Lombardo et al., 2015 (doi: 10.3791/52671) and tested different cell number to optimize the conditions with our cell lines. The aggregation is only observed for overexpression of SOX2 in MB157 and is consistent with previously published tumorsphere formation assays, where different cell lines show distinct tumorsphere phenotypes. We believe this accurately represents the phenotypes being studied. As an example of different tumorsphere phenotype depending on the cell lines used, you may refer to Shaw et al., 2012 (doi: 10.1007/s10911-012-9255-3).

16) Fig3D: The ChIP-seq tracks require the addition of units and scale for better comprehension.

The scale was already indicated on the y-axis, but you are correct that the unit was missing. Thank you for spotting this. We have now added the unit, "reads per million mapped reads," to the figure legend. The normalized bigWig files are scaled to 1 million mapped reads to ensure that the coverage can be compared accurately across multiple samples.

17) Fig3E: Inputs are missing and need to be incorporated.

We have added the inputs to Fig 3E as requested. Please note that the inputs are shown with a different exposure than the IPs to ensure good visualization of the bands.

18) To validate the statement pillared on findings, "These findings strongly corroborate a working model where SOX2 binding in close vicinity to RBPJ does interfere with the formation of a functional N1-ICD/RBPJ activation transcription complex thus repressing Notch signaling," it is suggested to execute competition binding assays. This can be achieved by progressively increasing either SOX2 or N1-ICD amounts and evaluating the interaction shift with RBJP.

The editor informed us, upon further cross-commenting with the referees, that this point does not need to be addressed.

19) "BC patient dataset analysis using The Cancer Genome Atlas (TCGA) and the Molecular Taxonomy of Breast Cancer International Consortium (METABRIC) revealed an enrichment for activating mutations and gene amplifications of NOTCH receptors in TNBC compared to HR+ and HER2+ patients (Fig. EV1A,B), indicating that tumor growth within these subgroups could be Notch-driven.": it is not clear what "subgroups" are meant here.

Thank you for pointing this out. We have now clarified this sentence as follows and hope to have improved clarity: "BC patient dataset analysis using The Cancer Genome Atlas (TCGA) and the Molecular Taxonomy of Breast Cancer International Consortium (METABRIC) revealed an enrichment for activating mutations and gene amplifications of *NOTCH* receptors in TNBC compared to HR⁺ and HER2⁺ patients (Fig. EV1A, B). The identification of genetic *NOTCH* alterations might be indicative for NOTCH-driven tumor growth, independently of the different BC subgroups (HR⁺, HER2⁺ or TNBC)".

20) Fig4D: the authors should improve consistency of their study: ChIP-seq focused on HES1 but the subsequent analyses important for SOX2 regulation focus on HEY2, HEYL, HES5. The authors should therefore provide the respective tracks and validate their observations via other methods (e.g. ChIP-qPCR).

We agree on the importance of consistency in our study and have provide the ChIP-seq tracks for *HEY2*, *HEYL*, and *HES5* as requested for your pursual. However, we did not observe any significant SOX2 binding peaks in close vicinity to RBPJ peaks within the *HES5* gene. Significant SOX2 peaks were observed in the vicinity of RBPJ peaks within intergenic

region of the *HEYL* gene. Significant but somewhat less convincing SOX2 peaks were also observed in close vicinity to RBPJ peaks within the promoter region of the *HEY2* gene. While promoters and enhancers for these genes are not well characterized in this context, it is possible that relevant peaks exist elsewhere. For example, there is a SOX2 peak in close proximity to RBPJ approximately 140 kb upstream of the *HEY2* gene, which could correspond with an endothelial enhancer previously described at around -161 kb (Sissaoui S. et al, 2020, Circ Res. PMID: 32065070).

HESS

HEYL

HEY2

HEY2 – 140kb upstream

21) Fig4D-E: The data presented would strongly benefit from additional verification of the binding of HEY2 and HEYL to the SOX2 regulatory region. Using a method such as ChIP-qPCR for this analysis is recommended. Furthermore, to evaluate the impact of HEY2 and HEYL, an experiment where these are overexpressed in MB157R and/or HCC1806, with an expected result of a downregulation in SOX2 mRNA levels, should be performed. This would provide further support for your findings.

The editor informed us, upon further cross-commenting with the referees, that this point does not need to be addressed.

No good antibodies for HEY2 or HEYL have been developed for ChIP that would allow to perform this experiment.

22) Fig5E: The take-home message from the METABRIC analysis needs to be clarified for better understanding and learning.

We have clarified the text to better convey the key findings: “Among these subgroups, patients characterized by a NOTCH^{High} signature and SOX2^{High} expression had the worst OS and RFS rates, whereas those with a NOTCH^{Low} signature and SOX2^{Low} expression showed the most favorable prognosis (Fig. 5F and EV4E). Taken together this analysis shows that

active Notch signaling and high levels of SOX2 can coexist within the same tumor, and associates with poor prognosis, similar to patients whose TNBC tumors express a Notch^{High} signature only or SOX2 only.” (page 15). Please not former Fig. 5E and EV5E has become Fig. 5F and EV4E in the revised version of our manuscript.

23) Fig5F: Alter the color code to improve comprehension. Differences between light grey and white could be challenging to discern.

Thank you for your suggestion. We have updated the color code in Fig. 5F to improve clarity and ensure that the differences between the data points are more easily distinguishable. Please not Fig. 5F has become Fig. 5G in the revised version of our manuscript. Moreover, as the previous color code corresponded to specific conditions such iEV, iSOX2 etc, we had to adapt this new color code for consistency reasons to following panels: Fig. 5I, EV Fig. 4 G, I, J and K. We hope that having applied the new color code we have improved the visibility of different data points.

24) Fig5H seems to be an extension of Fig5F. Hence, authors should consider providing information about what happens with iSOX2 transplanted tumors over longer period of time.

Point well taken. We would like to clarify that Fig5F and Fig5H are not from the same experiment. However, we do have additional data from a separate experiment with iSOX2 transplanted tumors at a longer time point (shown below for the perusal of the reviewers). As shown in Fig 5G (former Fig 5F) tumor growth of iSOX2 transplants is somewhat accelerated compared to transplants of iEV control, at short time points. However, at longer time points, as shown by the graph underneath, tumor growth of iSOX2 transplants is comparable to transplants of vehicle treated iEV control tumors.

25) Fig6B: Please provide an explanation for DTB MFI.

We have clarified this point in the Materials and Methods section: “Cell Trace Violet Mean Fluorescence Intensity (CTV MFI) was measured to determine the proliferation index. A low CTV MFI indicates a higher proliferation rate, and vice versa.” The data of Fig. 6B show that GSI and GSI + DTB affect proliferation as indicated by increased CTV MFI compared to DMSO control. In contrast DTB alone seem not to affect proliferation as its CTV MFI is

comparable to DMSO control, while it seems to modestly increase apoptosis (Fig.6 B, left panel), which might explain the modest reduction in *in vitro* cell growth when used as single agent (Fig. 6A, right panel).

26) Fig 6C: Please indicate why the 10-week period for GSI alone is not represented.

We had to euthanize the mice because they were close to reaching the humane endpoint based on tumor size and to be in agreement with the regulations of our animal license. Moreover, the GSI treatment alone was no longer sufficient to control tumor growth. This is why the 10-week period for GSI alone is not represented in the figure.

27) Fig 6F: It is not clear why GSI and DTB concentrations used in MB157R are higher than those for MB157, even though IC₅₀ is lower in this cell line. Please explain or correct the graphs accordingly.

Thank you for your observation. The combination treatment of MB157 cells shown in Fig 6A is based on a drug synergy matrix screen with the goal to show and use the lowest possible drug concentrations that would still show a synergistic effect.

In contrast, for MB157R (Fig 6F) the goal was to use a DTB concentration that induces a strong effect when used as single agent treatment to clearly demonstrate its efficacy.

We agree that using a lower concentration might still show an effect in MB157R. Indeed, the IC₅₀ of DTB for MB157R is lower than for MB157. However, we would like to highlight that DTB alone is insufficient *in vivo* to control tumor growth.

28) "Efficient targeting of the Notch pathway could, therefore, be beneficial for NOTCHHigh signature TNBC patients." The authors must respect good scientific practice rules and cite the relevant to date available literature! As example, recent publications related to this topic (PMID: 38172915) has to be mentioned (best already in the introduction) and accordingly discussed!

Thank you for pointing out the importance of citing relevant and recent literature. You may have overlooked this, but the reference you mentioned (Braune et al 2024) was cited and discussed in our original submission in the discussion section on page 20, where it aligned well with our results. Based on your recommendation, we have now also cited this paper earlier in the introduction, to ensure that our work is contextualized within the current state of research.

29) Typo: "thwarted"

We have reviewed the usage of the word "thwarted" and confirm that it is used correctly in the context of our manuscript.

30) The authors should provide the graphs of IC₅₀ estimation for the inhibitors utilized in this study, according to the cell system.

We added below a table summarizing the IC₅₀ values from the drug screen (see expanded table 5) of all the drugs for the different cell lines used in this study.

Cell lines	IC ₅₀ GSI	IC ₅₀ DTB	IC ₅₀ PTX
MB157	17.7nM	123.9µM	4.4nM
HCC1599	43.7nM	2.3µM	8.7nM
MB157R	108.1µM	5.8nM	4.6nM

HCC1806	4.6μM	95.5nM	3.2nM
---------	-------	--------	-------

31) Typo "CCDN1"

Thank you for spotting this typo. We have corrected the gene name to *CCND1* in the manuscript.

32) Consistency "(Fig. S4H)" ↓ EV4H

Thank you for pointing out the inconsistency in our figure references. We have corrected the mistake, and the figure is now correctly cited as Fig. EV3K (former Fig. EV4H).

7th Oct 2024

Dear Prof. Radtke,

Thank you for submitting your revised study, and please accept my apologies for the delay in getting back to you as one referee needed more time to complete his/her review. We have now received the reports from the two referees who evaluated your revised manuscript, and as you will see from the reports below, they are overall satisfied with the revisions. I will therefore be able to accept your manuscript once the following issues will be addressed:

1/ Please address the remaining minor concerns raised by referee #3.

2/ Manuscript text:

- Please remove the highlights in the text and only keep in track changes mode any new modification.
- "Summary" should be renamed "Abstract".
- Methods:
 - o Thank you for providing the Reagents and tools table. Please remove it from the manuscript and upload it as a separate file.
 - o Cells: please indicate whether the cells were authenticated and tested for mycoplasma contamination.
 - o Antibodies: please provide dilutions/concentrations.
 - o Animals: please provide the housing/husbandry conditions.
- Author contributions: CRediT has replaced the traditional author contributions section because it offers a systematic machine readable author contributions format that allows for more effective research assessment. Please remove the Authors Contributions from the manuscript and use the free text boxes beneath each contributing author's name in our system to add specific details on the author's contribution. More information is available in our guide to authors.
- Please update the "Declaration of interest" statement to 'Disclosure statement and competing interests' (<https://www.embopress.org/competing-interests>).
- The info about BioRender should be removed from the Acknowledgements and added to the Methods in a section with the title "Graphics" and following the format: Graphics: The visual abstract and schemes graphics were created with BioRender.com.

3/ Figures and Appendix:

- Dataset EV Tables 2 - 5 should be renamed "Dataset EV1 - 4" and a legend should be added to each dataset in a separate tab/worksheet
- Appendix: please include a table of content with page numbers. Please correct the nomenclature to "Appendix Figure S1" etc.
- EV Table 6 should be renamed "Table EV2". The callouts in the manuscript text will need to be updated accordingly.
- Please address the queries from our copy editors in the figure legends:
 1. Please note that the exact p values are not provided in the legends of figures 1d, f, m-n; 2i; 3b; 4c-d; 5h; 6a-b, d, f, h; EV 1j; EV 2c, g-h, j, n; EV 3j-l; EV 4f; EV 5e.
 2. Please indicate the statistical test used for data analysis in the legends of figures 1k; 2b, h; 3b; 4b; 6f; EV 1i.
 3. Please note that the p value is not represented in the figure 1h; 5i; however statistical test related information is provided in the legend of the corresponding figure. This needs to be rectified.

4/ Source Data:

Please check the following:

- Potential duplication in Figure 1A
- Blot in Figure 1L
- Blot N1-ICD missing in Figure 2D
- SD missing for Figure 3B

5/ Checklist:

- Please fill in the section Cell Materials / authentication and mycoplasma contamination.
- Please fill in the section Experimental study design and statistics / inclusion-exclusion criteria.

6/ Synopsis:

I introduced minor modifications in your synopsis to fit our format. Please let me know if you agree with the following, or amend as you see fit:

"A NOTCH1-SOX2 feedback mechanism was linked with resistance to GSI. SOX2-induced inhibition of Notch pathway in NOTCH-driven TNBC is associated with EMT and cancer stem cell features. Preventive GSI-PTX and second line DTB-PTX combination therapies reduced tumor growth and metastasis in TNBC.

- SOX2 induces tumor cell plasticity, characterized by EMT and cancer stem cell traits, causing resistance to GSI-mediated inhibition of Notch signaling

- SOX2 inhibits Notch signaling by binding to NOTCH1 and NOTCH target gene promoters and/or enhancers and interaction with RPB1
- Reciprocally, Notch signaling represses SOX2 expression via its target genes of the HEY family
- GSI-PTX and DTB-PTX combination therapies induce tumor growth control and reduce metastatic burden in stratified NOTCH-dependent and -independent TNBC, respectively .

Thank you for providing a nice synopsis picture. I have cropped a small portion (115x70 px) to serve as thumbnail for the table of content on our webpage (attached). Please let us know if you agree with this thumbnail, or provide another one in the same dimensions.

Please note that these would be the final versions as changes during proofing are usually not allowed

7/ As part of the EMBO Publications transparent editorial process initiative (see our Editorial at <http://embomolmed.embopress.org/content/2/9/329>), EMBO Molecular Medicine will publish online a Review Process File (RPF) to accompany accepted manuscripts.

This file will be published in conjunction with your paper and will include the anonymous referee reports, your point-by-point response and all pertinent correspondence relating to the manuscript. Let us know whether you agree with the publication of the RPF and as here, if you want to remove or not any figures from it prior to publication.

I look forward to receiving your revised manuscript.

With kind regards,

Lise Roth

***** Reviewer's comments *****

Referee #2 (Remarks for Author):

The authors have addressed all the concerns.

Referee #3 (Remarks for Author):

The manuscript, "Reciprocal inhibition of NOTCH and SOX2 shapes tumor cell plasticity and therapeutic escape in triple-negative breast cancer", authored by Morgane Fournier and colleagues, has been reevaluated after the authors addressed the previously raised comments. An examination of their revisions has demonstrated that they have effectively resolved most of the concerns, significantly improving the overall quality of their research. At this juncture, I want to express my acknowledgement of the considerable effort they have invested into this. Nevertheless, several minor points remain that should be addressed to further fortify the scientific robustness, rigor, and relevance of the manuscript prior to publication. The specifics of these points are delineated as follows:

In response to Major Comment #1: The authors' explanation regarding the heterogeneity of the MB157 cell-line is well-reasoned and satisfactory. To enhance clarity and offer a more comprehensive understanding of the study implications, it would be beneficial to include this newly presented data on the intrinsic heterogeneity of the MB157 cell-line in the supplementary figures. An explicit discussion of this element in the main text would also be advantageous. Such an addition would serve not only to elucidate the reported experimental phenomena more clearly but also to emphasize the significance of the identified reciprocal regulation between NOTCH and SOX2 signaling. Furthermore, addressing the methodological challenges encountered, such as the complexity of sorting cells based on intracellular markers, could highlight the inherent difficulties in conducting this type of scientific investigation.

In response to Major Comment #2: The authors have provided a detailed, albeit somewhat complex, response. I brought up this

comment initially with the intention of enhancing the comprehensibility of the manuscript. However, I continue to stress the importance of including a more exhaustive discussion about the well-documented role of NOTCH signaling in driving stem cell phenotypes in TNBC, as demonstrated in numerous previous studies. This consideration is essential to adequately align the current work with the existing literature, a viewpoint that Reviewer #2 also shares.

Point by point response to referees' comments and summary of revisions to manuscript: EMM-2024-19945_R2

We would like to thank all reviewers and the editors for their overall positive evaluation of our manuscript, as well as their insightful and helpful comments, which have helped to improve our study.

Please find below a detailed point-by-point response to the individual comments:

Dear Prof. Radtke,

Thank you for submitting your revised study, and please accept my apologies for the delay in getting back to you as one referee needed more time to complete his/her review. We have now received the reports from the two referees who evaluated your revised manuscript, and as you will see from the reports below, they are overall satisfied with the revisions. I will therefore be able to accept your manuscript once the following issues will be addressed:

1/ Please address the remaining minor concerns raised by referee #3.

Point 1: We added the co-immunofluorescence staining of N1-ICD and SOX2 for MB157 and MB157R cells cultured in vitro as requested to describe the cellular heterogeneity of corresponding cell lines. These data are now shown as Fig. EV4 E and mentioned in the results section on page 14 &15.

Point 2: We added a small paragraph and corresponding references regarding the association of Notch signaling and Sox2 for stem cell properties in breast cancer to the discussion section (page 21).

16th Oct 2024

Dear Prof. Radtke,

Thank you for submitting your revised files. I am pleased to inform you that your manuscript is accepted for publication and is now being sent to our publisher to be included in the next available issue of EMBO Molecular Medicine.

If you have any questions, please do not hesitate to contact the Editorial Office.

Thank you for your contribution to EMBO Molecular Medicine!

With kind regards,

Lise Roth
